



# Investigation of the global methane budget over 1980-2017 using GFDL-AM4.1

Jian He[1,2], Vaishali Naik[2], Larry W. Horowitz[2], Ed Dlugokencky[3], and Kirk Thoning[3]

[1]Program in Atmospheric and Oceanic Sciences, Princeton University, Princeton, New Jersey, USA
5   [2]NOAA Geophysical Fluid Dynamics Laboratory, Princeton, New Jersey, USA
[3]NOAA Earth System Research Laboratory, Boulder, Colorado, USA

*Correspondence to*: Jian He (jian.he@noaa.gov)

**Abstract.** Changes in atmospheric methane abundance have implications for both chemistry and climate as methane is both a strong greenhouse gas and an important precursor for tropospheric ozone. A better understanding of the drivers of trends and variability in methane abundance over the recent past is therefore critical for building confidence in projections of future methane levels. In this work, the representation of methane in the atmospheric chemistry model AM4.1 is improved by optimizing total methane emissions (to an annual mean of 576±32 Tg yr$^{-1}$) to match surface observations over 1980-2017. The simulations with optimized global emissions are in general able to capture the observed global trend, variability, seasonal cycle, and latitudinal gradient of methane. Simulations with different emission adjustments suggest that increases in methane sources (mainly from energy and waste sectors) balanced by increases in methane sinks (mainly due to increases in OH levels) lead to methane stabilization (with an imbalance of 5 Tg yr$^{-1}$) during 1999-2006, and that increases in methane sources combined with little change in sinks (despite small decreases in OH levels) during 2007-2012 lead to renewed methane growth (with an imbalance of 14 Tg yr$^{-1}$ for 2007-2017). Compared to 1999-2006, both methane emissions and sinks are greater (by 31 Tg yr$^{-1}$ and 22 Tg yr$^{-1}$, respectively) during 2007-2017. Our results also indicate that the energy sector is more likely a major contributor to the methane renewed growth after 2006 than wetland, as increases in wetland emissions alone are not able to explain the renewed methane growth with constant anthropogenic emissions. In addition, a significant increase in wetland emissions would be required starting in 2006, if anthropogenic emissions declined, for wetland emissions to drive renewed growth in methane, which is a less likely scenario. Simulations with varying OH levels indicate that 1% change in OH levels could lead to an annual mean of ~ 4 Tg yr$^{-1}$ difference in the optimized emissions and 0.08 year difference in the estimated tropospheric methane lifetime. Continued increases in methane emissions along with decreases in tropospheric OH concentrations during 2008-2015 prolong methane lifetime and therefore amplify the response of methane concentrations to emission changes. Uncertainties still exist in the partitioning of emissions among individual sources and regions.



# 1 Introduction

Atmospheric methane ($CH_4$) is the second most important anthropogenic greenhouse gas with a global warming potential 28-34 times that of carbon dioxide ($CO_2$) over a 100-year time horizon (Myhre et al., 2013). Methane is also a precursor for tropospheric ozone ($O_3$) - both an air pollutant and greenhouse gas - influencing ozone background levels (Fiore et al., 2002). Controlling methane has been shown to be a win-win for both climate and air quality (Shindell et al., 2012). From a preindustrial level of $722\pm25$ ppb (Etheridge et al., 1998; Dlugokencky et al., 2005), methane has increased by a factor of

~2.5 to a value of $1850\pm1$ ppb in 2017 (Dlugokencky et al., 2018), mostly due to anthropogenic activities (Dlugokencky et al., 2011). The global network of surface observations over the past 3-4 decades indicates that methane went through a period of rapid growth from the 1980s to 1990s, nearly stabilized from 1999 to 2006, and then renewed its rapid growth. Studies of the drivers of observed changes in methane trends and variability have focused on the contributions from changes in methane sources and sinks. Here, we apply a prototype of the new generation NOAA Geophysical Fluid Dynamics

Laboratory chemistry-climate model, GFDL-AM4.1 (Zhao et al., 2018a, b; Horowitz et al., manuscript in preparation) with an improved representation of methane, to explore the contributions of methane sources and sinks to its observed trends and variability.

Methane is emitted into the atmosphere from both anthropogenic activities (e.g., agriculture, energy, industry, transportation, waste management, and biomass burning) and natural processes (e.g., wetland, termites, oceanic and geological processes,

and volcanoes), and is removed from the atmosphere mainly by reaction with hydroxyl radical (OH) in the troposphere, with less contributions to destruction by reactions with excited atomic oxygen ($O(^1D)$) and atomic chlorine (Cl) in the stratosphere and uptake by soils. Measurements of the global distribution of surface methane beginning in 1983 have revealed that atmospheric methane approached steady state during 1983-2006 and renewed its growth since then. During 1983-2006, methane growth rates decreased from 12 ppb $yr^{-1}$ during 1984-1991 to 5 ppb $yr^{-1}$ during 1992-1998 (Nisbet et al., 2014;

Dlugokencky et al., 2018) and to $0.7\pm0.6$ ppb $yr^{-1}$ during 1999-2006 (Dlugokencky et al., 2018). After 2006, renewed methane growth started with a growth rate of $5.7\pm1.2$ ppb $yr^{-1}$ in 2007-2013 and reached $12.6 \pm 0.5$ ppb $yr^{-1}$ in 2014 and $10.0 \pm 0.7$ ppb $yr^{-1}$ in 2015 (Nisbet et al., 2016; Dlugokencky et al., 2018). Investigations of the drivers of the observed methane trend and interannual variability have mainly focused on the changes in the global methane budget. While anthropogenic activities are widely considered responsible for the long-term methane increase since pre-industrial times (Dlugokencky et

al., 2011), there is no consensus on the drivers for the methane stabilization during 1999-2006 and renewed growth since 2007. Previous studies have attributed the stabilization during 1999-2006 to the combined effects of alternatively increased anthropogenic emissions with decreased wetland emissions (Bousquet et al., 2006), decreased fossil fuel emissions (Dlugokencky et al., 2003; Simpson et al., 2012; Schaefer et al., 2016) or rice paddies emissions (Kai et al., 2011), stable emissions from microbial and fossil fuel sources (Levin et al., 2012), or variations of methane sinks (Rigby et al., 2008;

Montzka et al., 2011; Schaefer et al., 2016). The observed renewed growth since 2007 has been explained alternatively through increases in wetland emissions (Dlugokencky et al., 2009; Bousquet et al., 2011; Nisbet et al., 2016), increases in



agricultural emissions (Schaefer et al., 2016), increases in fossil fuel emissions (Rice et al., 2016; Worden et al., 2017), and decreases in sources compensated by decreases in sinks due to OH levels (Turner et al., 2017; Rigby et al, 2017). These different explanations reflect limitations in our understanding of recent changes in methane and its budget.

Previous work has generally combined observations of methane and its $\delta^{13}$C with isotopic source signatures weighted by their emissions with inverse models (top-down), process-based models (bottom-up), or box models to estimate methane emissions and sinks and their variability (Bousquet et al., 2006; Monteil et al., 2011; Rigby et al., 2012; Bloom et al., 2012; Kirschke et al., 2013; Ghosh et al., 2015; Schwietzke et al., 2016; Schaefer et al., 2016; Nisbet et al., 2014, 2016; Dalsøren et al., 2016; Turner et al., 2017; Rigby et al., 2017). Inverse models use observations to derive emissions, but usually prescribe

climatological OH, O($^1$D), and Cl levels or loss rates (e.g., Rice et al., 2016; Tsuruta et al., 2017). Box models, on the other hand, use methane observations together with those of other proxy chemicals (e.g., $^{13}$C/$^{12}$C ratio, ethane, carbon monoxide, methyl chloroform) to provide information on the global methane budget (e.g., Schaefer et al., 2016; Turner et al., 2017), but lack information on spatial variability or regional characteristics. With process-based models (e.g., wetlands) and inventories representing different source types (e.g., fossil fuel emissions) to drive chemistry transport models, the bottom-up approach

is able to estimate the methane budget for all individual sources and sinks. However, without observational constraints, there is considerable uncertainty in the total methane emissions derived from a combination of independent bottom-up estimates (Saunois et al., 2016).

Bottom-up global Earth System Models (ESMs) that realistically simulate the physical, chemical, and biogeochemical processes, as well as interactions and feedbacks among these processes, are useful tools to characterize the global methane

cycle and quantify the global methane budget and impacts on composition and climate. Dalsøren et al. (2016) investigated the evolution of atmospheric methane by driving a chemistry transport model with bottom-up emissions. While their model results are able to match the observed time evolution of methane without emission adjustments, surface methane is largely underpredicted in their study. Ghosh et al. (2015) optimized bottom-up emission data to investigate methane trends; however, OH trends and interannual variability were not considered in their chemistry transport model. Here, we apply a

prototype of the full-chemistry version of the Geophysical Fluid Dynamics Laboratory (GFDL) new-generation Atmospheric Model, version 4.1 (AM4.1, Zhao et al., 2018a, b; Horowitz et al., manuscript in preparation) to investigate the evolution of methane over 1980-2014. Our main objectives are to improve the representation of methane in GFDL-AM4, to comprehensively evaluate the model performance of methane predictions with an improved representation of methane budget, and to investigate possible drivers of the methane trends and variability. This paper is structured as follows: Section

2 describes the modeling approach, emission inventories, and observations used for model evaluation. Results of the model evaluation, global methane budget analysis, and model sensitivities are presented in Section 3. Finally, Section 4 summarizes the results and discusses the implication of these results.



## 2 Methodology and data

### 2.1 Model description and initialization

We use a prototype version of the new generation NOAA Geophysical Fluid Dynamics Laboratory chemistry-climate model, GFDL-AM4.1 (Zhao et al., 2018a, b; Horowitz et al., manuscript in preparation). A detailed description of the physics and dynamics in AM4.1 is provided by Zhao et al. (2018a, b). The version of AM4.1 with full interactive chemistry used in this work is described by Schnell et al. (2018). In its standard form, this model setup consists of a cubed sphere finite-volume dynamical core with a horizontal resolution of ~100 km with 49 vertical levels extending from the surface up to ~80 km. The

model's lowermost level is approximately 30 m thick. The chemistry and aerosol physics in this model have been updated from the previous version (GFDL-AM3; Naik et al., 2013), as described by Mao et al. (2013a, b) and Paulot et al. (2016). There are a total of 102 advected gas tracers and 18 aerosol tracers, 44 photolysis reactions, and 205 kinetic reactions included in the chemical mechanism in this version of AM4.1 to represent tropospheric and stratospheric chemistry.

The standard AM4.1 configuration uses global annual-mean methane concentrations as a lower boundary condition to

simulate the atmospheric distribution of methane. Although the model simulates reasonable global-mean methane abundances, large biases exist in the simulated latitudinal distribution and seasonal cycle. This modeling framework also does not allow for the full characterization of the drivers of methane trends and variability. To overcome this issue, we updated AM4.1 to be driven by methane emissions. Table 1 provides information on the emission datasets used in this work. Surface emissions from anthropogenic sources - including agriculture (AGR), energy production (ENE), industry (IND),

road transportation (TRA), residential, commercial, and other sectors (RCO), waste (WST), and international shipping (SHP)- are from the Community Emissions Data System (CEDS, version 2017-05-18, Hoesly et al., 2018) developed in support of the Coupled Model Intercomparison Project Phase 6 (CMIP6) for 1980-2014. Emissions for 2015-2017 are from a middle-of-the-road scenario of Shared Socioeconomic Pathways targeting a forcing level of 4.5 W m$^{-2}$ (SSP2-4.5), developed in support of the ScenarioMIP experiment within CMIP6 (Gidden et al., 2018). Biomass burning (BMB)

emissions are from van Marle et al. (2017) for 1980-2014 and from SSP2-4.5 for 2015-2017, and are vertically distributed over seven ecosystem-dependent altitude levels between the surface and 6 km above the surface, following the methodology of Dentener et al. (2006). Anthropogenic and biomass burning emissions are represented by monthly gridded emissions including seasonal and interannual variability. Natural emissions include wetland (WET) emissions from the WetCHARTs version 1.0 inventory (Bloom et al., 2017), ocean (OCN) emissions from Brasseur et al. (1998) with near-shore methane

fluxes from Lambert and Schmidt (1993) and Patra et al. (2011), termites (TMI) from Fung et al. (1991), and mud volcanoes (VOL) from Etiope and Milkov (2004) and Patra et al. (2011). Wetland emissions and ocean emissions are climatological monthly means without interannual variability. The remaining natural emissions are based on a climatological annual mean (repeated every month without seasonal variability). Trends in the total emissions and emissions from major sectors over 1980 to 2017 are shown in Figure 1. Trends in total emissions are primarily driven by trends in ENE, AGR, and WST

emissions, while BMB emissions contribute to interannual variability. Anthropogenic and biomass burning emissions of





other short-lived species also follow CEDS and SSP2-4.5 inventories. Natural emissions of other short-lived species are from Naik et al. (2013). Biogenic isoprene emissions are calculated interactively following Guenther et al. (2006).

The methane sinks considered in AM4.1 include oxidation by OH radicals, Cl, and $O(^1D)$, and dry deposition. Since the model does not represent tropospheric halogen chemistry, it does not consider removal of methane by Cl in the troposphere, which has been shown to be extremely minor (Gromov et al., 2018). The dry deposition flux of methane is calculated based on a monthly climatology of deposition velocities (Horowitz et al, 2003) calculated by a resistance-in-series scheme (Wesely, 1989; Hess et al., 2000) and used to mimic methane loss by soil uptake, which accounts for about 5% of the total methane sink (Kirschke et al., 2013; Saunois et al., 2016).

In this work, we included 12 additional methane tracers tagged by source sector to attribute methane from agriculture (CH4AGR), energy (CH4ENE), industry (CH4IND), transportation (CH4TRA), residents (CH4RCO), waste (CH4WST), shipping (CH4SHP), biomass burning (CH4BMB), ocean (CH4OCN), wetland (CH4WET), termites (CH4TMI), and mud volcanoes (CH4VOL). The tracers are emitted from corresponding sources, and undergo the same chemical pathways and dynamics as the full $CH_4$ tracer. For analysis, we combine CH4IND, CH4TRA, CH4RCO, and CH4SHP as other anthropogenic tracers (i.e., CH4OAT), and combine CH4OCN, CH4TMI, and CH4VOL as other natural tracers (i.e., CH4ONA).

Initially the model was spun up in a 50-year run with repetitive 1979 emissions until stable atmospheric burdens of methane and tagged tracers were obtained. After the spin-up, several sets of simulations were conducted for 1980-2017 to quantify the methane budget and investigate the impacts of changes in methane sources and sinks (see Section 2.2). All model simulations are forced with interannually-varying sea surface temperatures and sea ice from Taylor et al. (2000), prepared in support of the CMIP6 Atmospheric Model Intercomparison Project (AMIP) simulations. Horizontal winds are nudged to the National Centers for Environmental Prediction (NCEP) reanalysis (Kalnay et al., 1996) using a pressure-dependent nudging technique (Lin et al., 2012).

## 2.2 Simulation design

We conduct several sets of hindcast simulations for 1980-2017, as listed in Table 2, to quantify the methane budget and investigate the contributions of sources and sinks to the trend and variability of methane. The model simulation using the initial methane emissions inventory ($E_{init}$) described in Section 2.1 was found to largely underestimate the methane burden. Assuming that this mismatch is due to a bias in the simulated methane budget, we can either increase methane sources or decrease methane sinks to match the observations. We perform several optimization simulations that explore the sensitivity of methane to uncertainties in emissions of methane and levels of OH, the dominant sink for methane. Because OH trends and variability depend on a number of factors, including temperature, water vapor, $O_3$, and emissions of nitrogen oxide ($NO_x$), carbon monoxide (CO), and volatile organic compounds (VOCs), it is not straightforward to perturb OH. Previous work has shown that interannual variability of global OH is highly correlated with $NO_x$ from lightning (Fiore et al., 2006; Murray et al., 2013). Therefore, we apply a scaling factor to lightning $NO_x$ ($LNO_x$) emissions to indirectly adjust OH levels





without influencing its variability. The $LNO_x$ emissions are calculated interactively based on Horowitz et al. (2003) as a
function of subgrid convection parameterized in the model. The climatological global mean $LNO_x$ emission simulated by
AM4.1 is about 3.6 TgN yr$^{-1}$, within the range of 2-8 TgN yr$^{-1}$ estimated by previous studies (e.g., Schumann and Huntrieser,
2007).

Here, we test the sensitivity of simulated methane to three assumptions: 1) standard OH levels simulated by AM4.1 (referred
as "S0"); 2) low OH levels via applying a scaling factor of 0.5 to the default $LNO_x$ emission calculations (referred as "S1");
3) high OH levels via applying a factor of 2 to the default $LNO_x$ emission calculation (referred as "S2"). For each OH option,
we begin with initial methane emissions and then optimize global total emissions as described below to match simulated
methane with surface observations. Different OH levels could lead to different estimations of the optimized total emissions,
which provides a measure of uncertainties in our optimized total methane emissions.

We apply a simple mass balance approach to optimize global total methane emissions, following the methodology of Ghosh
et al. (2015). We calculate an increment $\Delta E$ by which global emissions need to be modified for each year. Unlike inverse
modeling studies such as (Houweling et al., 2017), we do not optimize emissions for each grid cell. Instead, we uniformly
scale emissions for particular sectors (as described below) globally for each year by the rate of the optimized emission total
($E_{opt} = E_{init} + \Delta E$) to the initial emissions ($E_{init}$). We assume that the spatial distribution of methane emissions from the initial
emission inventories are the best available information we have. Considering the large uncertainties in the anthropogenic and
wetland emissions, we perform two simulations in which we achieve the optimized emission totals by scaling either
anthropogenic sources, including biomass burning sector only (referred to as "Aopt") or the wetland sector only (referred to
as "Wopt") for the standard (S0) $LNO_x$ scenario. The purpose of conducting these simulations is to investigate the impact of
optimizing emissions from different sectors on methane predictions. For the Aopt case, eight anthropogenic sectors (i.e.,
AGR, ENE, IND, TRA, RCO, WST, SHP, and BMB) are uniformly scaled by the ratio of $\Delta E$ to total anthropogenic
emissions, keeping the fractions of individual sources unchanged. For the Wopt case, wetland emissions are rescaled to
increase this source by $\Delta E$. For S1 and S2 scenarios, we apply $\Delta E$ to wetland sector only. The total $E_{opt}$ emissions are the
same for both Aopt and Wopt cases. Time series of methane optimized total emissions and emissions from major sectors
from S0Aopt and S0Wopt over the 1980 to 2017 period are shown in Figure S2 in the Supplement.

### 2.3 Observations

We evaluate the simulated methane dry-air mole fraction (DMF) against a suite of ground-based and aircraft observations
and satellite retrievals of column-averaged $CH_4$ to thoroughly evaluate the model simulated spatial and temporal distribution
of methane. To evaluate surface $CH_4$, we use measurements from a globally distributed network of air sampling sites
maintained by the Global Monitoring Division (GMD) of the Earth System Research Laboratory at the National Oceanic and
Atmospheric Administration (NOAA) (Dlugokencky et al., 2018). The global estimates derived from surface measurements
are based on a number of sites at remote marine sea level locations with well-mixed marine boundary layer (MBL) to
represent background methane. The locations of MBL sites are shown in Figure S1 and the information for each MBL site is

listed in Table S1 in the Supplement. A function fit consisting of yearly harmonics and a polynomial trend, with fast fourier transform and low pass filtering of the residuals are applied to the monthly mean methane DMF to approximate the long-term trend and average seasonal cycle at each MBL site (Thoning et al., 1989; Thoning, 2019). A meridional curve (Tans et al., 1989) was fitted through these site values to get the latitudinal distribution of methane. The same sampling and processing approach (Thoning et al., 1989; Tans et al., 1989) is applied to the simulated monthly mean methane DMF to calculate global and zonal averages to facilitate consistent model-observation comparison. Besides the comparison with global estimates from MBL sites, we also evaluate the model performance at various GMD sites to investigate the contributions from local sources. For site specific evaluation, we sample the model grid cell at the location of the corresponding site and at the model layer with height closest to the altitude of the corresponding site.

Due to the sparseness of the ground-based observational sites, especially over continental regions, we also evaluate simulated methane against satellite retrievals to reveal information on regional characteristics. Total column-averaged methane DMFs are evaluated against satellite retrievals from the Scanning Imaging Absorption Spectrometer for Atmospheric Chartography (SCIAMACHY) instrument on board the European Space Agency's environmental research satellite ENVISAT (Frankenberg et al., 2011) for January 2003 to April 2012 and the Thermal And Near Infrared Sensor for carbon Observations – Fourier Transform Spectrometer (TANSO-FTS) instrument onboard the Japanese Greenhouse gases Observing SATellite (GOSAT) (Kuze et al., 2016) for April 2009 to December 2016. We compare monthly mean satellite retrievals with simulated monthly mean methane. Retrieval-specific averaging kernels are also applied to simulated monthly mean methane to calculate simulated column-averaged methane DMF.

To investigate background tropospheric methane variability, we compare the simulated vertical profiles with aircraft measurements from the High-performance Instrumented Airborne Platform for Environmental Research (HIAPER) Pole-to-Pole observation (HIPPO) campaigns from January 2009 to September 2011 (Wofsy et al., 2012). A total of 787 profiles were flown during the 5 campaigns with continuous profiling between approximately 150 m and 8500 m altitudes, but also including many profiles up to 14 km altitude. For each HIPPO mission, we spatially sample the model consistent with the observations and average the model for the months of the campaign to create climatological monthly means.

## 3 Results and discussions

### 3.1 Observations

The detailed model evaluation for S0Aopt and S0Wopt are discussed below. We first evaluate the mean climatological spatial distribution and seasonal variability simulated by the model and then evaluate the trends and variability.

### 3.1.1 Climatological evaluation

Figure 2 shows the model bias and correlation coefficient of simulated climatological mean surface methane DMF against NOAA GMD surface observations (Dlugokencky et al., 2018) for the 1983-2017. The mean seasonal cycle at individual



GMD sites is shown in Figure S3 in the Supplement. GMD sites with at least 20 years of observational records are selected for model climatological evaluation. The information of these sites is shown in Table S2 in the Supplement. As shown in

Figure 2a, simulations with optimization of either anthropogenic (S0Aopt) or wetland (S0Wopt) emissions are generally able to reproduce surface methane DMF with model biases within ±30 ppb at most sites. Both S0Wopt and S0Aopt simulate methane DMF relatively well over the Southern Hemisphere. Going from south to north, the low bias in methane DMF decreases and becomes a high bias over the tropics. Simulated methane in both S0Aopt and S0Wopt are moderately high biased over the tropical Pacific Ocean (by up to ~ 40 ppb), indicating possible overestimation of methane emissions over the

tropics and possible underestimations in OH levels. Large positive biases occur at Key Biscayne (25.7 N, 80.2 W) for both S0Wopt and S0Aopt, likely due to a model sampling bias, with model grid box overlapping land while samples are collected with onshore winds. Over middle and high latitudes of Northern Hemisphere, the simulated surface methane DMF shows low and high biases at individual sites, possibly due in part to uncertainties in the local emissions. As shown in Figure 2b, both S0Aopt and S0Wopt are able to capture the methane seasonal cycle at most sites (with a correlation coefficient (R)

larger than 0.5 for about 80% of sites). Both S0Aopt and S0Wopt are able to reproduce the methane seasonal cycle over the Southern Hemisphere. However, both S0Aopt and S0Wopt show poor performance in the seasonal cycle over the Southern tropical Pacific Ocean, with R < 0.5 (e.g., POCS10 and POCS15 in Figure S3 in the Supplement), but show good performance in the seasonal cycle over the Northern tropical Pacific Ocean, with R = 0.9 (e.g., POCN05, POCN10, and POCN15 in Figure S3 in the Supplement). Poor performance also exists at a few sites in middle and high northern latitudes

(e.g., AZR, UUM, LEF, MHD, and ICE shown in Figure S3 in the supplement), mainly due to overestimates of methane during summer. Uncertainties in the seasonality of methane emissions, OH abundances, and long-range transport could lead to biases in the seasonal cycle. In general, both S0Aopt and S0Wopt are able to capture the methane latitudinal gradient (e.g., R = 0.9). This suggests that the spatial distribution of methane in emissions is reasonable on the large scale despite uncertainties in representing local sources.

To investigate background tropospheric methane variability, Figure 3 shows the bias in the simulated vertical distribution of methane with respect to HIPPO observations for the S0Aopt and S0Wopt simulations. S0Aopt and S0Wopt simulations produce very similar methane profiles. Both S0Aopt and S0Wopt match observed methane profiles very well over Southern Hemisphere. Compared to HIPPO measurements, methane in both simulations is consistently high over the tropical Pacific Ocean (by up to ~ 50 ppb) from the surface to 700 mb during all HIPPO campaigns. These biases decrease with altitude and

decrease with latitude except for summer. In the Northern Hemisphere, both S0Wopt and S0Aopt simulations capture the observed methane from near the surface to 700 mb, but are generally biased low, except in summer when they are biased high, especially at mid-latitudes. Mid-latitude background methane is affected by both high-latitude and low-latitude air masses on synoptic scales. Biases over these regions could result from many processes (e.g., overestimation of the summer emissions, insufficient OH levels, and model transport). In general, the relative differences between the simulated methane

profiles and HIPPO measurements are within 2% over most regions, demonstrating the capability of the improved GFDL-AM4.1 for simulating tropospheric methane.



### 3.1.2 Trend evaluation

As described in Section 2.3, we applied a function fit consisting of yearly harmonics and a polynomial trend, with fast fourier transform and low pass filtering of the residuals to the monthly mean methane DMF (Thoning et al., 1989; Thoning, 260 2019) to estimate the long-term trend and growth rates discussed below. The comparisons of simulated global mean background surface methane trends and growth rates to NOAA-GMD observations are shown in Figures 4 and 5. Both S0Wopt and S0Aopt predict similar global mean surface methane DMF, trend, and growth rates, since the global methane budget (emissions and sinks) is the same in the two simulations. S0Wopt and S0Aopt are also able to reproduce the global annual mean surface methane DMF (with root-mean-square-error (RMSE) = 8.3 ppb in S0Wopt and 8.9 ppb in S0Aopt) over 265 1983-2017 and capture the methane trend very well (with R = 1.0 in both S0Wopt and S0Aopt), especially over the Southern Hemisphere. In general, the RMSE for S0Wopt is lower than that for S0Aopt, except over the southern tropics. The major discrepancies in the surface methane DMF between model simulations and observations are mainly over the low northern latitudes (0-30º N), especially the tropics (Figure 4d), where the RMSE is greater than 20 ppb. Over the high northern latitudes, S0Aopt overestimates background methane DMF by about 20-30 ppb during 1984-1998, whereas S0Wopt 270 performs better over this period. After 1998, S0Aopt reproduces the maximum methane DMF very well, while S0Wopt slightly underestimates methane DMF by up to 10 ppb. The agreement between the simulated and observed global methane trends increases our confidence in the optimized methane emission trends used in this work.

As shown in Figure 5, both simulations are in general able to reproduce the observed global methane growth rate (with R = 0.8 in both S0Wopt and S0Aopt), despite a slight mismatch (~1-year) during 1997-2007. Global methane simulated by 275 S0Aopt shows rapid growth during 1984-1991 (13.5±2.1 ppb yr$^{-1}$, annual mean ± standard deviation), slower growth during 1992-1998 (4.8±2.6 ppb yr$^{-1}$), a relative stabilization during 1999-2006 (1.2±4.7 ppb yr$^{-1}$), and renewed growth during 2007-2017 (6.1±2.8 ppb yr$^{-1}$). The simulated global methane growth rates by S0Wopt show similar trends (13.5±2.1 ppb yr$^{-1}$ during 1984-1991, 4.7±3.3 ppb yr$^{-1}$ during 1992-1998, 1.3±4.8 ppb yr$^{-1}$ during 1999-2006, and 6.1±2.8 ppb yr$^{-1}$ during 2007-2017). The simulated growth rates during 1984-1991 are slightly higher than the NOAA-GMD estimates (11.6±1.3 ppb yr$^{-1}$), 280 while the simulated growth rates during 1992-1998, 1999-2006, and 2007-2017 are within the ranges of NOAA-GMD estimates (5.6±3.5 ppb yr$^{-1}$, 0.7±3.1 ppb yr$^{-1}$, 6.9±2.6 ppb yr$^{-1}$, respectively). Over the tropics, both S0Aopt and S0Wopt overestimate methane growth rates (by about 5-10 ppb yr$^{-1}$) during 1984-1990, but are able to reproduce methane growth rates relatively well afterwards. Agreement of the methane growth rate is worse in the Northern Hemisphere than in the Southern Hemisphere, especially at the high northern latitudes, where R is smaller than 0.5. Over 30-90ºN, neither S0Aopt 285 nor S0Wopt is able to reproduce methane growth rates during 1984-1989 and there is a slight mismatch (~1-2 years) in methane growth rates afterwards. These biases indicate larger uncertainties in the methane emissions at high Northern Hemisphere than in other regions.

Comparisons of simulated surface methane DMF to NOAA-GMD observations at individual sites are shown in Figure S4 in the supplement. S0Aopt and S0Wopt simulated very similar methane DMF. Both simulations tend to be biased low over



Southern Hemisphere sites, but the low bias decreases northward. The simulations are moderately high biased (with RMSEs
up to ~ 40 ppb) over tropical regions (e.g., POCS15, POCS10, SMO, POCS05, POCN00, CHR, and POCN05). These sites
are mainly remote sites and surface methane DMF represents background methane levels. However, the model predicts
surface methane DMF relatively well at Ascension Island (ASC, 8°S, 14.4°W, 85 m), which is also a remote site. The ASC is
at higher altitude than other remote sites. This indicates that the model tends to overestimate background methane DMF at

the surface, but is able to capture background methane at higher levels. Moderate overestimates also occur at Mahe Island
(SEY, 4.7°S, 55.5°E), a location that could be affected by airmasses from polluted areas over the tropics and Northern
Hemisphere. Over middle and high latitude Northern Hemisphere, both S0Aopt and S0Wopt simulate surface methane DMF
relatively well at most sites, except at Key Biscayne (KEY, 25.7°N, 80.2°W), Tae-ahn Peninsula (TAP, 36.7°N, 126.1°W),
Park Falls (LEF, 45.9°N, 113.7°W), and Mace Head (MHD, 53.3°N, 9.9°W). KEY and MHD are remote sites and sampled

under onshore winds, whereas TAP and LEF are affected by local sources and model transport. The high biases at these sites
could be due in part to model sampling bias and uncertainties in local emissions. On the other hand, both S0Wopt and
S0Aopt are able to capture monthly variations in methane at most of the sites except at LEF, where R = 0.4 for S0Wopt and
0.5 for S0Aopt, respectively. In general, both S0Wopt and S0Aopt are able to reproduce the surface methane DMF and
capture the trend at most sites (e.g., with R greater than 0.5 at 98% of total sites and with RMSE less than 30 ppb at 74% of

total sites).

Unlike the evaluation of global mean surface methane DMF, which is based on observations from a number of sites with
well-mixed MBLs, the evaluation of global mean column-averaged methane DMF against satellite retrievals mainly covers
continents, considering the impacts from polluted areas and the contributions from the troposphere and the stratosphere.
Simulated monthly mean column-averaged methane DMF are compared with satellite retrievals (e.g., SCIAMACHY and

GOSAT) in Figure 6. The averaging kernels of SCIAMACHY and GOSAT are individually applied to the model to calculate
column-averaged methane abundances. Both simulations are able to capture the monthly variation of methane with R greater
than 0.9, but underestimate column-averaged methane, with RMSE of about 21 ppb and 29 ppb when compared to
SCIAMACHY and GOSAT retrievals, respectively. The biases increase poleward in both SCIAMACHY and GOSAT
comparisons; large uncertainty exists in the satellite retrievals over high latitudes (e.g., > 70°) due to large solar zenith angles

and potential high cloud cover. The underestimates are mainly due to biases in the middle/upper troposphere and
stratosphere (as shown in Figure 3). The differences in the column-averaged methane abundances between satellite retrievals
and model simulations are mostly within 2% except in polar regions where there are large uncertainties in the satellite
retrievals. Both simulations are also able to capture the latitudinal distribution of the column-averaged methane DMF with R
close to 1.

**3.2 Global methane budget**

Figure 7 shows time series of optimized total emissions, global sink, and global burden of methane based on S0Wopt. Since
global totals in S0Aopt and S0Wopt simulations are very close to each other, we only show the budget for S0Wopt. As





depicted in Figure 7, the simulated global methane burden steadily increases from 1980 to 1992, with a growth rate of 39 Tg yr$^{-1}$. During 1993-1998, the global methane burden growth slows down with a growth rate of 16 Tg yr$^{-1}$. The growth rates

simulated by the model agrees well with the observed growth rates during 1984-1997 as shown in Figure 5a. The simulated growth rate in global methane burden decreases to 4 Tg yr$^{-1}$ during 1999-2006 while it increases to 16 Tg yr$^{-1}$ during 2007-2017 and reaches over 20 Tg yr$^{-1}$ during 2014-2016. The changes in the global burdens are due to the imbalance between methane sources and sinks. As shown in Figure 7, the optimized emissions in general increase during 1980-2017, with an annual mean of 576±32 Tg yr$^{-1}$ (mean±standard deviation) and show much larger interannual variability during 1991-1993

and 1997-2000. Although there is an overall increasing trend in total global emissions, growth in annual mean emissions has increased from the 1980s (with an annual emission growth rate of 3.9 Tg yr$^{-1}$) to the 1990s (4.4 Tg yr$^{-1}$), but decreased to 0.3 Tg yr$^{-1}$ during 2000-2006, and increased again to 2.3 Tg yr$^{-1}$ during 2007-2017. The estimations of optimized emissions are based on the comparisons of simulated surface methane with surface observations. Uncertainties in the interannual variability of simulated OH levels and therefore methane sinks lead to uncertainties in the interannual variability of the

optimized emissions. The larger interannual variabilities during 1991-1993 and 1997-2000 are likely due to the strong El Niño events during 1991-1992 and 1997-1998.

Unlike methane emissions, the methane sink increases during 1980-2007, with relative stabilization during 2008-2014 but a resumed increase during 2015-2017. The annual mean methane sink during 1980-2017 is 560±44 Tg yr$^{-1}$ (mean±standard deviation). The trends in methane sink are affected by the changes in both methane and OH levels (assuming that other sinks are minor).

Figure 8 shows the tropospheric OH anomalies with respect to 1998-2007. An interesting finding is that AM4.1 predicts higher OH levels during 2008-2014 than 1998-2007 by 3.1%, whereas recent studies applying multispecies inversion with a box-model framework (e.g., Rigby et al., 2017; Turner et al., 2017) suggest a decline in OH levels after 2007. However, a recent study by Naus et al. (2018) found a shift to positive OH trend over 1994-2015 after applying bias corrections based on a 3-D CTM to a similar box model setup. In addition, OH levels simulated by AM4.1 decrease from

2013 to 2015 but increase again afterwards, leading to an increase in methane sinks during 2015-2017. As shown in Figure 7, higher methane sources than sinks during 1980-1998 lead to an increase in methane burden. A relative balance between methane sources and sinks during 1999-2006 leads to the methane stabilization. Compared to 1999-2006, both methane sources and sinks are higher during 2007-2017, but methane sources outweigh sinks after 2007 leading to renewed methane growth.

Table 3 provides a summary of decadal mean methane budget for 1980-2017. Compared to Kirschke et al. (2013) and Saunois et al. (2016), the total natural sources from the initial emission inventories (203 Tg yr$^{-1}$) are at the lower range of top-down estimates during this period, except for the 1990s, when they are slightly higher than top-down estimates but still much lower than the bottom-up estimates. This mainly results from wetland emissions in the initial emission inventories that are slightly higher than top-down estimates but much lower than bottom-up estimates during 1990s. The total anthropogenic

sources from the initial emission inventories are overall within the range of top-down or bottom-up estimates, except for 1980-1989, when they are lower than the estimates in Kirschke et al. (2013) and Saunois et al. (2016). The low values in the



1980s result mainly from low estimated sources from agriculture and waste sectors in the CEDS inventory. With the optimized global total emissions, the total sources used in this work and the total sinks estimated by AM4.1 are either in the range of top-down or bottom-up estimates by previous studies. As a result, the imbalance between total sources and total

sinks estimated in this work are overall within the range of estimates by previous studies although we find a smaller imbalance than previous estimates for the 2000s and afterwards. The atmospheric growth rates simulated by the model (sampled identically as for observations) are also comparable to the observed atmospheric growth rates.

### 3.3 Sensitivity to sector optimization

#### 3.3.1 Spatial distribution

As described in Section 2.2, the emission optimization is conducted for anthropogenic sectors (i.e., S0Aopt) and wetland sector (i.e., S0Wopt). Although global total methane emissions are the same for S0Aopt and S0Wopt, they have different allocations for anthropogenic and wetland sectors and different spatial distributions as well. Here we analyze the sensitivity of sector optimization on the spatial distribution of simulated methane concentrations. Figures 9 and 10 show the spatial distributions of the differences in the methane emissions and surface methane abundance between S0Aopt and S0Wopt

during the four periods (i.e., 1980-1989, 1990-1999, 2000-2006, and 2007-2017). Surface methane is always lower in Aopt than Wopt in the tropics (e.g., 15° S-10° N) during the four periods. This is mainly due to much lower wetland emissions in S0Aopt than in S0Wopt (Figure 10), which dominates total emissions over these regions (e.g., tropical South America and Central Africa). There is not much difference in surface methane over low and high southern latitudes (e.g., 15-90° S) between the two simulations. This agreement is mainly because larger anthropogenic emissions in S0Aopt compensate

smaller wetland emissions, producing only small differences in the total emissions, within 0.1 Tg yr$^{-1}$ (Figure 10). Unlike the Southern Hemisphere, surface methane concentrations are in general higher in S0Aopt than S0Wopt in the Northern Hemisphere, especially over the Eastern U.S. and Eurasia, due to much higher anthropogenic emissions in S0Aopt. The lower surface methane values in S0Aopt over northern Canada are due to much lower wetland emissions in S0Aopt.

Figure 11 shows the methane growth rates simulated by Aopt and Wopt during the four time periods. Global mean methane

growth rates simulated by Aopt and Wopt are very consistent during the four periods, with growth rates decreasing from 1980s to 1990s, stabilizing during 2000-2006, and increasing after 2007. During the 1980s and 1990s, methane growth rates in both S0Aopt and S0Wopt increase over most of the globe except a decrease over Russia, due to significant decreases in anthropogenic emissions (mainly from the energy sector) in the former Soviet Union, consistent with previous studies (Dlugokencky et al., 2011). During 2000-2006, methane growth rates increase significantly over East Asia in both S0Aopt

and S0Wopt while they decrease over tropical South America and Central Africa in S0Wopt but not in Aopt. This is mainly due to decreases in wetland emissions in the S0Wopt case, while wetland emissions are constant for each year in Aopt case. After 2007, both Aopt and Wopt suggest large increases in methane growth rates over East Asia (mainly due to increases in anthropogenic emissions) by up to ~38 ppb yr$^{-1}$ with smaller increases elsewhere (< 7 ppb yr$^{-1}$), with noticeable increases



also over the Arctic ( > 7 ppb yr$^{-1}$). The relatively large methane growth over the Arctic is mainly due to increases in
anthropogenic methane from lower latitudes.

As discussed in Sections 3.1 and 3.2, the similarity in S0Aopt and S0Wopt simulation results suggests that for 3-dimensional chemistry transport models, reasonable estimates of total global methane emissions are critical for global methane predictions, despite the uncertainties in the spatial distribution of the emissions and in the estimates of individual sources, which are more important for regional methane predictions. At the same time, accurate estimates of individual sources are necessary to attribute the methane trend and variability into individual sources.

### 3.3.2 Source tagged tracers

In this section, we apply Mann-Kendall (M-K) test to estimate the linear trend (different from long-term trend discussed in Section 3.1.2) of global mean source tagged tracers and total methane for 1983-1998, 1999-2006, and 2007-2017 to investigate possible drivers in total methane trends. Figure 12 compares the trends of source tagged tracers and total methane from S0Aopt and S0Wopt during 1983-1998, 1999-2006, and 2007-2017. As shown in Figure 12, both S0Aopt and S0Wopt are in general able to capture the methane trends during different time periods. For S0Aopt, globally, total methane shows an increasing trend of 10.5 ppb yr$^{-1}$ during 1983-1998, slightly greater than observations (8.8 ppb yr$^{-1}$), but correlated very well with the observations (R = 1.0). The tagged anthropogenic tracers all show increasing trends during 1983-1998 despite the increases in OH levels, with larger increasing trends by AGR (3.6 ppb yr$^{-1}$) and WST (3.6 ppb yr$^{-1}$) consistent with emission trends. Major anthropogenic tracers (e.g., CH4AGR, CH4ENE, and CH4WST) correlate very well with total methane, with R varying between 0.9 to 1.0 over this time period. Since wetland emissions and other natural emissions are constant every year in S0Aopt, with the increases in OH levels during 1983-1998, both CH4WET and CH4ONA decrease by -0.5 ppb yr$^{-1}$ and -0.1 ppb yr$^{-1}$, respectively, over this period. During 1999-2006, total methane has a small increasing trend of 1.3 ppb yr$^{-1}$, still slightly greater than observations (0.6 ppb yr$^{-1}$), but correlated relatively well with the observations (R = 0.8). During this time period, there are increasing trends in CH4ENE (2.6 ppb yr$^{-1}$) and CH4WST (2.3 ppb yr$^{-1}$) with slightly decreasing trends in CH4AGR (-0.1 ppb yr$^{-1}$), CH4BMB (-0.9 ppb yr$^{-1}$) and CH4OAT (-0.5 ppb yr$^{-1}$). Anthropogenic tracers such as CH4ENE and CH4WST correlate well with total methane, whereas CH4AGR shows a poor correlation with total methane, and CH4BMB and CH4OAT show an anticorrelation with total methane over this time period. Similarly, with the increases in OH levels during 1999-2006, both CH4WET and CH4ONA decrease, with a linear decreasing trend of -1.8 ppb yr$^{-1}$ and -0.4 ppb yr$^{-1}$. During 2007-2017, total methane shows a renewed increasing trend of 5.3 ppb yr$^{-1}$, slightly below observations (6.0 ppb yr$^{-1}$) but correlated relatively well with the observations (R = 1.0). During this time, CH4ENE shows a large increasing trend (5.8 ppb yr$^{-1}$), dominating the total methane trend. Interestingly, although there is a slight decrease in OH levels after 2008, with both CH4WET and CH4ONA still show decreasing trends of -1.1 ppb yr$^{-1}$ and -0.3 ppb yr$^{-1}$ during 2007-2017. Also, all the natural tracers show an anticorrelation with total methane during this period. The results from S0Aopt suggest that globally, anthropogenic tracers dominate total methane trends during the entire simulation period. During the 1980s and 1990s, emissions from agriculture, energy, and waste sectors are the major contributors to the methane





increase. During 1999-2006, where methane stabilizes, increases in methane sinks and methane sources alternatively dominate the trend for different tracers and therefore the imbalance between methane sinks and sources dominate the total methane trend. During 2007-2017, the energy sector is the major contributor to the methane renewed growth.

The source tagged tracers behave slightly different in S0Wopt. For S0Wopt, globally, total methane shows a similar increasing trend as S0Aopt (as discussed in section 3.1.2). The tagged anthropogenic tracers all show increasing trends during 1983-1998 except CH4ENE (-0.3 ppb yr$^{-1}$). Anthropogenic tracers (except CH4OAT) in general correlate well with total methane. CH4WET show a significant increasing trend during this period (7.0 ppb yr$^{-1}$) and correlate relatively well with total methane. During 1999-2006, anthropogenic tracers such as CH4ENE and CH4WST show increasing trends (1.8

ppb yr$^{-1}$ and 1.9 ppb yr$^{-1}$, respectively) and correlate relatively well with total methane (R = 0.7 and 0.8, respectively), whereas all other tracers show decreasing trends and are anticorrelated with total methane. During this time, our wetland tracer (CH4WET) shows a slightly decreasing trend (-0.6 ppb yr$^{-1}$), mainly due to the slightly higher CH4WET sinks (226 Tg yr$^{-1}$) than sources (223 Tg yr$^{-1}$). During 2007-2017, anthropogenic tracers such as CH4AGR, CH4ENE, and CH4WST show significant increasing trends (2.3 ppb yr$^{-1}$, 6.9 ppb yr$^{-1}$ , and 1.6 ppb yr$^{-1}$, respectively) and correlate quite well with

total methane (R = 1.0) whereas all other tracers except CH4OAT show decreasing trends and poor correlations with total methane. On the other hand, CH4WET shows a significant decreasing trend during this period (-4.6 ppb yr$^{-1}$) and an anticorrelation with total methane. The decreasing trend of CH4WET is due to higher CH4WET sinks (217 Tg yr$^{-1}$) than sources (206 Tg yr$^{-1}$) during this period. During 1983-1998, wetland emission growth is larger than anthropogenic emission growth due to emission optimization in S0Wopt, leading to the dominancy of wetland to drive global methane growth.

During 1999-2006, when methane stabilizes, increases in methane emissions from energy and waste sectors dominate the increases in total methane sources as well as their tagged tracers (i.e., CH4ENE and CH4WST), whereas increases in methane sinks dominate all other tracers. Therefore, the imbalance between total methane sinks and sources dominate the total methane trend, which is also the case in S0Aopt during this time period. During 2007-2017, the energy is the major contributor to the renewed methane growth similar to that in S0Aopt.

As shown in Figures 7 and 8, OH levels show a slight decrease and methane sinks are relatively stable during 2007-2013, but large interannual variability exists during 2013-2017. Decreasing OH levels could lead to increases in methane lifetime and therefore methane buildup. Combined with increases in the emissions, methane starts to increase again during this period. However, it is difficult to separate the contributions from methane emissions and sinks as optimized methane emissions are based on methane mass balance (e.g., changes in the methane loss would act as a feedback on estimates of optimized total

emissions). However, it is clear that the decrease in OH levels alone (e.g., if emissions are kept constant) would not be enough to reproduce the renewed growth. The remaining question then is which emission sector(s) is (are) the major contributor(s) to the renewed growth over 2007 to 2017. Both S0Wopt and S0Aopt suggest that energy is the major sector contributing to renewed methane growth. However, both cases depend largely on the initial emission inventory. For example, S0Wopt relies on the emission growth of other sectors from the initial emission inventory, which means if the





emission growth of a certain sector is overestimated or underestimated in the initial emission inventory, it would give a different result.

Based on evidence from isotopic composition ($\delta^{13}CH_4$), recent studies suggest increasing wetland emissions may be responsible for the renewed growth of methane (Dlugokencky et al., 2009; Nisbet et al., 2016). To test this hypothesis in our modeling framework, we conducted another sensitivity simulation for 2006-2014, by repeating 2006 anthropogenic
emissions for all the years but adjusting wetland emissions to ensure that the total methane emissions are the same as in S0Wopt (or S0Aopt), which would imply that the increases in methane emissions are only due to the increases in wetland emissions. This sensitivity simulation is referred to as "S0A06" and the trends for source tagged tracers and total methane are shown in Figure S5 in the supplement. Interestingly, in S0A06, anthropogenic tracers still an increasing trend during 2007-2014, with the trend in CH4ENE dominating (trend = 3.6 ppb yr$^{-1}$ and R = 1.0), whereas CH4WET shows a small
decreasing trend (trend = -1.0 ppb yr$^{-1}$ and R = -0.8) despite rising emissions. As OH levels slightly decrease during this time period, with constant emissions except wetland, one might expect possible increasing trends in all tagged tracers except CH4WET. In fact, CH4AGR, CH4ENE, CH4WST, and CH4BMB increase over 2007-2014 in S0A06, but at a slower rate than in S0Wopt (and S0Aopt) due to no emission growth for these tracers. On the other hand, the decreasing OH levels (Figure 8) would lead to less methane sink and therefore higher methane concentrations. Since methane loss is proportional
to the product of OH levels and methane concentrations and concentrations of CH4WET are much greater than other source tagged tracers, the loss of CH4WET is also much higher than other tracers. Higher CH4WET loss (224 Tg yr$^{-1}$) than CH4WET sources (207 Tg yr$^{-1}$) leads to a decreasing trend in CH4WET. Nevertheless, S0A06 results still suggest that the renewed growth during 2007-2014 is dominated by the increases of CH4ENE, which means OH trends play an important role in determining the increasing trend of total methane since emissions of the energy sector are kept constant in this
sensitivity simulation. In addition, increases in wetland emissions alone are not able to drive increases in CH4WET over this period, as CH4WET sinks are equally important for determining the trend in CH4WET under constant anthropogenic emissions condition. Our analysis also suggests that increases in other microbial sources (e.g., agriculture and waste) would be needed to match the observed negative trend in $\delta^{13}CH_4$ since 2007 (Nisbet et al., 2019).

We perform an additional sensitivity simulation to test the possibility of wetland emissions driving the methane trend during
the period of renewed methane growth by combining the emissions of S0Aopt and S0Wopt as follows: S0Aopt emissions for 1980-2005 and S0Wopt emissions for 2006-2014. This simulation is referred to as "S0Comb"; the trends for source tagged tracers and total methane are shown in Figure S6 in the supplement. For 2007-2014, all anthropogenic tracers show decreasing trends except CH4ENE (2.7 ppb yr$^{-1}$), whereas CH4WET shows a significant increasing trend (6.3 ppb yr$^{-1}$) and dominates the total methane trend. This is mainly due to lower anthropogenic emissions during this period than previous
periods, allowing sinks of anthropogenic methane tracers to start to take over their trends except for CH4ENE. At the same time, significantly higher wetland emissions during this period than previous periods dominate the increasing trend of CH4WET. Interestingly, even with the same wetland emissions in S0Wopt and S0Comb for 2006-2014, CH4WET shows different trends. This is mainly because the CH4WET concentrations at the beginning of 2006 are much lower in S0Comb



than in S0Wopt. Therefore, CH4WET loss is much lower in S0Comb (190 Tg yr⁻¹) compared to S0Wopt (220 Tg yr⁻¹) over

this time period, leading to an increasing CH4WET trend in S0Comb, but a decreasing trend in S0Wopt. S0Comb results

suggest the need for a significant increase in wetland emissions along with decreases in anthropogenic emissions starting in

2006, compared to the stabilization period, for wetland emissions to drive renewed growth in methane. However, this is a

less likely scenario as both top-down and bottom-up inventories indicate anthropogenic emissions increasing over 2007-

2014. A more likely scenario is that both anthropogenic and wetland emissions increase (i.e., higher during 2007-2014 than

1999-2006). However, in that case, the dominance of wetland emissions in driving the total methane trend would decrease

based on our analysis.

## 3.4 Sensitivity to OH levels

As described in Section 2.2, we perform two additional simulations for low and high OH levels (i.e., S1 and S2) for 1980-

2017 to investigate the sensitivity of methane predictions to different OH levels. For both OH cases, the interannual

variations in OH levels are the same as in S0 because the simulations are driven by the same meteorology. Figures 13(a) and

(b) show global tropospheric OH concentrations, methane OH loss, and methane tropospheric lifetime for the three cases

(i.e., S0, S1, and S2) in which wetland emissions are optimized (Wopt; Aopt shows a very similar global OH trend as Wopt).

Compared to S0, scaling LNO$_x$ production in the model by a factor of 0.5 leads to a reduction in simulated annual global

mean OH levels by -6.4 % in S1 over 1980-2017; scaling by a factor of 2 leads to an increase in simulated annual global

mean OH by +9.1% in S2 . The global mean OH levels increase from 1980 to 2008 (by 3.6%, with respect to 1980 level)

with a linear rate of increase of $4.1 \times 10^3$ molecule cm⁻³ yr⁻¹, a decrease from 2008 to 2015 (by 2.3%, with respect to 2008

level) with a mean rate of $-7.1 \times 10^3$ molecule cm⁻³ yr⁻¹, and an increase from 2015 to 2017 (by 4.6%, with respect to 2015

level) with a mean rate of $3.2 \times 10^4$ molecule cm⁻³ yr⁻¹. However, compared to the 1998-2007, OH levels during 2008-2015

and 2015-2017 are still higher by 2.5% and 1.3%, respectively. Changes in OH levels depend on a number of factors (e.g.,

temperature, water vapor, O$_3$, NO$_x$, CO, and VOCs). Therefore, OH is influenced by the specific chemistry and forcing data

used in the model. Since emission optimization is also based on methane sinks, the total optimized emissions in S1 are lower

than those in S0 by about 4.1% (with an annual mean of -23.7 Tg yr⁻¹), and the total optimized emissions in S2 are higher

than those in S0 by about 5.8% (or 33.4 Tg yr⁻¹). This indicates that a 1% change in OH levels could lead to about 4 Tg yr⁻¹

difference in the optimized emissions. Increasing methane loss due to OH is simulated for 1980 to 2007 in the three cases

due to increases in OH and methane concentrations (except over the stabilization period when methane was not increasing

but OH was increasing). During 2007-2013, the simulated decrease in OH levels combined with increasing methane

concentrations leads to relative stabilization in methane OH loss in the three cases. The large interannual variability in OH

levels during 2013-2017 dominates the interannual variability in methane OH loss despite the continued increases in

methane.

All three simulations show a similar trend for tropospheric methane lifetime, with a linear decrease from 1980 to 2007 (-0.04

year yr⁻¹ in S0, -0.05 year yr⁻¹ in S1, and -0.03 year yr⁻¹ in S2), a clear increasing trend during 2011-2015 (0.08 year yr⁻¹ in



all three simulations), and a decreasing trend during 2015-2017(-0.2 year yr$^{-1}$ in all three simulations). The mean tropospheric methane lifetime due to OH loss for 1980-2017 is 9.9±0.4 years in S0Wopt, which is about 0.5 year lower than S1Wopt (10.4±0.5 years), and about 0.7 year higher than S2Wopt (9.2 ±0.3 years), due to different OH levels and therefore

methane sinks, but with similar methane burdens. This indicates that a 1% change in OH levels could lead to about 0.08 year difference in the tropospheric methane lifetime. The mean tropospheric methane lifetimes simulated by the three simulations are within the uncertainty range of observation-derived estimates for the 2000s (Prather et al., 2012) and model estimates (Voulgarakis et al., 2013; Naik et al., 2013). All simulations show an increase in methane lifetime during 2011-2015, which could be a signal of the methane feedback on its lifetime (Holmes, 2018) in the model. Continued increases in methane

emissions (Figure 7) during this time period, along with decreases in tropospheric OH concentrations (Figure 13), lengthen the lifetime of methane and therefore amplify the methane's response to emission changes. If methane emissions continue to increase, we can expect stronger increases in atmospheric methane due to the amplifying effect of the methane-OH feedback as occurred during in the significant increases in methane growth rates during 2014 and 2015.

## 4 Conclusions

In this work, we thoroughly evaluate the methane budget simulated by the GFDL-AM4.1 and apply the model to quantify changes in global methane over the past decades. We simulate the DMF of methane and related tracers for 1980 to 2017 by driving the model with gridded emissions compiled from various sources. In order to match the long-term record of surface methane measurements, we optimize global total methane emissions using a simple mass-balance approach. Our optimized global total methane emissions are within the range of estimates by previous studies (both bottom-up and top-down). The

GFDL-AM4.1 simulations with emissions following two different optimizations (anthropogenic and wetlands) both reproduce observed methane growth rates for the different time periods (e.g., 9.4 ppb yr$^{-1}$ during 1984-1998, 1.1 ppb yr$^{-1}$ during 1999-2006, and 6.1 ppb yr$^{-1}$ during 2007-2017). The simulations are also able to capture the spatial distribution of methane as retrieved by satellites and vertical distribution of methane as measured from aircraft. Both simulations also reproduce observed global methane trends and variabilities, despite the different contributions from anthropogenic and

wetland emissions. This therefore suggests that the accurate estimates of global total emissions and of the interannual/seasonal variability in emissions are critical in predicting the global methane trend and its variability, despite uncertainties in the estimates of individual sources.

We then explore the causes of methane trends and variability over 1980-2017 to sources and sinks. The simulation with optimization of anthropogenic emissions shows anthropogenic emissions to be the major contributors to the rapid methane

growth during 1980s and 1990s, whereas the simulation with optimization of wetland emissions shows wetlands to be the major contributors during these periods. However, both simulations suggest increases in methane sources (mainly from energy and waste sectors), balanced by the increases in methane sinks (mainly due to increases in OH levels), lead to



methane stabilization during 1999-2006, and that the energy sector is the major contributor to the renewed growth of methane after 2006.

Two additional sensitivity simulations further investigate the contributions of wetlands to the methane renewed growth during 2007-2014. The simulation with repeating 2006 emissions for all the sectors except wetland suggests increases in wetland emissions alone are not able to explain the renewed methane growth because sinks are equally important for determining the trend under constant anthropogenic emissions. Results from a simulation with combined optimizations (i.e., 1980-2005 optimized anthropogenic emissions and 2006-2014 optimized wetland emissions) suggest that a significant

increase in wetland emissions along with decreases in anthropogenic emissions starting in 2006 compared to the stabilization period (1999-2006) is required for wetland emissions to drive renewed growth in methane, which is a less likely scenario.

Two additional sensitivity simulations, with low and high OH levels (by scaling $LNO_x$ production in the model by a factor of 0.5 and 2), further investigate methane OH loss and tropospheric lifetime. In general, OH trends dominate methane OH loss trends during different methane growth periods except 2007-2013, when methane OH loss shows little change due to the

decrease in OH levels combined with the increase in methane concentrations. The results also indicate that a 1% change in OH levels could lead to about 4 Tg yr$^{-1}$ difference in the optimized emissions and 0.08 year difference in the estimated tropospheric methane lifetime. The increasing methane lifetime during 2011-2015 in all the OH sensitivity simulations indicate a possible methane feedback on its lifetime in the model. Continued increases in methane emissions along with decreases in tropospheric OH concentrations extend the lifetime of methane and therefore amplify methane's response to

emission changes.

Essentially, the global atmospheric methane trend is driven by the competition between its emissions and sinks. Emissions dominate sinks leading to an increasing trend while sinks dominate emissions leading to a decreasing trend. Our model results suggest that the methane stabilization during 1999-2006 is mainly due to increasing emissions balanced by increasing sinks, whereas the methane renewed growth during 2007-2013 is mainly due to increasing sources combined with little

change in sinks despite small decreases in OH levels. The significant increases in methane growth during 2014-2015 are mainly due to increasing sources combined with decreasing sinks. Most of the model simulations conducted here suggest that increases in energy sources drive the renewed methane growth, in agreement with previous studies (e.g., Rice et al., 2016; Hausmann et al., 2016; Worden et al., 2017), but in disagreement with other studies that consider emissions from microbial sources as the major contributor (e.g., Nisbet et al., 2016; Schaefer et al., 2016). However, optimization of

emissions from anthropogenic sources depends on the "shares" of individual anthropogenic sectors in the initial emission inventories. Uncertainties in these shares could lead to uncertainties in the emission adjustment for each anthropogenic sector. Recent studies using methane isotopic composition suggest that renewed growth in methane is more likely due to the increases in biogenic sources (e.g., Schaefer et al., 2016) as the ratio $\delta^{13}C$ is shifting to more negative values since 2007. However, it also implies increases in isotopically lighter fossil fuel emissions, or decreases in isotopically heavy sources

(e.g., biomass burning), or increases in both microbial and fossil fuel emissions but with increases in microbial emissions stronger than those from fossil fuel sources (Nisbet et al., 2019). It is quite possible that, rather than the energy sector, the



increases in the agriculture and waste sectors may drive the renewed methane growth. In that case, it is possible that the growth of agriculture and waste emissions could be underestimated in the optimized emissions, while the growth of energy emissions could be overestimated.

The optimized emission totals estimated in this work represent temporal and spatial distribution of methane total sources reasonably well. However, the emission adjustments are either applied to anthropogenic sectors only or wetland sector only. Uncertainties therefore exist on the distribution of the emission adjustments to individual sectors. Without accurate estimates of emissions from individual sources, it would be difficult to attribute the methane trend and variability to specific sectors. The application of methane isotopes and additional observational constraints (e.g., ethane and $\delta^{13}CH_4$) could potentially help

better partition the emission adjustments to different sectors. In addition, the spatial distribution of optimized emissions depends on the spatial information in the initial emission inventories. Uncertainties in the spatial distribution from the initial emission inventories may remain in the optimized emissions. Our model evaluation suggests that the optimized inventory may overestimate tropical emissions. A process-based emission model (e.g., wetland emissions) coupled with AM4.1 may better represent the spatial and temporal patterns of the emissions than was possible in the present work.

**Author contribution**

Jian He and Vaishali Naik designed the research. Jian He developed the model configuration, performed model simulations, analyzed model results, and prepared the manuscript with contributions from all co-authors. Vaishali Naik provided GFDL-model ready CMIP6 emissions. Larry Horowitz provided meteorological data for nudging. Ed Dlugokencky provided surface observations. Kirk Thoning provided scripts to process observational data. All authors contributed to the discussion

of results.

**Acknowledgements**

This work is supported by the Carbon Mitigation Initiative at Princeton University. Atmospheric methane dry air mole fractions are obtained from the NOAA ESRL Carbon Cycle Cooperative Global Air Sampling Network (Dlugokencky et al., 2018, ftp://aftp.cmdl.noaa.gov/data/trace_gases/ch4/flask/surface/). The globally averaged marine surface monthly mean

data and annual mean growth rates are obtained from www.esrl.noaa.gov/gmd/ccgg/trends_ch4/. HIPPO data is obtained from Wofsy et al. (2012), Merged 10-second Meteorology, Atmospheric Chemistry, Aerosol Data (R_20121129). Satellite data is obtained from http://www.temis.nl/climate/methane.html. We are grateful to Prabir Patra for providing methane emissions for nearshore exchange and mud volcanoes. We also thank Fabien Paulot for processing sea surface temperatures and sea ice data and the GFDL model development team for developing the AM4.1.



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




**Table 1. Emission inventories used in this study**

| Source Category | Database | Temporal Variability | References |
|---|---|---|---|
| Anthropogenic | CEDS v2017-05-18 | 1980-2014 monthly data | Hosely et al. (2018) |
| | SSP2-4.5 | 2015-2017 monthly data | Gidden et al. (2018) |
| Biomass Burning | BB4MIP | 1980-2014 monthly data | van Marle et al. (2017) |
| | SSP2-4.5 | 2015-2017 monthly data | Gidden et al. (2018) |
| Wetlands | WetChart v1.0 | Climatological monthly mean (with seasonal variability) for 1980-2017 | Bloom et al. (2017) |
| Ocean | MOZART | Climatological monthly mean (with seasonal variability) for 1980-2017 | Brasseur et al. (1998) |
| Near-shore | TransCom-CH4 | Climatological annual mean (no seasonal variability) for 1980-2017 | Lambert and Schmidt (1993), Patra et al. (2011) |
| Termites | NASA-GISS | Climatological annual mean (no seasonal variability) for 1980-2017 | Fung et al. (1991) |
| Mud volcanoes | TransCom-CH4 | Climatological annual mean (no seasonal variability) for 1980-2017 | Etiope and Milkov (2004), Patra et al. (2011) |

**Table 2. List of simulations conducted using GFDL-AM4.1 to explore trends and variability in methane**

| Simulations | Description |
|---|---|
| S0Aopt | Standard AM4.1 configuration, but with optimized anthropogenic emissions |
| S0Wopt | Standard AM4.1 configuration, but with optimized wetland emissions |
| S1Wopt | AM4.1 configuration with low OH levels ($LNO_x$ emissions scaled by a factor of 0.5), and optimized wetland emissions |
| S2Wopt | AM4.1 configuration with high OH levels ($LNO_x$ emissions scaled by a factor of 2), and optimized wetland emissions |





**Table 3. Global methane budget (Tg CH$_4$ yr$^{-1}$) during 1980-2017**

| Period of time | 1980-1989 | 1990-1999 | 2000-2009 | 2003-2012 | 1999-2006 | 2007-2017 |
|---|---|---|---|---|---|---|
| **Sources** | | | | | | |
| Natural sources | 203<br>203 [150-267][a]<br>355 [244-466][b] | 203<br>182 [167-197][a]<br>336 [230-465][b] | 203<br>218 [179-273][a]<br>347 [238-484][b]<br>234 [194-292][c]<br>382 [255-519][d] | 203<br>231 [194-296][c]<br>384 [257-524][d] | 203 | 203 |
| Natural wetlands | 166<br>167 [115-231][a]<br>225 [183-266][b] | 166<br>150 [144-160][a]<br>206 [169-265][b] | 166<br>175 [142-208][a]<br>217 [177-284][b]<br>166 [125-204][c]<br>183 [151-222][d] | 166<br>167 [127-202][c]<br>185 [153-227][d] | 166 | 166 |
| Other natural sources | 37 | 37 | 37 | 37 | 37 | 37 |
| Oceans | 9.5 | 9.5 | 9.5<br>18 [2-40][b]<br>14 [5-25][d] | 9.5 | 9.5 | 9.5 |
| Termites | 20 | 20 | 20 | 20 | 20 | 20 |
| Mud volcanoes | 7.5 | 7.5 | 7.5 | 7.5 | 7.5 | 7.5 |
| Anthropogenic sources | 289<br>348 [305-383][a]<br>308 [292-323][b] | 311<br>372 [290-453][a]<br>313 [281-347][b] | 340<br>335 [273-409][a]<br>331 [304-368][b]<br>319 [255-357][c]<br>338 [329-342][d] | 358<br>328 [259-370][c]<br>352 [340-360][d] | 328 | 377 |
| Agriculture and waste | 159<br>208 [187-220][a]<br>185 [172-197][b] | 172<br>239 [180-301][a]<br>188 [177-196][b] | 185<br>209 [180-241][a]<br>200 [187-224][b]<br>183 [112-241][c]<br>190 [174-201][d] | 191<br>188 [115-243][c]<br>195 [178-206][d] | 181 | 200 |
| Biomass burning | 13 | 18 | 15<br>18 [15-20][d] | 15<br>18 [15-21][d] | 15 | 14 |
| Fossil fuels | 104<br>94 [75-108][a]<br>89 [89-89][b] | 107<br>95 [84-107][a]<br>84 [66-96][b] | 127<br>96 [77-123][a]<br>96 [85-105][b]<br>101 [77-126][c]<br>112 [107-126][d] | 139<br>105 [77-133][c]<br>121 [114-133][d] | 120 | 150 |
| Other anthropogenic sources | 14 | 14 | 13 | 13 | 12 | 13 |
| ΔE[e,f] | 47 [23-79] | 60 [36-94] | 52 [29-85] | 44 [21-79] | 57 [34-93] | 40 [17-73] |
| **Sinks[f]** | | | | | | |
| Total chemical loss | 486 [462-519]<br>490 [450-533][a]<br>539 [411-671][b] | 540 [516-573]<br>525 [491-554][a]<br>571 [521-621][b] | 577 [553-610]<br>518 [510-538][a]<br>604 [483-738][b] | 584 [560-617]<br>515[c] | 570 [546-603] | 592 [568-625] |




| | | | 514[c] | | | |
|---|---|---|---|---|---|---|
| OH loss | 442 [419-476] 468 [382-567][b] | 486 [462-519] 479 [457-501][b] | 526 [502-559] 528 [454-617][b] | 534 [510-567] | 519 [495-552] | 542 [519-576] |
| O1D loss | 38 46 [16-67][b] | 47 67 [51-83][b] | 43 51 [16-84][b] | 43 | 44 | 42 |
| Cl loss | 5 25 [13-37][b] | 7 25 [13-37][b] | 7 25 [13-37][b] | 7 | 8 | 7 |
| Soils | 13 21 [10-27][a] 28 [9-47][b] | 14 27 [27-27][a] 28 [9-47][b] | 14 32 [26-42][a] 28 [9-47][b] 32 [27-38][c] | 14 33 [28-38][c] | 14 | 14 |
| **Totals**[f] | | | | | | |
| Sum of sources | 539 [515-571] 551 [500-592][a] 663 [536-789][b] | 574 [549-608] 554 [529-596][a] 649 [511-812][b] | 595 [572-628] 548 [526-569][a] 678 [542-852][b] 552 [535-566][c] 719 [583-861][d] | 605 [582-640] 558 [540-568][c] 736 [596-884][d] | 589 [565-625] | 620 [597-653] |
| Sum of sinks | 499 [475-532] 511 [460-559][a] 539 [420-718][b] | 554 [530-586] 542 [518-579][a] 596 [530-668][b] | 591 [567-624] 540 [514-560][a] 632 [592-785][b] 546[c] | 598 [574-632] 548[c] | 584 [560-617] | 606 [582-639] |
| Imbalance | 40 [39-40] 30 [16-40][a] | 20 [19-22] 12 [7-17][a] | 4 [4-5] 8 [-4-19][a] 6[c] | 7 [8-8] 10[c] | 5 [5-8] | 14 [15-14] |
| Atmospheric growth | 36 34[a] 32[g] | 19 17[a,g] | 4.8 6[a,g] 6.0 [4.9-6.6][c] | 7.4 10.0 [9.4-10.6][c,g] | 3.5 1.9±1.6[g] | 16.6-17.2 18.9±1.7[g] |

[a]Values are based on Kirschke et al. (2013) top-down approach.
[b]Values are based on Kirschke et al. (2013) bottom-up approach.
[c]Values are based on Saunois et al. (2016) top-down approach.
[d]Values are based on Saunois et al. (2016) bottom-up approach.
[e]ΔE is calculated based on the methodology of Ghosh et al. (2015).
[f]The ranges are based on the low OH (S1Wopt) and high OH cases (S2Wopt) and the decadal mean values shown in the table are based on the default OH (S0Wopt).
[g]The observed atmospheric growth rates (Tg yr$^{-1}$) are estimated based on a few MBL sites (Dlugokencky et al., 2018), which are not the same as the Imbalance Row (based on the entire globe).






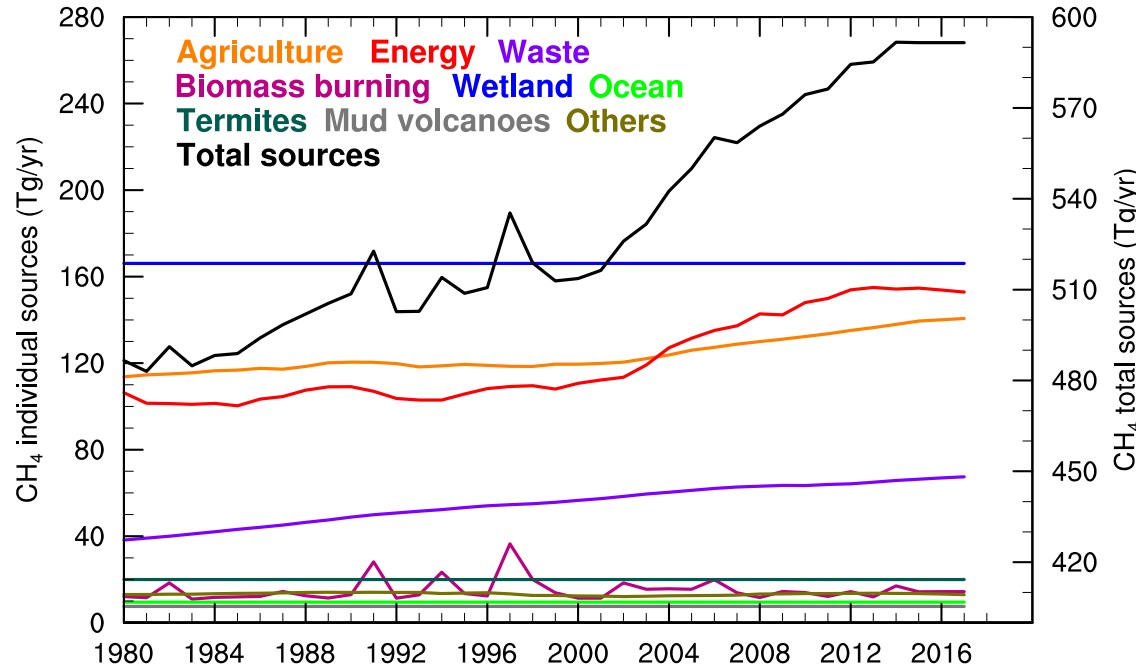

**Figure 1. Time series of methane emissions from the initial methane inventories for the period of 1980-2017. The emissions for major sectors are shown on the left y axis, including agriculture sector, energy production sector, waste sector, biomass burning sector, wetland sector, ocean and near-shore fluxes, termites, mud volcanoes, and other sources (i.e., industrial processes, surface transportation, international shipping, residential, commercial, and others). The total methane emissions from the initial emission inventories (black line) are shown on the right y axis.**



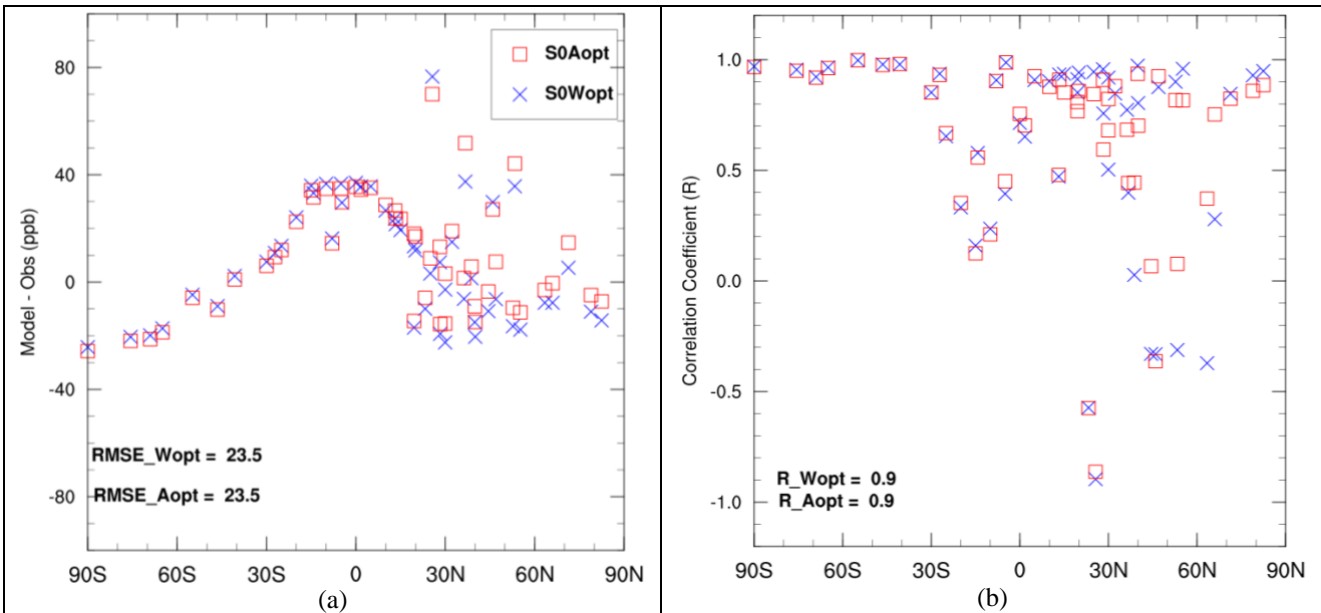

**Figure 2. Model bias (a) and correlation coefficient (b) of simulated climatological mean surface methane concentrations against NOAA GMD observations for the 1983-2017 time period. GMD sites with at least 20-year observations are selected for model climatological evaluation. In Fig.2a, each red square or blue cross represents model mean bias by S0Aopt or S0Wopt at the corresponding GMD site. Root-mean-square-error (RMSE) is shown for all the GMD sites in Fig.2a. In Fig.2b, each red square or blue cross represents correlation of climatological seasonal variability by S0Aopt or S0Wopt at the corresponding GMD site. Spatial correlation (R) is shown for all the GMD sites in Fig.2b.**







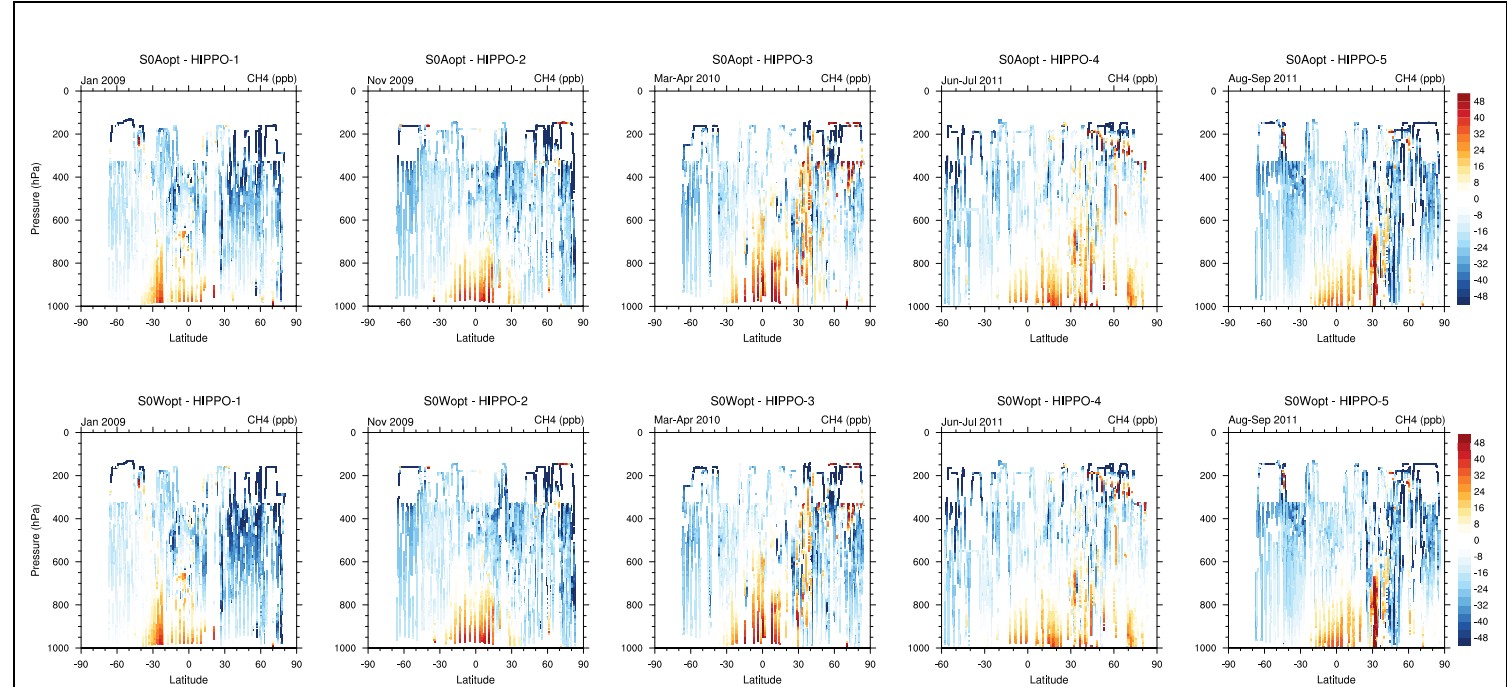

**Figure 3.** Comparison of vertical distribution of methane from S0Aopt and S0Wopt simulations with measurements from individual HIPPO campaigns. Months of campaign are given at the top left of the individual plots.



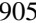


**Figure 4. Comparison of GFDL-AM4.1 simulated methane trends with NOAA-GMD surface observations. Dash line represents smoothed trends (i.e., 12-month running mean) from deseasonalized data. A meridional curve (Tans et al., 1989) was fitted through NOAA-GMD site observations to get the latitudinal distribution of methane. A function fit consisting of yearly harmonics and a polynomial trend, with fast fourier transform and low pass filtering of the residuals are applied to the monthly mean methane DMF (Thoning et al., 1989; Thoning, 2019) to approximate the long-term trend.**





**Figure 5. Comparisons of GFDL-AM4.1 simulated methane growth rates with NOAA-GMD surface observations.**




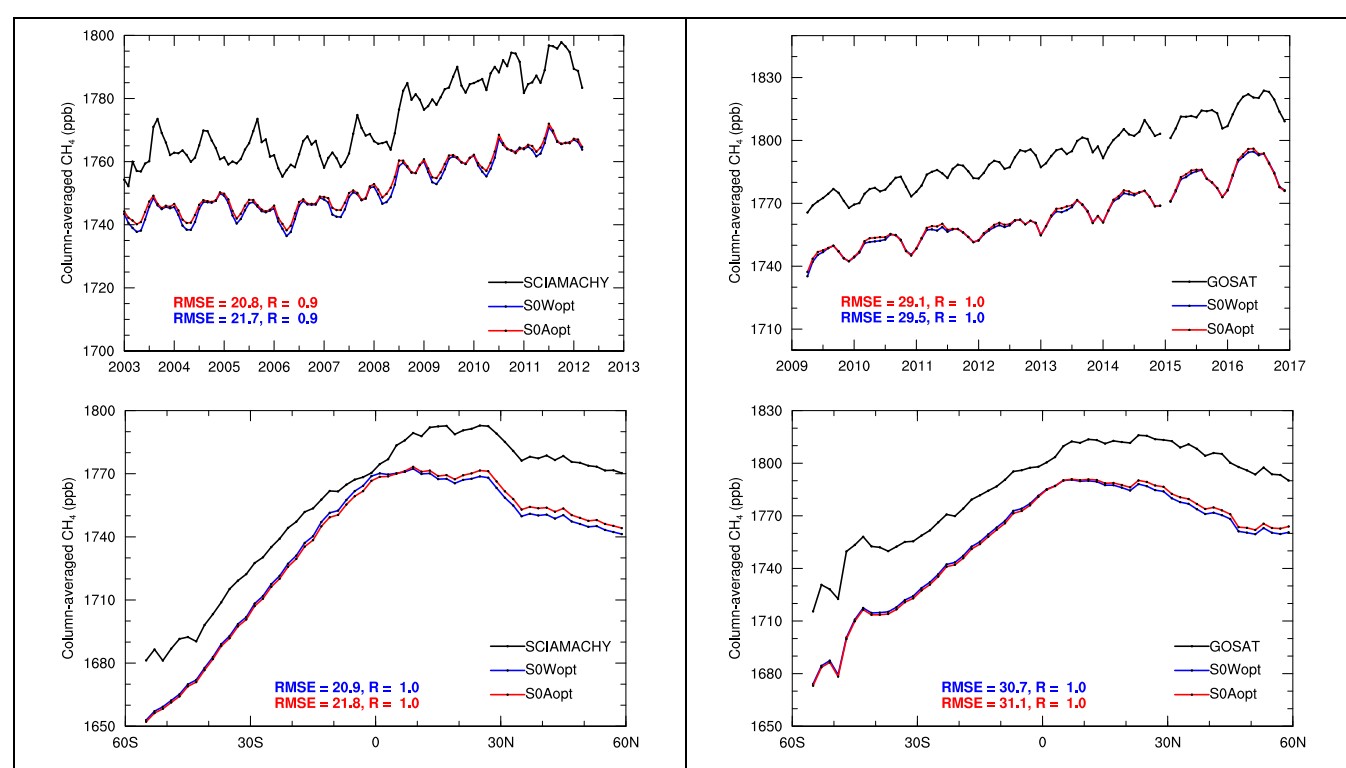

**Figure 6. Comparisons of column-averaged methane concentrations with SCIAMACHY (left) and GOSAT (right) satellite retrievals.**

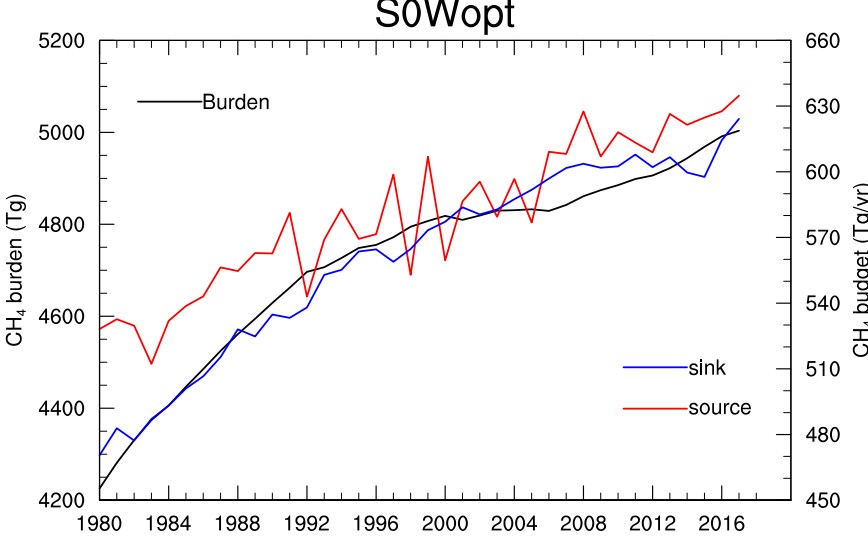

**Figure 7. Time series of global methane burden (black line, left Y axis), methane sources (red line, right Y axis), and methane sinks (blue line, right Y axis) by S0Wopt.**



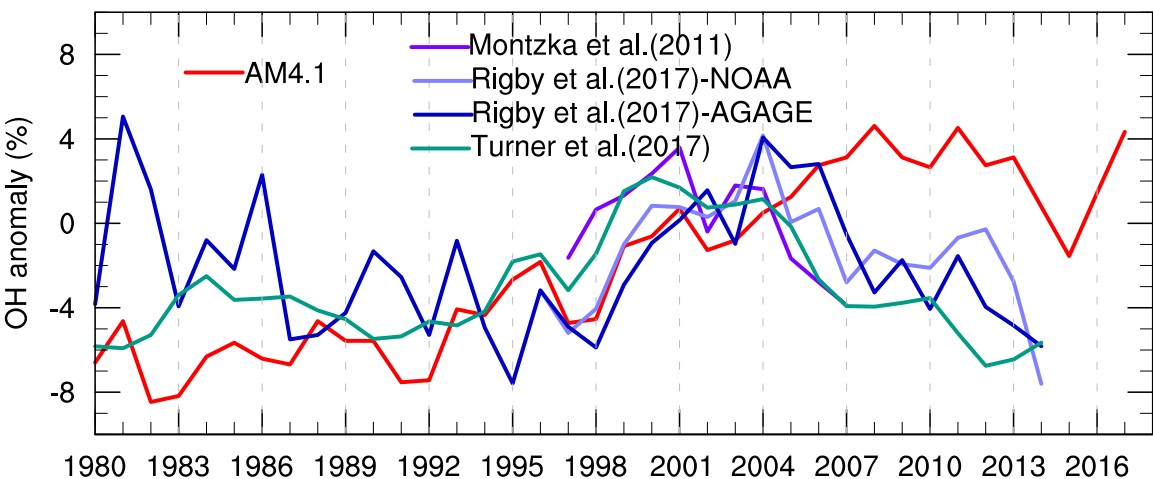

**Figure 8. Time series of global tropospheric OH anomalies with respect to 1998-2007. Results of Montzka et al. (2011) are shown in dark purple (with the mean interannual variability of OH as ±2.3% for the period of 1998-2007). Results of Rigby et al. (2017) derived from NOAA observations are shown in light blue (with the mean interannual variability of OH as ±2.3% for the period of 1998-2007 and ±2.6% for the period of 1980-2014), and derived from AGAGE observations are shown in dark blue (with the mean interannual variability of OH as ±3.0% for the period of 1998-2007 and ±3.1% for the period of 1980-2014). Results from Turner et al. (2017) are shown in green (with the mean interannual variability of OH as ±2.0% for the period of 1998-2007 and ±2.5% for the period of 1980-2014). OH anomalies in this work are shown in red (with the mean interannual variability of OH as ±2.2% for the period of 1998-2007 and ±4.1% for the period of 1980-2014).**



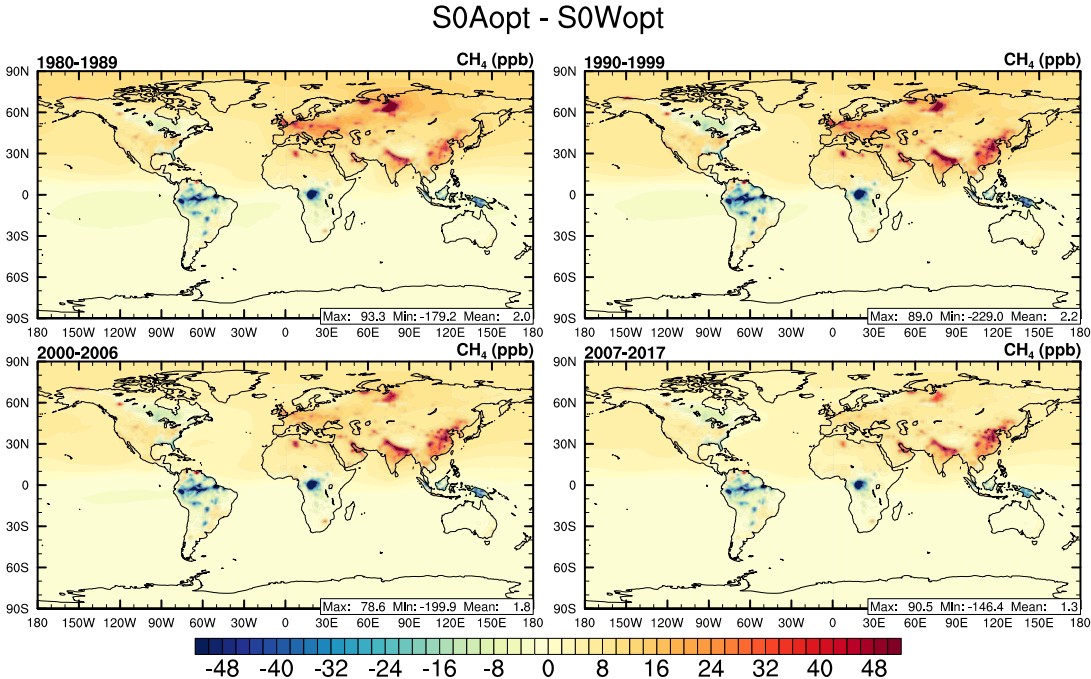

**Figure 9. Absolute difference in surface CH₄ predictions between S0Aopt and S0Wopt for the periods of 1980-1989, 1990-1999, 2000-2006, and 2007-2017.**

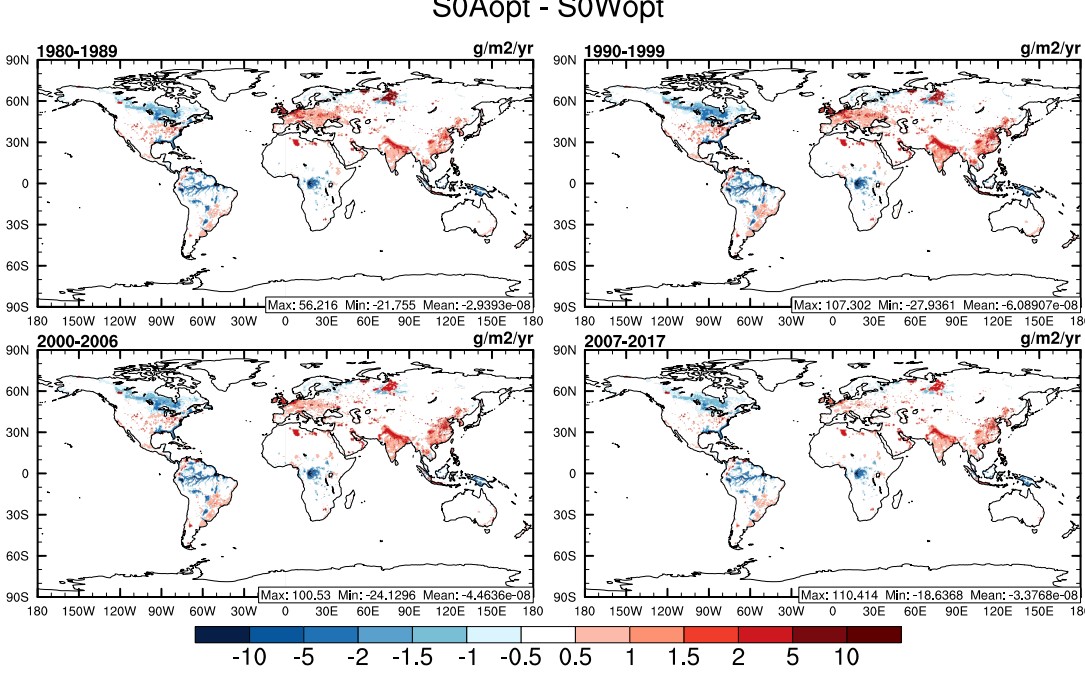

**Figure 10. Absolute difference in total annual methane emissions between S0Aopt and S0Wopt for the periods of 1980-1989, 1990-1999, 2000-2006, and 2007-2017.**



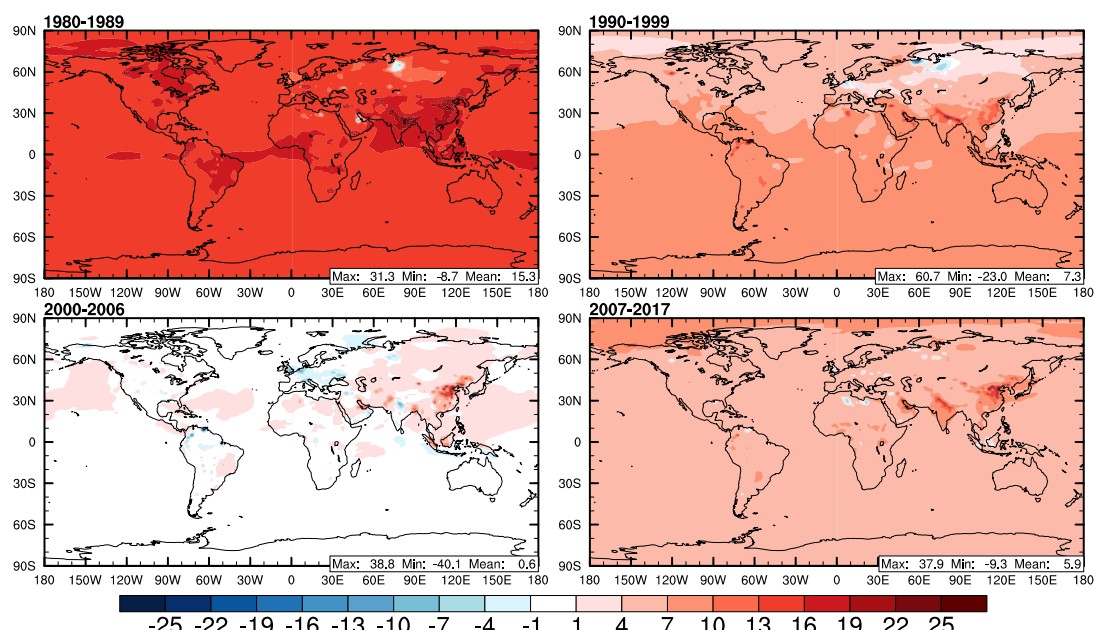


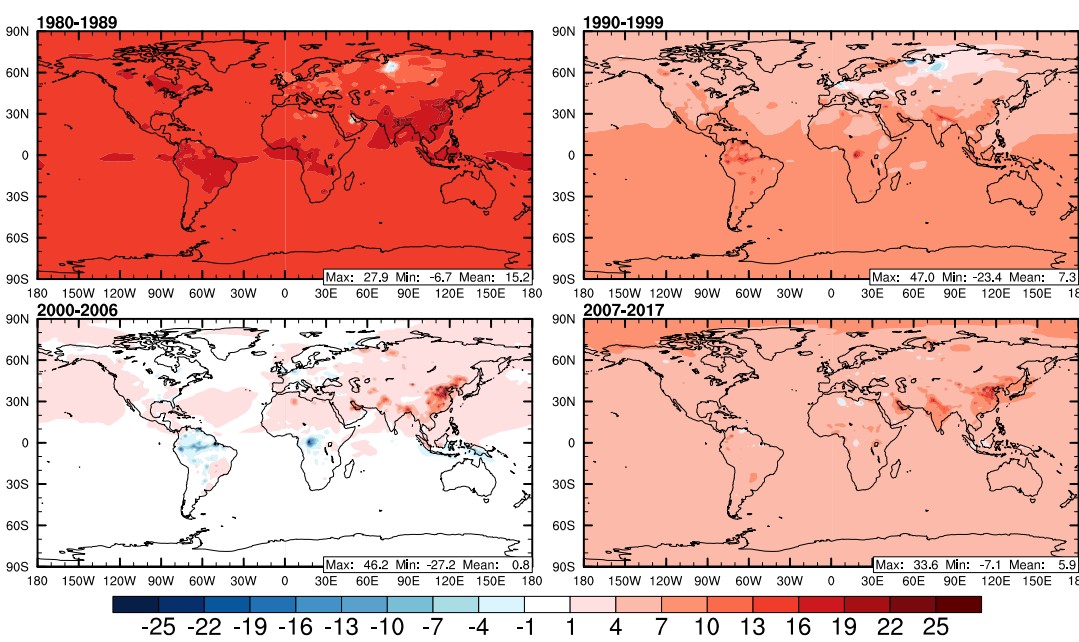

**Figure 11.** Spatial distribution of surface methane growth (ppb/yr) by S0Aopt (upper panel) and S0Wopt (lower panel) for the periods of 1980-1989, 1990-1999, 2000-2006, and 2007-2017.

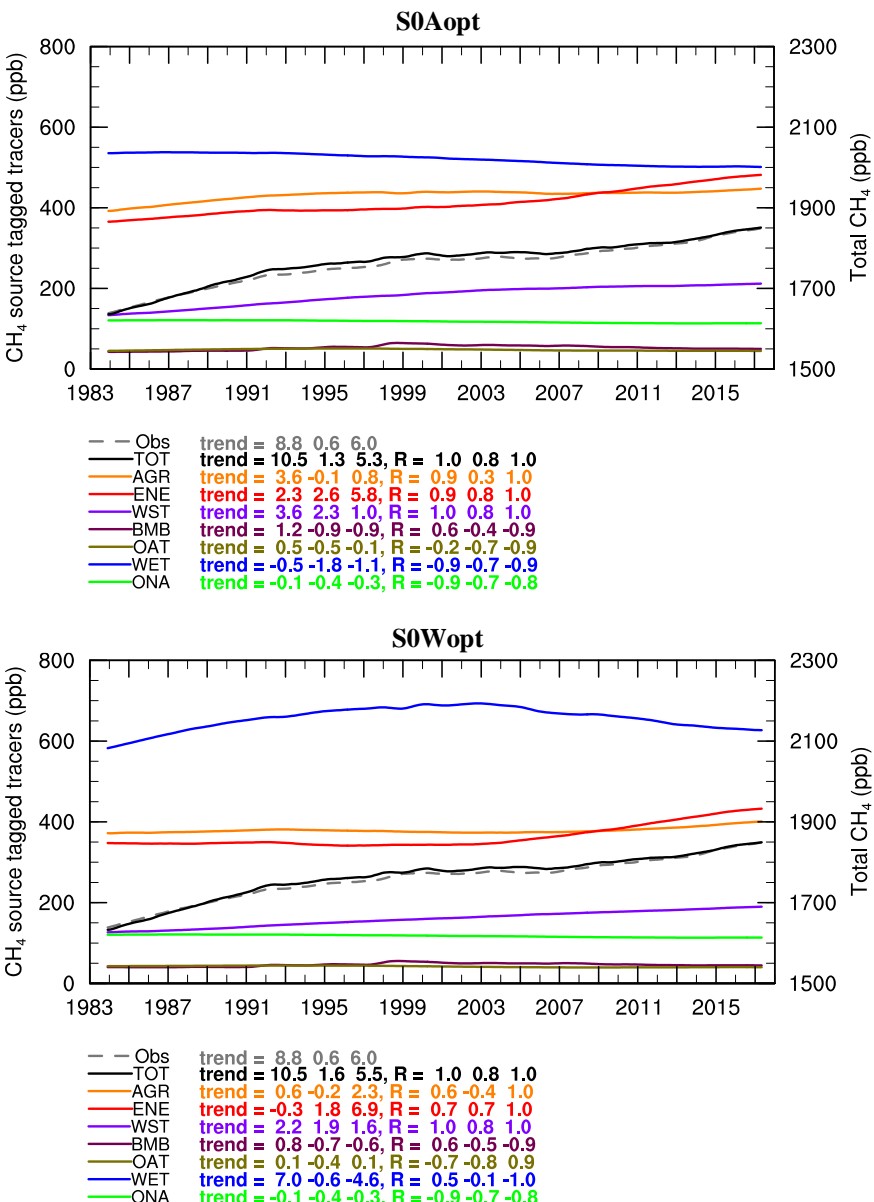

**Figure 12. Global deseasonalized methane trends for source tagged tracers (left y axis) and total methane (TOT, right y axis). The source tagged tracers include tracer for agriculture sector (AGR), energy sector, (ENE), waste sector (WST), biomass burning sector (BMB), other anthropogenic sectors (OAT), wetland sector (WET), and other natural sectors (ONA). The gray dashed line represents total methane trend from NOAA-GMD observations. The trends (i.e., linear growth rates, ppb yr⁻¹) for the periods of 1983-1998,1999-2006, and 2007-2017 are shown below the figure. The correlation coefficient (i.e., R) for TOT is compared to observations and for other source tagged tracers is compared to total methane for three periods.**





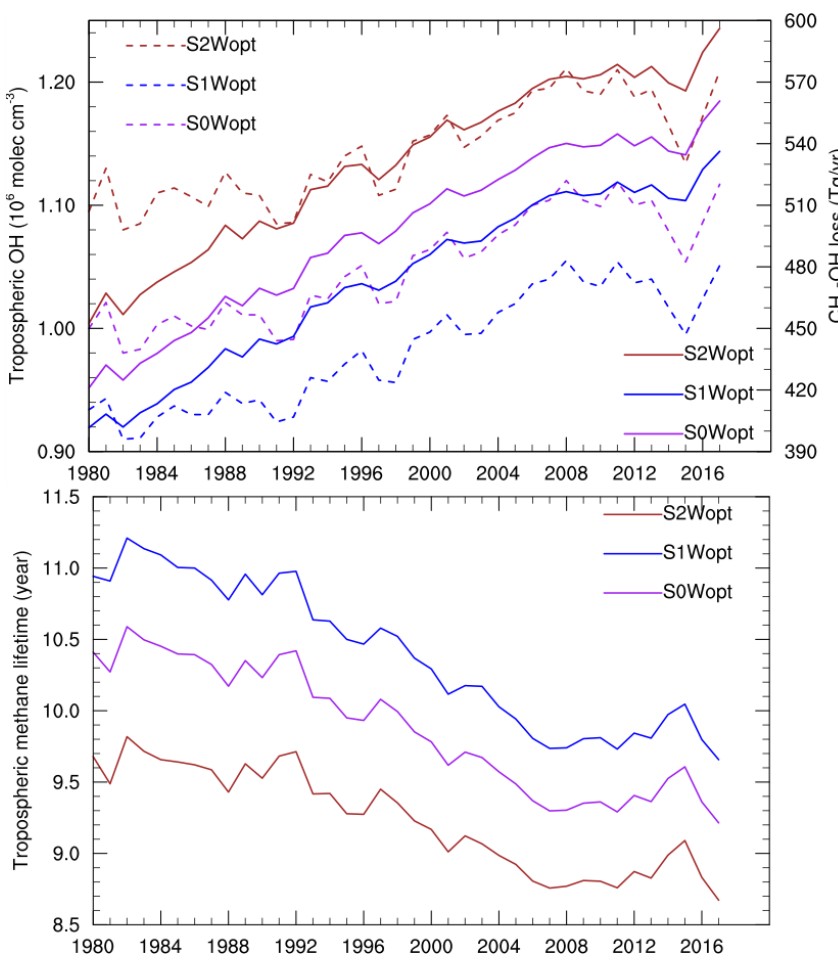

**Figure 13. Time series of global tropospheric OH levels (left Y axis, dash line) and methane OH loss (right Y axis, solid line) from S0Wopt (purple), S1Wopt (blue), and S2Wopt (brown) in the upper panel and time series of methane tropospheric lifetime from S0Wopt (purple), S1Wopt (blue), and S2Wopt (brown) in the lower panel.**