# Peer review of "Investigation of the global methane budget over 1980-2017 using GFDL-AM4.1"

_Atmospheric Chemistry and Physics, 2019_

## Referee Comment (RC1) · Anonymous Referee #1 · 5 Aug 2019

The work describes CH4 budget using an atmospheric chemistry-transport model that is developed at the GFDL. The authors have taken in to account all possible causes of variabilities in CH4 budget, such as the emissions and loss due to tropospheric hydroxyl (OH). As shown in the manuscript, OH variability is of as much importance as the emissions in explaining the CH4 growth rate variabilties in different decades in the period of 1980s to 2010s. The manuscript is generally well written. However, I felt toward the end of the manuscript is a bit of stretch and could be reduced (I have made some suggestions in my specific comments). The manuscript can be accepted after a major revision.

Specific comments: Line 49 & 62ff: can the growth rate discussions in the introduction be made concise and put together at one place.

[Figure]

Line 80ff: I think there are other prominent inverse modelling results trying to explain the recent regrowth of CH4 concentrations.

Line 135-137: This is a quite strange statement. After reading the whole manuscript I do not believe you have tried to address these couple of issues to a great extent. May be remove?

Line 156: Not from wetchart? I mean does wetchart not have IAV?

Line 206: Not clear if this is after LNOx scaling? please make this statement precise (e.g., Control).

Line 249ff: "The meridional curve" needed some clarifications here, e.g., selected sites within a latitude band to get the mean CH4 at 5 different latitude bands or something like that.

Line 296: Sometimes the sites like Key Biscayne are sampled by moving the model grids to the ocean side. You might check that out.

Line 315ff: The tropical bias in all HIPPO is a bit strange! Not OH but transport (or emissions)? I am suspecting this because the bias due to OH would appear at all altitudes (timescale ∼1yr), because the bias is in the lower troposphere, if the vertical transport is slow, you would find more CH4 is accumulated in the lower troposphere (timescale∼week)

Line 326: do you run CH3CCl3 & SF6?, say within the CCMI framework?

Line 346ff: suggesting too much emissions in the NH, where most of Anthro emissions are...May be you can test this better by site-level comparisons.

Line 353ff: I cannot find this 1 year mismatch (please be clear), instead I find a persistent offset during 1984-1991 (how the major and minor ticks marked in Fig. 5; the labeled ticks only should be major?

Line 369: How can you say that? I thought your optimization was not good for this

period, because the number of observation sites may not have coverred the global reasonably well. I mean biased high toward the NH. Could you check how many SH sites you have data before 1988.

Line 374: Most likely due to an overestimation of China emissions (e.g., Saeki and Patra, GOSL, 2017, and references therein) (regional inversion is needed for adjusting such regional emission biases)

Line 378: "...which is also a remote site" and remote from China emissions

Lines 394ff: I am not very sure if the comparison with GOSAT/SCIA are adding any values to this work. Better be kept aside for a full paper, unless the reasons for the mean offsets are figured out and discussed. For instance you could compare your results with the ACE-FTS data to find out if there is any bias in the stratospheric CH4 as there is no significant offsets in the tropospheric CH4 is seen in comparison with surface data and HIPPO.

Line 426ff: The emission increase in the 1990s is apparently linked to OH increase in AM; which sector can provide this extra emissions. I think this result is very different from what I have seen in the literature, and thus needing some explanation. Surprisingly, the emission increase rate in the 1990s is greater than the recent regrowth period.

Lines 484ff: The discussions using Fig 9-11 aren't that interesting as presented. I would recommend the authors to move these plots to the supplement or show 1-2 panels in the main text; for example all the 4 panels in Fig 9 & 10 are essentially showing very similar distributions. The S0Aopt and S0Wopt are also showing similar behaviour. This is mainly because the emission (E)-a priori emissions are the same in both the simulations, and the correction emissions Del-E following Anthropogenic or Wetland emission patterns only play minor role.

I am actually curious if you could use some of the continental sites, e.g., NWR, LEF,

SGP or TAP, and use the model-measurement comparisons to say whether the S0Aopt or S0Wopt are more realistic.

Line 519: Such high correlations are a bit surprising, if I see the lines in Fig. 12. For example AGR show -ve trend, yet show positive correlation. How is that possible?

Line 633: This is similar to the essential conclusion in some other publications as well, where ENE and Animals were made responsible for the post-2006 CH4 growth rate. I guess it is extremely difficult to separate emissions from Animals and Wetlands by 13C signature in CH4.

Lines 638ff: I am curious if inconsistency between the tropospheric OH and CH4-loss by OH are arising from the spin-up. Did you spun-up the simulations using different OH from the 1970s?

---

## Referee Comment (RC2) · Anonymous Referee #2 · 28 Aug 2019

The author presents an analysis of the methane global budget using an atmospheric chemistry model over almost 4 decades (1980-2017) and considering changes in both emissions and sinks. As conclusion, they provide likely scenarios of changes in sources and sinks of methane explaining the methane growth different periods of these past decades.

GENERAL COMMENT

The manuscript is well written and organize, and quite easy to follow. The methodology and different steps are well explained. Understanding the methane budget and particularly its changes is crucial to establish pathways for emission reduction. As a result, this study contributes significantly to the debate.

[Figure]

However part of the result section is quite lengthy and may be shortened and more straightforward– see specific comments below. The comparisons to the observation (surface and satellite) as done here, do not really help in discriminating better scenarios.

Regarding the abstract and conclusion of the paper: there might not be a single driver of the recent methane growth, but a combination of different increasing/decreasing sources/sinks (Saunois et al., 2017). Different studies (as cited) may appear in disagreement as only the major driver (one source) is highlighted (in the abstract, title or conclusion) but considering the uncertainty of the estimates, this could be a combination – with high uncertainty on the relative contributions; but this does not mean that there are no agreement between studies. One of the objective is to "investigate the possible driverS" of the methane changes. So why keeping a single source as a driver? Why not considering the combination and trying to estimate the relative contribution? Also, testing different initial emissions would further strengthen the resulting range.

Finally, I would recommend publication in ACP after major revisions addressing the general and specific comments highlighted in this review.

SPECIFIC COMMENTS

Section 1 – Introduction

Lines 38 to 42. Removes those lines: it's stated later and better near the end of the section.

Line 43- 47. Add a general reference here (e.g. Saunois et al. (2016)).

Line 60-67. There have been other studies that tried to highlight some more likely scenarios combining changes from individual sources. E.g., Saunois et al., (2017) suggested a combination (with relative contributions) of changes from the different methane sources contributing to the renewed growth (increasing from fossil, agriculture and waste emissions, decrease of biomass burning, . . .).

Section 2 – Methodology and data

Line 125. "BMB emissions contribute to IAV"... because wetland emissions are considered constant in the study. Wetland emissions contribute a lot to methane emission IAV (Kirschke et al., 2013).

Line 151. "... largely underestimate...". What is largely here? Please quantify.

Line 153. "optimization simulations"? The optimization approach is described later (from line 169), but this description should directly follow the first mentioning of "optimization approach". Which observations are used for this "optimization"? surface marine boundary layer – described after? Please specify and put Section 2.3 Observation before Section Simulation Design.

Line 174. The relative contributions from the different sectors are also kept as in the initial emission set up, and depend on it. As a result this highly depend on the initial set of emissions. Testing other set of initial emissions would really be valuable to assess the uncertainty related to the set up and strengthen the results. EDGARv4.3.2 instead of CEDS for example. Other wetland emissions including IAV...

Line 180. DeltaE include IAV missing from the initial emissions. In the case of SAopt, this is attributed to anthropogenic emissions, is that realistic?

Line 184. The reader would probably want to have fast access to FigS2. . Indeed all the discussion is on emissions from SAopt and SWopt and not on the initial emissions. That would be better to combine Fig1 and Fig S2

Line 200. At which frequency are the observations considered? Frequency of the sampling of the model? Monthly?

Section 3 Results and discussions.

Line 241. Testing different wetland emission data sets – with different seasonality may help to quantify the influence of wetland emission seasonality on the observed bias.

OH influence could be seen in simulation S2 with higher OH: is S2 performing better?

Line 244. Is it reasonable to state that SWopt perform a bit better that SAopt? So far no comparison is made between the two simulations.

Line 253: The reasons of the biases compared to HIPPO are exactly the same as those seen with surface observations. See comment for line 241.

Line 255. 2% difference with the updated GFDL-AM4.1, using optimized emissions. How much was the difference when using initial emissions? How much better is it compared to the previous version of GFDL-AM4.1?

Line 264. Isn't it by construction? As total emissions are optimized using surface marine boundary layer observations. . .

Line 274. Not clear about the 1-year mismatch. There is not always a 1-year delay between spikes. . .

Line 275-281. Are those numbers necessary in the text? What is the point? That could be summarize in a Table for the full numbers and in the text summarize to "SWopt performs better over this period while SAopt performs better over that period". The differences in growth rate are much lower than their own range of uncertainty.

Line 283. Does it imply that the results (following results on the emissions changes) are less robust for this region? This is unfortunate as most of the methane emissions occur in the NH.

Line 286. Indeed. Related to wetland emissions? Is it related to the missing IAV in wetland initial emissions?

Line 301. Could the bias model sampling be overcome/reduced? Does the bias change with the grid box choice for coastal sites?

Line 306-319. The model observation comparison using GOSAT and SCHIAMACHY is not really helpful. As the model is optimized against surface observations, it's saying that surface observation and satellite data see similar trends (but are offset by latitudinal biases), that biases exist in the satellite data (latitudinal biases), and that uncertainties in the transport model (especially in the stratosphere but not only) can explain the difference between model and satellite columns. Nothing new for an atmospheric model assessment, and this comparison does not discriminate one simulation from the other. This part may be removed and put in the supplementary material.

Line 325. By construction?

Line 332-333. This sentence should be removed and put in Section2.

Line335_336. This sentence could be removed with causes of interannual variability put in sentence of line 329-330. Also 1991-1992 high IAV is also related to Mt Pinatubo eruption (decrease in CH4) (ref. e.g., Banda et al., 2016, https://www.atmos-chem-phys.net/16/195/2016/; Dlugokencky, E. J. et al. Changes in CH4 and CO growth rates after the eruption of Mt Pinatubo and their link with changes in tropical tropospheric UV flux. Geophys. Res. Lett. 23, 2761–2764 (1996); 21. Bousquet, P. et al. Contribution of anthropogenic and natural sources to atmospheric methane variability. Nature 443, 439–443 (2006)).

Line 343. Is it possible to overplot Naus et al. OH level on Figure 8?

Table 3. Single values are provided for the emission estimates while several simulations were performed. The authors compare their initial emission to the literature – which could have been done in Section2. However they stated that there initial emission were underestimate and then all the paper is on the optimized emissions. So Table 3 should compare their optimized emissions with the literature!

Line 351-353. Total natural emissions from bottom-up estimates are much lower than top-down because there are not constrained and just an "addition" from independent individual source estimates, knowing the large uncertainty of each natural source... The initial emissions should be much comparable to top-down estimates, as the large

source from freshwater is not included in the initial emissions (about 100Tg in the bottom up methane budget).

Line 353-354. Not really, see comment above. The difference between the initial emission of this study and the bottom-up estimate from Kirschke et al and Saunois et al., is mainly driven by source not included in the initial emission set up (freshwater), and probably double counted in the bottom-up budget. Estimates for other natural sources from Kirschke et al., 2013 and Saunois et al., 2016 should be added. This is also due to the use of a climatological value for wetland emissions from the 2000s applied to the whole period. As IAV and trends are missing in the initial emissions, some signal is lost compared to estimates reported by Kirschke et al. and Saunois et al.

Line 356. Remove Saunois et al., 2016 as the study starts in 2000.

Line 358 and following. Well, that would be better to show the range of total anthropogenic and wetland optimized emissions and to compare these ones with the literature.

Table 3. It also presents values for the more recent period. These values may be compared with the updated global methane budget recently released in ESSDD Saunois et al., 2019 (https://www.earth-syst-sci-data-discuss.net/essd-2019-128/)

Section 3.3.1. These results are quite expected knowing the distribution of anthropogenic emissions and wetland emissions. There are interesting but could be more when compared against observations. If the surface methane DMF and growth rate observed values at each sites are over plotted on these spatial distributions, does it help in discriminating which optimized emissions fit the best?

Line 400-401. By construction?

Line 406. .. are KEPT constant...

Line 432 and line 437. How can we explain higher sink?

Line 453. Indeed. . .Is testing a different initial inventory an option? It would help, confirming or infirming some results. . .

Line 454. Other sectors (than wetland?)

Section3.3.2: could this section be shortened? It's quite hard to follow. . . Instead of describing each simulation in detail (with numbers etc..), would it be shorter to conclude for each period, which sector(s) drive the changes (increase. . .) and if the two simulations agree or not? And keep the details and numbers for the supplementary. . ..

Line 457 and following. The two sensitivity tests may be presented in Section 2 as well. And included in the above discussion..

Section 3.4 Sensitivity to OH. How does the model compare with the other CCMI models? (see Zhao et al., 2019 Fig 4 and 7. https://www.atmos-chem-phys-discuss.net/acp-2019-281/acp-2019-281.pdf)

TECHNICAL COMMENTS

Line 87. 1980-2017 instead of 1980-2014

---

## Author Comment (AC2) · 18 Oct 2019

**We thank the reviewer for the insightful comments and suggestions on our manuscript. Below, our responses in bold text follow the reviewer's comments shown in plain text.**

The author presents an analysis of the methane global budget using an atmospheric chemistry model over almost 4 decades (1980-2017) and considering changes in both emissions and sinks. As conclusion, they provide likely scenarios of changes in sources and sinks of methane explaining the methane growth different periods of these past decades.

GENERAL COMMENT The manuscript is well written and organize, and quite easy

to follow. The methodology and different steps are well explained. Understanding the methane budget and particularly its changes is crucial to establish pathways for emission reduction. As a result, this study contributes significantly to the debate.

**Reply: We thank the reviewer for the positive comments.**

However part of the result section is quite lengthy and may be shortened and more straightforward– see specific comments below. The comparisons to the observation (surface and satellite) as done here, do not really help in discriminating better scenarios.

**Reply: We thank the reviewer for the suggestions. We have revised the manuscript and shortened the result section.**

Regarding the abstract and conclusion of the paper: there might not be a single driver of the recent methane growth, but a combination of different increasing/decreasing sources/sinks (Saunois et al., 2017). Different studies (as cited) may appear in disagreement as only the major driver (one source) is highlighted (in the abstract, title or conclusion) but considering the uncertainty of the estimates, this could be a combination – with high uncertainty on the relative contributions; but this does not mean that there are no agreement between studies. One of the objective is to "investigate the possible driverS" of the methane changes. So why keeping a single source as a driver? Why not considering the combination and trying to estimate the relative contribution? Also, testing different initial emissions would further strengthen the resulting range.

**Reply: We agree with the reviewer that it is more likely a combination of different drivers instead of one single driver that lead to recent methane growth, which is also indicated by the model results discussed in the manuscript. The more important question is the relative contributions from these drivers, which, as mentioned by the reviewer, is highly uncertain. We have included additional discussions in the revised manuscript regarding the relative contributions. However, as mentioned in the manuscript, we need additional observational constraints (e.g.,**

methane isotopes, ethane) to better quantify the relative contributions, which is
the next step of the work. With different initial emissions, the optimized methane
emissions totals would be similar but spatial distribution and seasonality could
be different. We did one sensitivity test with time-varying wetland emissions
(i.e., S0Origtswet) as shown in Figures S2 and S3 in the Supplement. Using
time-varying wetland emissions does not alter our results much. Based on the
results comparison in Figures S2 and S3, which suggest better performance by
climatological wetland emissions (i.e., S0Orig), we decide to start with this wet-
land emission. As this manuscript is more focused on global totals, even with
different initial emissions (e.g., different wetland emissions), after the emission
optimization, the impacts would be very small on global averages, but could
be large on regional scales. Also, the relative contributions would be differ-
ent. However, the Aopt and Wopt sensitivities conducted in this work provide
extreme scenarios for the contributions from anthropogenic sources and wet-
land sources. As mentioned earlier, additional observational constraints could
be applied to refine the resulting ranges.

Finally, I would recommend publication in ACP after major revisions addressing the
general and specific comments highlighted in this review.

**Reply: We thank the reviewer for the suggestions. We have revised the
manuscript and addressed the comments point by point below.**

SPECIFIC COMMENTS Section 1 – Introduction Lines 38 to 42. Removes those lines:
it's stated later and better near the end of the section.

**Reply: We have removed these lines in the revised manuscript.**

Line 43- 47. Add a general reference here (e.g. Saunois et al. (2016)).

**Reply: We have added the reference in the revised manuscript.**

Line 60-67. There have been other studies that tried to highlight some more likely

scenarios combining changes from individual sources. E.g., Saunois et al., (2017) suggested a combination (with relative contributions) of changes from the different methane sources contributing to the renewed growth (increasing from fossil, agriculture and waste emissions, decrease of biomass burning, . . .).

**Reply: We have included the statement of combined changes in sources with the reference in the revised manuscript as below:**

**"The observed renewed growth since 2007 has been explained alternatively through increases in tropical emissions (Houweling et al., 2014; Nisbet et al., 2016) such as agricultural emissions (Schaefer et al., 2016; Patra et al., 2016) and tropical wetland emissions (Bousquet et al., 2011; Maasakkers et al., 2019), increases in fossil fuel emissions (Rice et al., 2016; Worden et al., 2017), decreases in sources compensated by decreases in sinks due to OH levels (Turner et al., 2017; Rigby et al, 2017), or a combination of changes in different sources such as increases in fossil, agriculture, and waste emissions with decreases in biomass burning emissions (Saunois et al., 2017)."**

Section 2 – Methodology and data Line 125. "BMB emissions contribute to IAV". . . because wetland emissions are considered constant in the study. Wetland emissions contribute a lot to methane emission IAV (Kirschke et al., 2013).

**Reply: We have clarified this statement in the revised manuscript as below:**

**"Although wetlands are in reality a major contributor to interannual variability in methane emissions (Bousquet et al., 2006; Kirschke et al., 2013), our use of climatological wetland emissions causes the interannual variability in our methane emissions to be dominated by BMB emissions."**

Line 151. ". . . largely underestimate. . .". What is largely here? Please quantify.

**Reply: We have evaluated the model results with initial emission inventories, which is shown in Figure S2 in the Supplement. The simulated methane DMF is**

**about 126 ppb lower than observed CH4 DMF from NOAA GMD surface observations (with RMSE = 120 ppb).**

Line 153. "optimization simulations"? The optimization approach is described later (from line 169), but this description should directly follow the first mentioning of "optimization approach". Which observations are used for this "optimization"? surface marine boundary layer – described after? Please specify and put Section 2.3 Observation before Section Simulation Design.

**Reply: Thank you for the suggestion. We have reordered the sections such that the section on Observations appears before Simulation Design.**

Line 174. The relative contributions from the different sectors are also kept as in the initial emission set up, and depend on it. As a result this highly depend on the initial set of emissions. Testing other set of initial emissions would really be valuable to assess the uncertainty related to the set up and strengthen the results. EDGARv4.3.2 instead of CEDS for example. Other wetland emissions including IAV. . .

**Reply: We agree that our results depend on the individual contributions in the initial emission inventories. Using different initial emission inventories may help assess the associated uncertainties of individual sector contributions to a certain degree, but to better quantify the individual contributions and assess the uncertainties, we could apply additional observational constraints, which is our next step of the study.**

**Methane anthropogenic emissions from EDGARv4.3.2 are about 5-9% lower than CEDS anthropogenic emissions, mainly due to lower emissions from energy sector (Janssens-Maenhout et al., 2019). The share of ENE to total anthropogenic emissions in EDGARv4.3.2 (i.e., 33%) are also lower than those in CEDS (i.e., 38%), but both increase after 2006. We are unable to test EDGARv4.3.2 emissions in this work but will consider it for future work.**

[Figure]

**To address the reviewer's point about wetland emissions with IAV, we performed a test simulation with wetland emissions including IAV for 2001-2015 (i.e., S0Origtswet) based on an extended ensemble version of WetCharts (Bloom et al., 2017) as shown in Figures S2 and S3. The results from this simulation are very similar to those from climatological wetland emissions (i.e., S0Orig). We expect that optimization of wetland emissions will cause the original IAV in emissions to be lost.**

Line 180. DeltaE include IAV missing from the initial emissions. In the case of SAopt, this is attributed to anthropogenic emissions, is that realistic?

**Reply: The IAV of methane emissions are mainly dominated by that from wetland and biomass burning. However, IAV could also exist in anthropogenic emissions due to the dependence of microbial methane sources, such as rice paddies, on soil temperature and precipitation (e.g., Knox et al., 2016). The optimization to match observations resulted in higher IAV in total emissions than in the initial emissions. Because the purpose of S0Aopt is to investigate the role of changes in total anthropogenic emissions (anthro plus BB) rather than individual sectors, we applied this IAV to all sectors which we acknowledge introduces some unrealistic IAV in the anthropogenic emissions. We chose this experimental construct to limit the number of sensitivity simulations.**

Line 184. The reader would probably want to have fast access to FigS2. . Indeed all the discussion is on emissions from SAopt and SWopt and not on the initial emissions. That would be better to combine Fig1 and Fig S2

**Reply: We have combined these figures as new Fig 1 in the revised manuscript.**

Line 200. At which frequency are the observations considered? Frequency of the sampling of the model? Monthly?

**Reply: Since the frequency of the model output is monthly, the observations are**

[Figure]

**also monthly-based.**

Section 3 Results and discussions. Line 241. Testing different wetland emission data sets – with different seasonality may help to quantify the influence of wetland emission seasonality on the observed bias. OH influence could be seen in simulation S2 with higher OH: is S2 performing better?

**Reply: We agree that different wetland emission datasets can help assess the impacts of wetland emission seasonality on the observed bias. We did a sensitive test with time-varying wetland emissions (i.e., S0Origtswet) for 2001-2015 as shown in Figures S2 and S3 in the Supplement. In terms of seasonality, we did not find much difference between S0Orig and S0Origtswet. In a future study, we plan to simulate wetland emissions in the land model (LM4.1) coupled to AM4.1 to better capture the spatial and temporal variability of wetland emissions than prescribed emissions.**

**The S2 performance is similar to S0 as we re-optimized methane emissions based on higher OH case. Higher OH case does not change the spatial and temporal variability of OH but only the magnitude of OH levels.**

Line 244. Is it reasonable to state that SWopt perform a bit better that SAopt? So far no comparison is made between the two simulations.

**Reply: As we optimize the global total emissions instead of spatial distribution, globally, the performance of Wopt and Aopt is very similar to each other, however there are regional differences because of the differences in the spatial distribution of emissions. For example, in the Southern Hemisphere, Wopt performance is very similar to Aopt. In the Northern Hemisphere, Wopt performs better than Aopt at KUM, POCN20, ASK, and TAP sites, while it performs worse at KEY, WIS, and UTA. It is really site-specific. We have included site-specific comparisons between S0Wopt and S0Aopt in the revised manuscript as below:**

"S0Aopt and S0Wopt simulate very similar surface methane DMF and their comparison with NOAA-GMD observations at individual sites show both simulations to be biased low over Southern Hemisphere sites, but the low bias decreases northward (Figure S5 in the Supplement). The simulations are biased moderately high (with RMSEs up to 40 ppb) over tropical regions (e.g., POCS15, POCS10, SMO, POCS05, POCN00, CHR, and POCN05). These sites are mainly remote sites and surface methane DMF represents background methane levels. The overestimates are likely due to overestimation of emissions over Southeast Asia (e.g., Saeki and Patra, 2017, Patra et al., 2016, and Thompson et al., 2015), which could affect these remote sites through transport. However, the model predicts surface methane DMF relatively well at Ascension Island (ASC, 8oS, 14.4oW, 85 m), which is also a remote site without impacts from East Asia. The high biases over the tropics suggest a need to improve regional emissions (e.g., Southeast Asia). Moderate overestimates also occur at Mahe Island (SEY, 4.7oS, 55.5oE), a location that could be affected by air masses from polluted areas over the tropics and Northern Hemisphere. Over middle and high latitudes of the Northern Hemisphere, both S0Aopt and S0Wopt simulate surface methane DMF relatively well at most sites, except at Key Biscayne (KEY, 25.7oN, 80.2oW), Tae-ahn Peninsula (TAP, 36.7oN, 126.1oW), Park Falls (LEF, 45.9oN, 113.7oW), and Mace Head (MHD, 53.3oN, 9.9oW). KEY, MHD, and TAP are sampled under onshore winds, whereas LEF are affected by local sources and model transport. The high biases at these sites could be due in part to model sampling bias (e.g., model grid box overlapping land while samples are collected at coast with onshore winds) and uncertainties in local emissions (e.g., possible overestimation in the emissions over East Asia). On the other hand, both S0Wopt and S0Aopt are able to capture monthly variations in methane at most of the sites except at LEF, where R = 0.4 for S0Wopt and 0.5 for S0Aopt, respectively. In general, both S0Wopt and S0Aopt are able to reproduce the surface methane DMF and capture the trend at most sites (e.g., with R greater than 0.5 at 98% of total sites and with RMSE less than

**30 ppb at 74% of total sites). As shown in Figure S5, S0Aopt in general better estimates methane trends and growth over low latitudes of the Southern Hemisphere (e.g., SMO) and middle/high latitudes of the Northern Hemisphere (e.g., ASK, KEY, WIS, UTA, NWR, UUA, LEF, CBA, STM, and ALT) than S0Wopt. Based on the site-level comparisons between S0Wopt and S0Aopt, anthropogenic emissions over Southeast Asia are likely overestimated in both S0Aopt and S0Wopt, while they could be underestimated at WLG and NWR in S0Wopt but be reasonably well represented in S0Aopt."**

Line 253: The reasons of the biases compared to HIPPO are exactly the same as those seen with surface observations. See comment for line 241.

**Reply: Please see reply to comment for line 241.**

Line 255. 2% difference with the updated GFDL-AM4.1, using optimized emissions. How much was the difference when using initial emissions? How much better is it compared to the previous version of GFDL-AM4.1?

**Reply: With initial emissions, the simulated methane profiles are about 12% lower than HIPPO measurements. For the standard version of GFDL-AM4.1, which uses prescribed methane concentrations, the simulated methane profiles are about 5% higher than HIPPO measurements in the Southern Hemisphere and 4% lower in the Northern Hemisphere.**

Line 264. Isn't it by construction? As total emissions are optimized using surface marine boundary layer observations. . .

**Reply: The match of global means is certainly by design, but not that for latitude bands. As shown in Figure 4 in the main text, the model is still able to capture methane trends and variability at different latitude bands. We have revised the sentences as below:**

**"S0Wopt and S0Aopt are also able to reproduce global annual mean surface**

**methane DMF (with root-mean-square-error (RMSE) = 10.4 ppb in S0Wopt and 11.6 ppb in S0Aopt) over 1983-2017, which is expected from emission optimization. Meanwhile, both simulations are able to reproduce the methane timeseries very well (with R = 1.0 in both S0Wopt and S0Aopt) over different latitude bands as shown in Figure 4."**

Line 274. Not clear about the 1-year mismatch. There is not always a 1-year delay between spikes. . .

**Reply: We corrected a bug in the scripts for model evaluation and updated all the relevant plots. We then updated results and discussions in the revised manuscript as below:**

**"Table 3 summarizes methane growth rates during 1984-1991, 1992-1998, 1999-2006, and 2007-2017. S0Aopt and S0Wopt simulate very similar methane growth rates as their emission totals are the same. During 1984-1991, both S0Aopt and S0Wopt slightly overestimate methane growth rates by 2 ppb yr-1, possibly due to fewer available observations used for emission optimization during this time period than afterwards. After 1991, the simulated methane growth rates are in general comparable to the observations (with annual mean difference within ±1 ppb yr-1). The major discrepancies in the simulated methane growth rates and observations occur over the tropics and high northern latitudes as shown in Figure 4. Over the tropics, both S0Aopt and S0Wopt overestimate methane growth rates (by about 5-10 ppb yr-1) during 1984-1990 when there were limited observations available, but are able to reproduce methane growth rates relatively well afterwards. Agreement of the methane growth rate is worse in the Northern Hemisphere than in the Southern Hemisphere, especially at high northern latitudes, mainly due to the large bias during 1984-1988 and a slight shift in peak growth (or peak decrease) during 1997-2005. The number of observational MBL sites does not provide adequate coverage of the globe, especially in the 1980s, which could have different impacts on the Northern and Southern Hemisphere**

when optimizing global total emissions. In general, S0Aopt estimates slightly better methane growth rates than S0Wopt, especially over 30-90o N. The biases in methane growth rates also suggest a need to refine regional emissions."

Line 275-281. Are those numbers necessary in the text? What is the point? That could be summarize in a Table for the full numbers and in the text summarize to "SWopt performs better over this period while SAopt performs better over that period". The differences in growth rate are much lower than their own range of uncertainty.

**Reply: As suggested by the reviewer, we have summarized the growth rates in the table and revised these sentences in the revised manuscript (see reply to comment line 274).**

Line 283. Does it imply that the results (following results on the emissions changes) are less robust for this region? This is unfortunate as most of the methane emissions occur in the NH.

**Reply: We corrected a bug in the scripts for model evaluation and updated all the relevant plots. After the correction, both Aopt and Wopt are able to reproduce methane growth rates despite there is a slight mismatch ( 1-2 years) during 1998-2008. But compared to other regions, the correlation is slightly worse over 30-90o N. This suggests a need to improve/optimize regional emissions.**

Line 286. Indeed. Related to wetland emissions? Is it related to the missing IAV in wetland initial emissions?

**Reply: Many reasons could lead to such biases over 30-90N. Uncertainties in the IAV of both wetland and anthropogenic emissions could lead to such bias. The biases also suggest a need to improve/optimize regional emissions.**

Line 301. Could the bias model sampling be overcome/reduced? Does the bias change with the grid box choice for coastal sites?

**Reply: Yes. If we move the grid box to the ocean side, the bias is reduced sig-**

**nificantly. For example, at KEY and MHD sites, the RMSEs are reduced from 90 ppb to 33 ppb and from 50 ppb to 20 ppb, respectively.**

Line 306-319. The model observation comparison using GOSAT and SCHIAMACHY is not really helpful. As the model is optimized against surface observations, it's saying that surface observation and satellite data see similar trends (but are offset by latitudinal biases), that biases exist in the satellite data (latitudinal biases), and that uncertainties in the transport model (especially in the stratosphere but not only) can explain the difference between model and satellite columns. Nothing new for an atmospheric model assessment, and this comparison does not discriminate one simulation from the other. This part may be removed and put in the supplementary material.

**Reply: As suggested by the reviewer, we have moved this part to the Supplement.**

Line 325. By construction?

**Reply: Yes. Both simulations reproduce the global growth rates by design. But the model with optimized emissions is still able to reproduce the growth rates at different latitude bands.**

Line 332-333. This sentence should be removed and put in Section2.

**Reply: As suggested by the reviewer, we have moved this sentence to Section2.**

Line335-336. This sentence could be removed with causes of interannual variability put in sentence of line 329-330.

**Reply: As suggested by the reviewer, we have combined these sentences as below:**

**"As shown in Figure 5, the optimized emissions in general increase during 1980-2017, with an annual mean of 580±34 Tg yr-1 (mean±standard deviation) and show much larger interannual variability during 1991-1993 and 1997-2000, which**

**is likely due to the strong El Niño events during 1991-1992 and 1997-1998 as well as the Mt Pinatubo eruption in 1991 (Dlugokencky et al., 1996; Bousquet et al., 2006; Bândă et al., 2016)."**

Also 1991-1992 high IAV is also related to Mt Pinatubo eruption (decrease in CH4) (ref. e.g., Banda et al., 2016, https://www.atmos-chemphys.net/16/195/2016/; Dlugokencky, E. J. et al. Changes in CH4 and CO growth rates after the eruption of Mt Pinatubo and their link with changes in tropical tropospheric UV flux. Geophys. Res. Lett. 23, 2761–2764 (1996); 21. Bousquet, P. et al. Contribution of anthropogenic and natural sources to atmospheric methane variability. Nature 443, 439–443 (2006)).

**Reply: We thank the reviewer for providing the references. We have included these references into the revised manuscript.**

Line 343. Is it possible to overplot Naus et al. OH level on Figure 8?

**Reply: We have included OH anomaly from Naus et al. (2019) in the Figure 6 in the revised manuscript.**

Table 3. Single values are provided for the emission estimates while several simulations were performed. The authors compare their initial emission to the literature – which could have been done in Section2. However they stated that there initial emission were underestimate and then all the paper is on the optimized emissions. So Table 3 should compare their optimized emissions with the literature!

**Reply: Table 3 includes numbers for optimized emissions as reflected in rows of and sum of sources. Also, for and sum of sources, we include estimated ranges based on different OH levels as well. To address the reviewer's comment, we have included estimated ranges for individual sectors based on the Aopt and Wopt under different OH levels, which is now shown in Table 4 in the revised manuscript.**

Line 351-353. Total natural emissions from bottom-up estimates are much lower than

top-down because there are not constrained and just an "addition" from independent individual source estimates, knowing the large uncertainty of each natural source. . . The initial emissions should be much comparable to top-down estimates, as the large source from freshwater is not included in the initial emissions (about 100Tg in the bottom up methane budget).

**Reply: We thank the reviewer for the explanation. We have included this discussion in the revised manuscript as below:**

**"Since there is no observational constraint on bottom-up estimates, total natural emissions are simply summed over independent individual sources, which could be overestimated in the bottom-up approach considering the relatively large uncertainties in each individual source. In addition, in the bottom-up estimate from Kirschke et al. (2013) and Saunois et al. (2016), some other natural sources, such as freshwater, are not included in the initial emission inventories in this work; however, they are likely double counted in the bottom-up estimates (e.g., high-latitude inland waters are likely also considered as wetland areas) as pointed out in Saunois et al. (2019)."**

Line 353-354. Not really, see comment above. The difference between the initial emission of this study and the bottom-up estimate from Kirschke et al and Saunois et al., is mainly driven by source not included in the initial emission set up (freshwater), and probably double counted in the bottom-up budget. Estimates for other natural sources from Kirschke et al., 2013 and Saunois et al., 2016 should be added. This is also due to the use of a climatological value for wetland emissions from the 2000s applied to the whole period. As IAV and trends are missing in the initial emissions, some signal is lost compared to estimates reported by Kirschke et al. and Saunois et al.

**Reply: We thank the reviewer for the explanation. We have included this discussion in the revised manuscript (see reply above) .**

Line 356. Remove Saunois et al., 2016 as the study starts in 2000.

**Reply: Removed.**

Line 358 and following. Well, that would be better to show the range of total anthropogenic and wetland optimized emissions and to compare these ones with the literature. Table 3. It also presents values for the more recent period. These values may be compared with the updated global methane budget recently released in ESSDD Saunois et al., 2019 (https://www.earth-syst-sci-data-discuss.net/essd-2019-128/)

**Reply: We have updated Table 3 (now Table 4) with values from Saunois et al. (2019).**

Section 3.3.1. These results are quite expected knowing the distribution of anthropogenic emissions and wetland emissions. There are interesting but could be more when compared against observations. If the surface methane DMF and growth rate observed values at each sites are over plotted on these spatial distributions, does it help in discriminating which optimized emissions fit the best?

**Reply: The overlay plots have been generated as shown in Figure S7 and Figure S9 in the Supplement. But S0Aopt and S0Wopt gives very similar methane DMF and growth rates. We included site-level comparisons in Section 3.1. As also suggested by the other reviewer, we have removed this part into the Supplement.**

Line 400-401. By construction?

**Reply: For global trends, the match of model simulations with observations are by design.**

Line 406. .. are KEPT constant. . .

**Reply: We have corrected this in the revised manuscript as below:**

**"Since wetland emissions and other natural emissions are kept constant every year in S0Aopt, with increases in OH levels during 1983-1998, all tagged natural tracers show a weak decreasing trend."**

[Figure]

Line 432 and line 437. How can we explain higher sink?

**Reply: Although concentrations of tracer CH4WET decrease, with the increases in OH levels during 1999-2006, CH4WET sinks also increase, which could be higher than wetland emissions. Also, during this time period, CH4WET decreasing trend is very week (e.g., -0.6 ppb/yr). During 2007-2017, wetland emissions are lower compared to 1999-2006 (see Figure 1) while OH levels are higher. Therefore, the decreasing trend (e.g., -4.6 ppb/yr) is much larger compared to 1999-2006.**

Line 453. Indeed. . .Is testing a different initial inventory an option? It would help, confirming or infirming some results. . .

**Reply: See reply to General comments above.**

Line 454. Other sectors (than wetland?)

**Reply: Yes. For S0Wopt, except wetland, which is optimized, all other sources are based on the initial emission inventories.**

Section3.3.2: could this section be shortened? It's quite hard to follow. . . Instead of describing each simulation in detail (with numbers etc..), would it be shorter to conclude for each period, which sector(s) drive the changes (increase. . .) and if the two simulations agree or not? And keep the details and numbers for the supplementary. . ..

**Reply: We thank the reviewer for the suggestion. We have shortened and revised this section as Section 3.3 in the revised manuscript.**

Line 457 and following. The two sensitivity tests may be presented in Section 2 as well. And included in the above discussion..

**Reply: As suggested by the reviewer, we have included two sensitivity tests in Section 2 as well as in Section3.3.**

Section 3.4 Sensitivity to OH. How does the model compare with the other CCMI models? (see Zhao et al., 2019 Fig 4 and 7. https://www.atmos-chem-physdiscuss.net/acp-2019-281/acp-2019-281.pdf)

**Reply: In general, our OH trend is within the range of OH trends in Zhao et al. (2019). From 1980 to 2000, OH in AM4.1 increases by 4.7%, comparable to 4.6$\pm$2.4% in Zhao et al. (2019). During 2000-2010, OH anomaly varies from -0.29 mole cm-3 to 0.34 molec cm-3.**

TECHNICAL COMMENTS Line 87. 1980-2017 instead of 1980-2014

**Reply: Thanks, corrected.**

**References: Bloom, A. A., Bowman, K. W., Lee, M., Turner, A. J., Schroeder, R., Worden, J. R., Weidner, R., McDonald, K. C., and Jacob, D. J.: A global wetland methane emissions and uncertainty dataset for atmospheric chemical transport models (WetCHARTs version 1.0), Geosci. Model Dev., 10, 2141-2156, https://doi.org/10.5194/gmd-10-2141-2017, 2017.**

**Knox, S. H., Matthes, J. H., Sturtevant, C., Oikawa, P. Y., Verfaillie, J., and Baldocchi, D.: Biophysical controls on interannual variability in ecosystem-scale CO2 and CH4 exchange in a California rice paddy, J. Geophys. Res. Biogeosci., 121, 978–1001, doi:10.1002/2015JG003247, 2016.**

**Janssens-Maenhout, G., Crippa, M., Guizzardi, D., Muntean, M., Schaaf, E., Dentener, F., Bergamaschi, P., Pagliari, V., Olivier, J. G. J., Peters, J. A. H. W., van Aardenne, J. A., Monni, S., Doering, U., Petrescu, A. M. R., Solazzo, E., and Oreggioni, G. D.: EDGAR v4.3.2 Global Atlas of the three major greenhouse gas emissions for the period 1970–2012, Earth Syst. Sci. Data, 11, 959–1002, https://doi.org/10.5194/essd-11-959-2019, 2019.**

Please also note the supplement to this comment:

[Figure]

https://www.atmos-chem-phys-discuss.net/acp-2019-529/acp-2019-529-AC2-supplement.pdf

[Figure]

**Supplement:**

**S1. Observational data**

**Table S1. List of NOAA-GMD marine boundary layer (MBL) sites**

| Code | Name | State | Country | Latitude | Longitude | Elevation (m) |
|---|---|---|---|---|---|---|
| ALT | Alert | Nunavut | Canada | 82.451 | -62.507 | 190 |
| AMS | Amsterdam Island | N/A | France | -37.798 | 77.538 | 55 |
| ASC | Ascension Island | N/A | United Kingdom | -7.967 | -14.4 | 85 |
| AVI | St. Croix | Virgin Islands | United States | 17.75 | -64.75 | 3 |
| BME | St. Davids Head | Bermuda | United Kingdom | 32.368 | -64.648 | 12 |
| BMW | Tudor Hill | Bermuda | United Kingdom | 32.265 | -64.879 | 30 |
| BRW | Barrow | Alaska | United States | 71.323 | -156.611 | 11 |
| CBA | Cold Bay | Alaska | United States | 55.21 | -162.72 | 21.3 |
| CGO | Cape Grim | Tasmania | Australia | -40.683 | 144.69 | 94 |
| CHR | Christmas Island | N/A | Republic of Kiribati | 1.7 | -157.152 | 0 |
| CMO | Cape Meares | Oregon | United States | 45.478 | -123.969 | 30 |
| CRZ | Crozet Island | N/A | France | -46.434 | 51.848 | 197 |
| EIC | Easter Island | N/A | Chile | -27.16 | -109.428 | 47 |
| GMI | Mariana Islands | N/A | Guam | 13.386 | 144.656 | 0 |
| HBA | Halley Station | Antarctica | United Kingdom | -75.605 | -26.21 | 30 |
| ICE | Storhofdi | Vestmannaeyjar | Iceland | 63.4 | -20.288 | 118 |
| KEY | Key Biscayne | Florida | United States | 25.665 | -80.158 | 1 |
| KUM | Cape Kumukahi | Hawaii | United States | 19.52 | -154.82 | 3 |
| MBC | Mould Bay | Northwest Territories | Canada | 76.247 | -119.353 | 30 |
| MHD | Mace Head | County Galway | Ireland | 53.326 | -9.899 | 5 |
| MID | Sand Island | Midway | United States | 28.21 | -177.38 | 11 |
| POC000* | Pacific Ocean (0 N) | N/A | N/A | 0 | -155 | 10 |
| POCN05* | Pacific Ocean (5 N) | N/A | N/A | 5 | -151 | 10 |
| POCN10* | Pacific Ocean (10 N) | N/A | N/A | 10 | -149 | 10 |
| POCN15* | Pacific Ocean (15 N) | N/A | N/A | 15 | -145 | 10 |
| POCN20* | Pacific Ocean (20 N) | N/A | N/A | 20 | -141 | 10 |
| POCN25* | Pacific Ocean (25 N) | N/A | N/A | 25 | -139 | 10 |
| POCN30* | Pacific Ocean (30 N) | N/A | N/A | 30 | -135 | 10 |
| POCS05* | Pacific Ocean (5 S) | N/A | N/A | -5 | -159 | 10 |

| | | | | | | |
|---|---|---|---|---|---|---|
| POCS10* | Pacific Ocean (10 S) | N/A | N/A | -10 | -161 | 10 |
| POCS15* | Pacific Ocean (15 S) | N/A | N/A | -15 | -164 | 10 |
| POCS20* | Pacific Ocean (20 S) | N/A | N/A | -20 | -167 | 10 |
| POCS25* | Pacific Ocean (25 S) | N/A | N/A | -25 | -171 | 10 |
| POCS30* | Pacific Ocean (30 S) | N/A | N/A | -30 | -176 | 10 |
| POCS35* | Pacific Ocean (35 S) | N/A | N/A | -35 | 180 | 10 |
| PSA | Palmer Station | Antarctica | United States | -64.92 | -64 | 10 |
| RPB | Ragged Point | N/A | Barbados | 13.165 | -59.432 | 15 |
| SHM | Shemya Island | Alaska | United States | 52.711 | 174.126 | 23 |
| SMO | Tutuila | N/A | American Samoa | -14.247 | -170.564 | 42 |
| SPO | South Pole | Antarctica | United States | -89.98 | -24.8 | 2810 |
| STM | Ocean Station M | N/A | Norway | 66 | 2 | 0 |
| SYO | Syowa Station | Antarctica | Japan | -69.013 | 39.59 | 14 |
| USH | Ushuaia | N/A | Argentina | -54.848 | -68.311 | 12 |
| ZEP | Ny-Alesund | Svalbard | Norway and Sweden | 78.907 | 11.888 | 474 |

* Latitude values given for the POCN and POCS sites are the centers of latitude bands of +/- 2.5 degrees, and observations can fall anywhere within those bands.

**Table S2. List of NOAA-GMD sites with at least 20-year observations**

| Code | Name | State | Country | Latitude | Longitude | Elevation (meters) |
|------|------|-------|---------|----------|-----------|--------------------|
| ALT | Alert | Nunavut | Canada | 82.451 | -62.507 | 190 |
| ASC | Ascension Island | N/A | United Kingdom | -7.967 | -14.4 | 85 |
| ASK | Assekrem | N/A | Algeria | 23.262 | 5.632 | 2710 |
| AZR | Terceira Island | Azores | Portugal | 38.766 | -27.375 | 19 |
| BMW | Tudor Hill | Bermuda | United Kingdom | 32.265 | -64.879 | 30 |
| BRW | Barrow | Alaska | United States | 71.323 | -156.611 | 11 |
| CBA | Cold Bay | Alaska | United States | 55.21 | -162.72 | 21.3 |
| CGO | Cape Grim | Tasmania | Australia | -40.683 | 144.69 | 94 |
| CHR | Christmas Island | N/A | Republic of Kiribati | 1.7 | -157.152 | 0 |
| CRZ | Crozet Island | N/A | France | -46.434 | 51.848 | 197 |
| EIC | Easter Island | N/A | Chile | -27.16 | -109.428 | 47 |
| GMI | Mariana Islands | N/A | Guam | 13.386 | 144.656 | 0 |
| HBA | Halley Station | Antarctica | United Kingdom | -75.605 | -26.21 | 30 |
| HUN | Hegyhatsal | N/A | Hungary | 46.95 | 16.65 | 248 |
| ICE | Storhofdi | Vestmannaeyjar | Iceland | 63.4 | -20.288 | 118 |
| IZO | Izana | Tenerife | Spain | 28.309 | -16.499 | 2372.9 |
| KEY | Key Biscayne | Florida | United States | 25.665 | -80.158 | 1 |
| KUM | Cape Kumukahi | Hawaii | United States | 19.52 | -154.82 | 3 |
| LEF | Park Falls | Wisconsin | United States | 45.945 | -90.273 | 472 |
| MHD | Mace Head | County Galway | Ireland | 53.326 | -9.899 | 5 |
| MID | Sand Island | Midway | United States | 28.21 | -177.38 | 11 |
| MLO | Mauna Loa | Hawaii | United States | 19.536 | -155.576 | 3397 |
| NWR | Niwot Ridge | Colorado | United States | 40.053 | -105.586 | 3523 |
| POCN00* | Pacific Ocean (0 N) | N/A | N/A | 0 | -155 | 10 |
| POCN05* | Pacific Ocean (5 N) | N/A | N/A | 5 | -151 | 10 |
| POCN10* | Pacific Ocean (10 N) | N/A | N/A | 10 | -149 | 10 |
| POCN15* | Pacific Ocean (15 N) | N/A | N/A | 15 | -145 | 10 |
| POCN20* | Pacific Ocean (20 N) | N/A | N/A | 20 | -141 | 10 |
| POCN25* | Pacific Ocean (25 N) | N/A | N/A | 25 | -139 | 10 |
| POCN30* | Pacific Ocean (30 N) | N/A | N/A | 30 | -135 | 10 |

| | | | | | | |
|---|---|---|---|---|---|---|
| POCS05[*] | Pacific Ocean (5 S) | N/A | N/A | -5 | -159 | 10 |
| POCS10[*] | Pacific Ocean (10 S) | N/A | N/A | -10 | -161 | 10 |
| POCS15[*] | Pacific Ocean (15 S) | N/A | N/A | -15 | -164 | 10 |
| POCS20[*] | Pacific Ocean (20 S) | N/A | N/A | -20 | -167 | 10 |
| POCS25[*] | Pacific Ocean (25 S) | N/A | N/A | -25 | -171 | 10 |
| POCS30[*] | Pacific Ocean (30 S) | N/A | N/A | -30 | -176 | 10 |
| PSA | Palmer Station | Antarctica | United States | -64.92 | -64 | 10 |
| RPB | Ragged Point | N/A | Barbados | 13.165 | -59.432 | 15 |
| SEY | Mahe Island | N/A | Seychelles | -4.682 | 55.532 | 2 |
| SHM | Shemya Island | Alaska | United States | 52.711 | 174.126 | 23 |
| SMO | Tutuila | N/A | American Samoa | -14.247 | -170.564 | 42 |
| SPO | South Pole | Antarctica | United States | -89.98 | -24.8 | 2810 |
| STM | Ocean Station M | N/A | Norway | 66 | 2 | 0 |
| SYO | Syowa Station | Antarctica | Japan | -69.013 | 39.59 | 14 |
| TAP | Tae-ahn Peninsula | N/A | Republic of Korea | 36.738 | 126.133 | 16 |
| USH | Ushuaia | N/A | Argentina | -54.848 | -68.311 | 12 |
| UTA | Wendover | Utah | United States | 39.902 | -113.718 | 1327 |
| UUM | Ulaan Uul | N/A | Mongolia | 44.452 | 111.096 | 1007 |
| WIS | Weizmann Institute of Science at the Arava Institute | Ketura | Israel | 29.965 | 35.06 | 151 |
| WLG | Mt. Waliguan | N/A | Peoples Republic of China | 36.288 | 100.896 | 3810 |
| ZEP | Ny-Alesund | Svalbard | Norway and Sweden | 78.907 | 11.888 | 474 |

[*] Latitude values given for the POCN and POCS sites are the centers of latitude bands of +/- 2.5 degrees, and observations can fall anywhere within those bands.

[Figure]

**Figure S1. NOAA-GMD marine boundary layer (MBL) sites selected for background methane calculation.**

**S2. Evaluation of model simulations with initial emission inventories**

We conducted two model simulations with the initial methane emissions inventories for 1980-2017: 1) the initial emissions described in Section 2.1 in the main text (referred to as "S0Orig"); 2) same as S0Orig but with time-varying wetland emissions based on an extended ensemble version of WetCHARTs for 2001-2015 (Bloom et al., 2017), which is referred to as "S0Origtswet". The model evaluation of the two simulations are shown in Figure S2 and S3.

[Figure]

**Figure S2. Comparison of GFDL-AM4.1 simulated methane concentrations and growth rates with NOAA-GMD surface observations with initial emission inventories. For the upper plot in each panel, dash line represents smoothed trends (i.e., 12-month running mean) from deseasonalized monthly data. A meridional curve (Tans et al., 1989) was fitted through NOAA-GMD site observations to get the latitudinal distribution of methane. A function fit consisting of yearly harmonics and a polynomial trend, with fast fourier transform and low pass filtering of the residuals are applied to the monthly mean methane DMF (Thoning et al., 1989; Thoning, 2019) to approximate the long-term trend. For the lower plot in each panel, the growth rates are calculated from the time derivative of the dash line in the corresponding upper plot.**

[Figure]

**Figure S3. Comparison of GFDL-AM4.1 simulated methane seasonal cycles of 2001-2015 with NOAA-GMD surface observations with initial emission inventories.**

**S3. Surface evaluation at individual sites**

[Figure]

[Figure]

[Figure]

[Figure]

[Figure]

**Figure S4. Comparisons of methane seasonal cycles against NOAA-GMD observations.**

[Figure]

[Figure]

[Figure]

[Figure]

[Figure]

**Figure S5. Comparisons of surface CH₄ dry-mole fractions and growth rates to NOAA-GMD observations.**

**S4. Satellite evaluation**

[revised manuscript text omitted]

---

## Author Response (AR1)

Response to Referee #1 for comments on "Investigation of the global methane budget over 1980-2017 using GFDL-AM4.1" by He et al.

We thank the reviewer for the insightful comments and suggestions on our manuscript. Below, our responses in bold text follow the reviewer's comments shown in plain text.

5

10

General comments:

The work describes CH4 budget using an atmospheric chemistry-transport model that is developed at the GFDL. The authors have taken in to account all possible causes of variabilities in CH4 budget, such as the emissions and loss due to tropospheric hydroxyl (OH). As shown in the manuscript, OH variability is of as much importance as the emissions in explaining the CH4 growth rate variabilities in different decades in the period of 1980s to 2010s. The manuscript is generally well written. However, I felt toward the end of the manuscript is a bit of stretch and could be reduced (I have made some suggestions in my specific comments). The manuscript can be accepted after a major revision.

**Reply:**

We thank the reviewer for the positive comments. As suggested by the reviewer, we have shortened Section 3 and revised the manuscript carefully. Below are our point-by-point responses.

Specific comments:

Line 49 & 62ff: can the growth rate discussions in the introduction be made concise and put together at one place.

**Reply:**

20 As suggested by the reviewer, we have removed the sentence highlighting past studies on drivers of methane trend and variability avoid redundancy. However, we keep the sentence on the observed changes in growth rate to motivate the study.

Line 80ff: I think there are other prominent inverse modelling results trying to explain the recent regrowth of CH4 concentrations.

**Reply:**

We have revised this sentence and included additional references in the revised manuscript as below:

"The observed renewed growth since 2007 has been explained alternatively through increases in tropical emissions
(Houweling et al., 2014; Nisbet et al., 2016) such as agricultural emissions (Schaefer et al., 2016; Patra et al., 2016) and tropical wetland emissions (Bousquet et al., 2011; Maasakkers et al., 2019), increases in fossil fuel emissions (Rice et al., 2016; Worden et al., 2017), decreases in sources compensated by decreases in sinks due to OH levels (Turner et al., 2017; Rigby et al, 2017), or a combination of changes in different sources such as increases in fossil, agriculture, and waste emissions with decreases in biomass burning emissions (Saunois et al., 2017)."

35

25

Line 135-137: This is a quite strange statement. After reading the whole manuscript I do not believe you have tried to address these couple of issues to a great extent. May be remove?

**Reply:**

We have removed these sentences as suggested by the reviewer.

40

Line 156: Not from wetchart? I mean does wetchart not have IAV?

**Reply:**

We use WETCHARTs version 1.0 (Bloom et al., 2017) for wetland emissions. The seasonality and spatial distributions are based on ensemble mean. We then repeated the emissions every year. There is also an extend

- 45 ensemble version of WETCHARTs with interannual variability of wetland emissions only for 2001-2015. We did a sensitivity test with this version to compare with climatological wetland emissions. The results are shown in Figures S2 and S3 in the Supplement. We find a better model performance with climatological wetland emissions. Besides, with the optimization on wetland emissions, the signal of the initial interannual variability of wetland emissions would be lost anyway. Therefore, we keep this version (i.e., climatological wetland emissions) as the starting point for wetland emission on timization
- 50 wetland emission optimization.

Line 206: Not clear if this is after LNOx scaling? please make this statement precise (e.g., Control).

**Reply:**

55

60

This magnitude is for standard AM4.1 without scaling. We have revised the sentences in the revised manuscript to make it clearer as below.

"The climatological global mean LNOx emission simulated by standard AM4.1 is about 3.6 TgN yr-1, within the range of 2-8 TgN yr-1 estimated by previous studies (e.g., Schumann and Huntrieser, 2007). We additionally apply scaling factors (e.g., 0.5 and 2.0) to LNOx emissions, producing LNOx at the lower and upper limits of the estimated range for sensitivity simulations described below."

Line 249ff: "The meridional curve" needed some clarifications here, e.g., selected sites within a latitude band to get the mean CH4 at 5 different latitude bands or something like that.

**Reply:**

65 We have included detailed description in Section 2.2 in the revised manuscript as below:

"The global estimates are based on spatial and temporal smoothing of CH4 measurements from 45 surface marine boundary layer (MBL) sites. Locations of MBL sites are shown in Figure S1, and information for each MBL site is listed in Table S1 in the Supplement. First, the average trend and seasonal cycle are approximated for each sampling site by

- 70 fitting a second-order polynomial and four harmonics to the data. We characterize deviations from this average behaviour by transforming the residuals to frequency domain, then multiplying by a low pass filter (Thoning et al., 1989; Thoning, 2019). Zonal and global averages are determined by extracting values at synchronized times steps from the smoothed fits to the data, then fitting another curve as a function of latitude (Tans et al., 1989). We divide these fits into sine (latitude) = 0.05 intervals, which define a matrix of zonally averaged CH4 as a function of time and latitude."
- 75 Line 296: Sometimes the sites like Key Biscayne are sampled by moving the model grids to the ocean side. You might check that out.

**Reply:**

We thank the reviewer for the suggestion. Moving the model grid to the ocean side reduces the model bias. Similar issue also exists at Mace Head site.

80

Line 315ff: The tropical bias in all HIPPO is a bit strange! Not OH but transport (or emissions)? I am suspecting this because the bias due to OH would appear at all altitudes (timescale  $\sim$ 1yr), because the bias is in the lower troposphere, if the vertical transport is slow, you would find more CH4 is accumulated in the lower troposphere (timescale~week)

**Reply:**

- 85 We agree with the reviewer that the bias could be due to transport or emissions, which we already mentioned in the manuscript at lines 251-253. Also, OH is much higher over tropics than higher latitudes. It is possible that OH over tropics are overestimated in the model. Unfortunately, without observations, we are not able to rule out any possibilities.
- 90 Line 326: do you run CH3CCl3 & SF6?, say within the CCMI framework?

**Reply:**

**We do not run with CH3CCl3 & SF6.**

Line 346ff: suggesting too much emissions in the NH, where most of Anthro emissions are...May be you can test this better 95 by site-level comparisons.

**Reply:**

We have updated all the plots in the revised manuscript due to a bug fix in the scripts for model evaluation. As shown in Figure 4 in the main text, the model is able to capture methane trend very well over high latitudes, with R = 1.0 and RMSE < 10 ppb in the Northern Hemisphere and RMSE ~ 11 ppb in the Southern Hemisphere. For site-level comparisons, which are shown in Figures S4-5, over Northern Hemisphere, the model is able to reproduce methane DMFs at most of sites with RMSE < 25 ppb and R >0.9 although there are some overestimations at certain sites possibly due to overestimations in the local sources. Site-level comparisons are presented at lines 298-319 in the revised manuscript.

105 Line 353ff: I cannot find this 1 year mismatch (please be clear), instead I find a persistent offset during 1984-1991 (how the major and minor ticks marked in Fig. 5; the labeled ticks only should be major?

**Reply:**

We have updated all the plots in the revised manuscript due to a bug fix in the scripts for model evaluation. As shown in Figure 4, the model fails to reproduce methane growth rates during 1984-1991, especially over tropics. This could be due in part to the fewer available observations used for emission optimization during this time period. The plots of

be due in part to the fewer available observations used for emission optimization during this time period. The plots of methane growth rates are also updated with same major and minor ticks as shown for methane trend plots in Figure 4.

Line 369: How can you say that? I thought your optimization was not good for this period, because the number of observation sites may not have covered the global reasonably well. I mean biased high toward the NH. Could you check how many SH sites you have data before 1988.

**Reply:**

We agree with the reviewer that we have fewer observations available before 1988 used for emission optimization. Over Southern Hemisphere, there are only six sites (i.e., SPO, HBA, PSA, CGO, ASC, and SMO) as shown in Figure

120 S5 that at least have one-year data available before 1988, which are much fewer than the number of sites over Northern Hemisphere. Line 374: Most likely due to an overestimation of China emissions (e.g., Saeki and Patra, GOSL, 2017, and references therein) (regional inversion is needed for adjusting such regional emission biases)

**125 Reply:**

We thank the reviewer for the references. We have included them in the revised manuscript as below:

"The overestimates are likely due to overestimation of emissions over Southeast Asia (e.g., Saeki and Patra, 2017, Patra et al., 2016, and Thompson et al., 2015), which could affect these remote sites through transport."

130 Line 378: "...which is also a remote site" and remote from China emissions

**Reply:**

We agree with the reviewer's comment. This again suggests an overestimation in the emissions over Southeast Asia. We have included this in the revised manuscript as below:

135 "However, the model predicts surface methane DMF relatively well at Ascension Island (ASC, 8°S, 14.4°W, 85 m), which is also a remote site without impacts from East Asia."

Lines 394ff: I am not very sure if the comparison with GOSAT/SCIA are adding any values to this work. Better be kept aside for a full paper, unless the reasons for the mean offsets are figured out and discussed. For instance you could compare your results with the ACE-FTS data to find out if there is any bias in the stratospheric CH4 as there is no significant offsets in the tropospheric CH4 is seen in comparison with surface data and HIPPO.

**Reply:**

140

165

**We have moved the model evaluation against satellite observations to the Supplement.**

145 Line 426ff: The emission increase in the 1990s is apparently linked to OH increase in AM; which sector can provide this extra emissions. I think this result is very different from what I have seen in the literature, and thus needing some explanation. Surprisingly, the emission increase rate in the 1990s is greater than the recent regrowth period.

**Reply:**

In S0Wopt, the total emission growth during the 1980s is mainly from emission growth from agriculture (0.7 Tg yr-1), energy (0.3 Tg yr-1), waste (1.0 Tg yr-1), and wetland (1.8 Tg yr-1), while the total emission growth during the 1990s is mainly from waste (0.8 Tg yr-1) and wetland (3.7 Tg yr-1). Therefore, these extra emissions in the 1990s are mainly from wetland in S0Wopt scenario.

In S0Aopt, the total emission growth during the 1980s is mainly from emission growth from agriculture (1.4 Tg yr-1), energy (0.9 Tg yr-1), and waste (1.4 Tg yr-1), while the total emission growth during the 1990s is mainly from

- 155 agriculture (1.3 Tg yr-1), energy (1.1 Tg yr-1), waste (1.6 Tg yr-1), and biomass burning (0.4 Tg yr-1). Therefore, these extra emissions in the 1990s are mainly from energy, waste, and biomass burning sectors in S0Aopt scenario. The emission increase rate in the 1990s is greater than the recent growth period because OH decreases in the recent growth period. When we optimize emissions, we consider the impacts of OH trends. In the recent growth period, OH is decreasing, and therefore methane lifetime increases. This amplifies the responses of methane concentrations to the
- 160 changes in the emissions. Therefore, a smaller increase in the emissions during this period can lead to larger increases in methane concentrations compared to that in the 1990s.

Lines 484ff: The discussions using Fig 9-11 aren't that interesting as presented. I would recommend the authors to move these plots to the supplement or show 1-2 panels in the main text; for example all the 4 panels in Fig 9 & 10 are essentially showing very similar distributions. The S0Aopt and S0Wopt are also showing similar behaviour. This is mainly because the

4

emission (E)-a priori emissions are the same in both the simulations, and the correction emissions Del-E following Anthropogenic or Wetland emission patterns only play minor role. I am actually curious if you could use some of the continental sites, e.g., NWR, LEF, SGP or TAP, and use the model-measurement comparisons to say whether the S0Aopt or S0Wopt are more realistic.

**170 **Reply:**

We thank the reviewer for these suggestions. We have moved Figures 9-11 to Figures S7-10 in the Supplement. We also include site-level comparisons in the revised manuscript at lines 286-309. Despite the differences in the relative contributions of individual sources in S0Aopt and S0Wopt, due to the mixed-source effect and transport, the performance by S0Wopt and S0Aopt are very similar at most of sites. It is difficult to tell which one is more realistic

**175 in general. However, we do see general higher CH4 concentrations in S0Aopt than S0Wopt over the Northern Hemisphere.**

As shown in the Supplement Figure S4-5, over tropics, S0Wopt is in general better with smaller biases at KUM, POCN20, POCN25, and MID, which suggests overestimations in the Southeast Asian emissions. Also, anthropogenic emissions may be underestimated in S0Wopt at WLG and NWR and overestimated at TAP site based on the

180 comparisons of S0Aopt and S0Wopt, whereas wetland emissions may be overestimated in S0Wopt at LEF site.

Line 519: Such high correlations are a bit surprising, if I see the lines in Fig. 12. For example AGR show -ve trend, yet show positive correlation. How is that possible?

**Reply:**

- 185 The high correlation is mainly for periods of 1983-1998 and 2007-2014. In Figure 12 (now Figure 7 in the revised manuscript), the negative trend for AGR during 1999-2006 is very weak (i.e., -0.2 ppb/yr), almost no significant trend. The correlation here is more dominated by the interannual variability than linear trend. The positive correlation therefore suggests the interannual variability of CH4AGR agrees with that of total CH4.
- 190 Line 633: This is similar to the essential conclusion in some other publications as well, where ENE and Animals were made responsible for the post-2006 CH4 growth rate. I guess it is extremely difficult to separate emissions from Animals and Wetlands by 13C signature in CH4.

Reply:

We agree with the reviewer's comments that rely on methane 13C signatures only is not able to distinguish emissions from animals and wetlands.

Lines 638ff: I am curious if inconsistency between the tropospheric OH and CH4-loss by OH are arising from the spin-up. Did you spun-up the simulations using different OH from the 1970s?

Reply:

- 200 In our spin-up simulations, we drive the model with 1979 emissions for 50 spin-up years. During spin-up, OH changes every year based on the simulated chemistry. We used the spun-up atmospheric conditions for all the production runs, including low-OH and high-OH cases. In other words, we use the same initial conditions for low and high OH cases. As OH is short-lived species, the initial condition of OH has little impact on OH concentrations and trends. Based on our model tests, we find a very close relationship between tropospheric OH and lighting NOx. The changes
- 205 in CH4-OH loss are not only affected by changes in OH but also CH4. Decreases in OH levels during 2008-2015 does necessary lead to decreases in CH4-OH loss as CH4 is increasing during this time period.

Response to Referee #2 for comments on "Investigation of the global methane budget over 1980-2017 using GFDL-AM4.1" by He et al.

**We thank the reviewer for the insightful comments and suggestions on our manuscript. Below, our responses in bold text follow the reviewer's comments shown in plain text.**

The author presents an analysis of the methane global budget using an atmospheric chemistry model over almost 4 decades (1980-2017) and considering changes in both emissions and sinks. As conclusion, they provide likely scenarios of changes in sources and sinks of methane explaining the methane growth different periods of these past decades.

**215 GENERAL COMMENT**

The manuscript is well written and organize, and quite easy to follow. The methodology and different steps are well explained. Understanding the methane budget and particularly its changes is crucial to establish pathways for emission reduction. As a result, this study contributes significantly to the debate.

**220 Reply:**

225

**We thank the reviewer for the positive comments.**

However part of the result section is quite lengthy and may be shortened and more straightforward– see specific comments below. The comparisons to the observation (surface and satellite) as done here, do not really help in discriminating better scenarios.

**Reply:**

**We thank the reviewer for the suggestions. We have revised the manuscript and shortened the result section.**

- 230 Regarding the abstract and conclusion of the paper: there might not be a single driver of the recent methane growth, but a combination of different increasing/decreasing sources/sinks (Saunois et al., 2017). Different studies (as cited) may appear in disagreement as only the major driver (one source) is highlighted (in the abstract, title or conclusion) but considering the uncertainty of the estimates, this could be a combination with high uncertainty on the relative contributions; but this does not mean that there are no agreement between studies. One of the objective is to "investigate the possible driverS" of the
- 235 methane changes. So why keeping a single source as a driver? Why not considering the combination and trying to estimate the relative contribution? Also, testing different initial emissions would further strengthen the resulting range.

**Reply:**

- We agree with the reviewer that it is more likely a combination of different drivers instead of one single driver that lead to recent methane growth, which is also indicated by the model results discussed in the manuscript. The more important question is the relative contributions from these drivers, which, as mentioned by the reviewer, is highly uncertain. We have included additional discussions in the revised manuscript regarding the relative contributions. However, as mentioned in the manuscript, we need additional observational constraints (e.g., methane isotopes, ethane) to better quantify the relative contributions, which is the next step of the work.
- 245 With different initial emissions, the optimized methane emissions totals would be similar but spatial distribution and seasonality could be different. We did one sensitivity test with time-varying wetland emissions (i.e., S0Origtswet) as shown in Figures S2 and S3 in the Supplement. Using time-varying wetland emissions does not alter our results much. Based on the results comparison in Figures S2 and S3, which suggest better performance by climatological wetland emissions (i.e., S0Orig), we decide to start with this wetland emission.
- 250 As this manuscript is more focused on global totals, even with different initial emissions (e.g., different wetland emissions), after the emission optimization, the impacts would be very small on global averages, but could be large on regional scales. Also, the relative contributions would be different. However, the Aopt and Wopt sensitivities conducted in this work provide extreme scenarios for the contributions from anthropogenic sources and wetland sources. As mentioned earlier, additional observational constraints could be applied to refine the resulting ranges.

255

Finally, I would recommend publication in ACP after major revisions addressing the general and specific comments highlighted in this review.

**Reply:**

260 We thank the reviewer for the suggestions. We have revised the manuscript and addressed the comments point by point below.

SPECIFIC COMMENTS Section 1 – Introduction

265 Lines 38 to 42. Removes those lines: it's stated later and better near the end of the section.

Reply:

**We have removed these lines in the revised manuscript.**

270 Line 43- 47. Add a general reference here (e.g. Saunois et al. (2016)).

**Reply:**

**We have added the reference in the revised manuscript.**

275 Line 60-67. There have been other studies that tried to highlight some more likely scenarios combining changes from individual sources. E.g., Saunois et al., (2017) suggested a combination (with relative contributions) of changes from the different methane sources contributing to the renewed growth (increasing from fossil, agriculture and waste emissions, decrease of biomass burning, . . .).

**280 Reply:**

We have included the statement of combined changes in sources with the reference in the revised manuscript as below:

"The observed renewed growth since 2007 has been explained alternatively through increases in tropical emissions (Houweling et al., 2014; Nisbet et al., 2016) such as agricultural emissions (Schaefer et al., 2016; Patra et al., 2016) and tropical wetland emissions (Bousquet et al., 2011; Maasakkers et al., 2019), increases in fossil fuel emissions (Rice et al., 2016; Worden et al., 2017), decreases in sources compensated by decreases in sinks due to OH levels (Turner et al., 2017; Rigby et al, 2017), or a combination of changes in different sources such as increases in fossil, agriculture, and waste emissions with decreases in biomass burning emissions (Saunois et al., 2017)."

290

Section 2 – Methodology and data

Line 125. "BMB emissions contribute to IAV". . . because wetland emissions are considered constant in the study. Wetland emissions contribute a lot to methane emission IAV (Kirschke et al., 2013).

**295 Reply:**

We have clarified this statement in the revised manuscript as below:

"Although wetlands are in reality a major contributor to interannual variability in methane emissions (Bousquet et al., 2006; Kirschke et al., 2013), our use of climatological wetland emissions causes the interannual variability in our methane emissions to be dominated by BMB emissions."

Line 151. "... largely underestimate...". What is largely here? Please quantify.

**Reply:**

**305 We have evaluated the model results with initial emission inventories, which is shown in Figure S2 in the Supplement. The simulated methane DMF is about 126 ppb lower than observed CH4 DMF from NOAA GMD surface observations (with RMSE = 120 ppb).**

Line 153. "optimization simulations"? The optimization approach is described later (from line 169), but this description
 should directly follow the first mentioning of "optimization approach". Which observations are used for this "optimization"? surface marine boundary layer – described after? Please specify and put Section 2.3 Observation before Section Simulation Design.

**Reply:**

**315 Thank you for the suggestion. We have reordered the sections such that the section on Observations appears before Simulation Design.**

Line 174. The relative contributions from the different sectors are also kept as in the initial emission set up, and depend on it. As a result this highly depend on the initial set of emissions. Testing other set of initial emissions would really be valuable to assess the uncertainty related to the set up and strengthen the results. EDGARv4.3.2 instead of CEDS for example. Other

320 assess the uncertainty related to the set up and strengthen the results. EDGARv4.3.2 inste wetland emissions including IAV. . .

**Reply:**

We agree that our results depend on the individual contributions in the initial emission inventories. Using different initial emission inventories may help assess the associated uncertainties of individual sector contributions to a certain degree, but to better quantify the individual contributions and assess the uncertainties, we could apply additional observational constraints, which is our next step of the study.

Methane anthropogenic emissions from EDGARv4.3.2 are about 5-9% lower than CEDS anthropogenic emissions, mainly due to lower emissions from energy sector (Janssens-Maenhout et al., 2019). The share of ENE to total anthropogenic emissions in EDGARv4.3.2 (i.e., 33%) are also lower than those in CEDS (i.e., 38%), but both increase after 2006. We are unable to test EDGARv4.3.2 emissions in this work but will consider it for future work.

To address the reviewer's point about wetland emissions with IAV, we performed a test simulation with wetland emissions including IAV for 2001-2015 (i.e., S0Origtswet) based on an extended ensemble version of WetCharts (Bloom et al., 2017) as shown in Figures S2 and S3. The results from this simulation are very similar to those from climatological wetland emissions (i.e., S0Orig). We expect that optimization of wetland emissions will cause the original IAV in emissions to be lost.

340 Line 180. DeltaE include IAV missing from the initial emissions. In the case of SAopt, this is attributed to anthropogenic emissions, is that realistic?

**Reply:**

The IAV of methane emissions are mainly dominated by that from wetland and biomass burning. However, IAV could also exist in anthropogenic emissions due to the dependence of microbial methane sources, such as rice paddies, on soil temperature and precipitation (e.g., Knox et al., 2016). The optimization to match observations resulted in higher IAV in total emissions than in the initial emissions. Because the purpose of S0Aopt is to investigate the role of changes in total anthropogenic emissions (anthro plus BB) rather than individual sectors, we applied this IAV to all sectors which we acknowledge introduces some unrealistic IAV in the anthropogenic emissions. We chose this avancemental construct to limit the number of considivity simulations.

350 experimental construct to limit the number of sensitivity simulations.

Line 184. The reader would probably want to have fast access to FigS2. . Indeed all the discussion is on emissions from SAopt and SWopt and not on the initial emissions. That would be better to combine Fig1 and Fig S2

355

**Reply: We have combined these figures as new Fig 1 in the revised manuscript.**

Line 200. At which frequency are the observations considered? Frequency of the sampling of the model? Monthly?

360

**Since the frequency of the model output is monthly, the observations are also monthly-based.**

Section 3 Results and discussions.

365 Line 241. Testing different wetland emission data sets – with different seasonality may help to quantify the influence of wetland emission seasonality on the observed bias. OH influence could be seen in simulation S2 with higher OH: is S2 performing better?

**Reply:**

**Reply:**

370 We agree that different wetland emission datasets can help assess the impacts of wetland emission seasonality on the observed bias. We did a sensitive test with time-varying wetland emissions (i.e., S0Origtswet) for 2001-2015 as shown in Figures S2 and S3 in the Supplement. In terms of seasonality, we did not find much difference between S0Orig and S0Origtswet. In a future study, we plan to simulate wetland emissions in the land model (LM4.1) coupled to AM4.1 to better capture the spatial and temporal variability of wetland emissions than prescribed emissions.

**375**

The S2 performance is similar to S0 as we re-optimized methane emissions based on higher OH case. Higher OH case does not change the spatial and temporal variability of OH but only the magnitude of OH levels.

Line 244. Is it reasonable to state that SWopt perform a bit better that SAopt? So far no comparison is made between the two simulations.

**Reply:**

As we optimize the global total emissions instead of spatial distribution, globally, the performance of Wopt and Aopt is very similar to each other, however there are regional differences because of the differences in the spatial distribution of emissions. For example, in the Southern Hemisphere, Wopt performance is very similar to Aopt. In the Northern Hemisphere, Wopt performs better than Aopt at KUM, POCN20, ASK, and TAP sites, while it performs worse at KEY, WIS, and UTA. It is really site-specific. We have included site-specific comparisons between S0Wopt and S0Aopt in the revised manuscript as below:

- 390 "S0Aopt and S0Wopt simulate very similar surface methane DMF and their comparison with NOAA-GMD observations at individual sites show both simulations to be biased low over Southern Hemisphere sites, but the low bias decreases northward (Figure S5 in the Supplement). The simulations are biased moderately high (with RMSEs up to ~ 40 ppb) over tropical regions (e.g., POCS15, POCS10, SMO, POCS05, POCN00, CHR, and POCN05). These sites are mainly remote sites and surface methane DMF represents background methane levels. The overestimates are likely due to
- 395 overestimation of emissions over Southeast Asia (e.g., Saeki and Patra, 2017, Patra et al., 2016, and Thompson et al., 2015), which could affect these remote sites through transport. However, the model predicts surface methane DMF relatively well at Ascension Island (ASC, 8°S, 14.4°W, 85 m), which is also a remote site without impacts from East Asia. The high biases over the tropics suggest a need to improve regional emissions (e.g., Southeast Asia). Moderate overestimates also occur at Mahe Island (SEY, 4.7°S, 55.5°E), a location that could be affected by air masses from
- 400 polluted areas over the tropics and Northern Hemisphere. Over middle and high latitudes of the Northern Hemisphere, both S0Aopt and S0Wopt simulate surface methane DMF relatively well at most sites, except at Key Biscayne (KEY, 25.7°N, 80.2°W), Tae-ahn Peninsula (TAP, 36.7°N, 126.1°W), Park Falls (LEF, 45.9°N, 113.7°W), and Mace Head (MHD, 53.3°N, 9.9°W). KEY, MHD, and TAP are sampled under onshore winds, whereas LEF are affected by local sources and model transport. The high biases at these sites could be due in part to model sampling bias (e.g., model grid

- 405 box overlapping land while samples are collected at coast with onshore winds) and uncertainties in local emissions (e.g., possible overestimation in the emissions over East Asia). On the other hand, both S0Wopt and S0Aopt are able to capture monthly variations in methane at most of the sites except at LEF, where R = 0.4 for S0Wopt and 0.5 for S0Aopt, respectively. In general, both S0Wopt and S0Aopt are able to reproduce the surface methane DMF and capture the trend at most sites (e.g., with R greater than 0.5 at 98% of total sites and with RMSE less than 30 ppb at 74% of total sites). As
- 410 shown in Figure S5, S0Aopt in general better estimates methane trends and growth over low latitudes of the Southern Hemisphere (e.g., SMO) and middle/high latitudes of the Northern Hemisphere (e.g., ASK, KEY, WIS, UTA, NWR, UUA, LEF, CBA, STM, and ALT) than S0Wopt. Based on the site-level comparisons between S0Wopt and S0Aopt, anthropogenic emissions over Southeast Asia are likely overestimated in both S0Aopt and S0Wopt, while they could be underestimated at WLG and NWR in S0Wopt but be reasonably well represented in S0Aopt."

415

Line 253: The reasons of the biases compared to HIPPO are exactly the same as those seen with surface observations. See comment for line 241.

**Reply:**

**420 Please see reply to comment for line 241.**

Line 255. 2% difference with the updated GFDL-AM4.1, using optimized emissions. How much was the difference when using initial emissions? How much better is it compared to the previous version of GFDL-AM4.1?

**425 Reply:**

With initial emissions, the simulated methane profiles are about 12% lower than HIPPO measurements. For the standard version of GFDL-AM4.1, which uses prescribed methane concentrations, the simulated methane profiles are about 5% higher than HIPPO measurements in the Southern Hemisphere and 4% lower in the Northern Hemisphere.

**430**

440

Line 264. Isn't it by construction? As total emissions are optimized using surface marine boundary layer observations...

**Reply:**

The match of global means is certainly by design, but not that for latitude bands. As shown in Figure 4 in the main text, the model is still able to capture methane trends and variability at different latitude bands. We have revised the sentences as below:

"SOWopt and S0Aopt are also able to reproduce global annual mean surface methane DMF (with root-mean-squareerror (RMSE) = 10.4 ppb in S0Wopt and 11.6 ppb in S0Aopt) over 1983-2017, which is expected from emission optimization. Meanwhile, both simulations are able to reproduce the methane timeseries very well (with R = 1.0 in both S0Wopt and S0Aopt) over different latitude bands as shown in Figure 4."

Line 274. Not clear about the 1-year mismatch. There is not always a 1-year delay between spikes...

**Reply:**

445 We corrected a bug in the scripts for model evaluation and updated all the relevant plots. We then updated results and discussions in the revised manuscript as below:

"Table 3 summarizes methane growth rates during 1984-1991, 1992-1998, 1999-2006, and 2007-2017. S0Aopt and S0Wopt simulate very similar methane growth rates as their emission totals are the same. During 1984-1991, both S0Aopt

450 and S0Wopt slightly overestimate methane growth rates by ~2 ppb yr-1, possibly due to fewer available observations used for emission optimization during this time period than afterwards. After 1991, the simulated methane growth rates are in general comparable to the observations (with annual mean difference within  $\pm 1$  ppb yr-1). The major discrepancies in the simulated methane growth rates and observations occur over the tropics and high northern latitudes as shown in Figure 4. Over the tropics, both S0Aopt and S0Wopt overestimate methane growth rates (by about 5-10 ppb yr-1) during 1984-

- 455 1990 when there were limited observations available, but are able to reproduce methane growth rates relatively well afterwards. Agreement of the methane growth rate is worse in the Northern Hemisphere than in the Southern Hemisphere, especially at high northern latitudes, mainly due to the large bias during 1984-1988 and a slight shift in peak growth (or peak decrease) during 1997-2005. The number of observational MBL sites does not provide adequate coverage of the globe, especially in the 1980s, which could have different impacts on the Northern and Southern
- 460 Hemisphere when optimizing global total emissions. In general, S0Aopt estimates slightly better methane growth rates than S0Wopt, especially over 30-90° N. The biases in methane growth rates also suggest a need to refine regional emissions."

Line 275-281. Are those numbers necessary in the text? What is the point? That could be summarize in a Table for the full numbers and in the text summarize to "SWopt performs better over this period while SAopt performs better over that period". The differences in growth rate are much lower than their own range of uncertainty.

**Reply:**

**As suggested by the reviewer, we have summarized the growth rates in the table and revised these sentences in the revised manuscript (see reply to comment line 274).**

Line 283. Does it imply that the results (following results on the emissions changes) are less robust for this region? This is unfortunate as most of the methane emissions occur in the NH.

**475 Reply:**

We corrected a bug in the scripts for model evaluation and updated all the relevant plots. After the correction, both Aopt and Wopt are able to reproduce methane growth rates despite there is a slight mismatch (~1-2 years) during 1998-2008. But compared to other regions, the correlation is slightly worse over 30-90° N. This suggests a need to improve/optimize regional emissions.

**480**

Line 286. Indeed. Related to wetland emissions? Is it related to the missing IAV in wetland initial emissions?

**Reply:**

**Many reasons could lead to such biases over 30-90N. Uncertainties in the IAV of both wetland and anthropogenic emissions could lead to such bias. The biases also suggest a need to improve/optimize regional emissions.**

Line 301. Could the bias model sampling be overcome/reduced? Does the bias change with the grid box choice for coastal sites?

**490 Reply:**

Yes. If we move the grid box to the ocean side, the bias is reduced significantly. For example, at KEY and MHD sites, the RMSEs are reduced from ~90 ppb to 33 ppb and from ~50 ppb to 20 ppb, respectively.

Line 306-319. The model observation comparison using GOSAT and SCHIAMACHY is not really helpful. As the model is optimized against surface observations, it's saying that surface observation and satellite data see similar trends (but are offset by latitudinal biases), that biases exist in the satellite data (latitudinal biases), and that uncertainties in the transport model (especially in the stratosphere but not only) can explain the difference between model and satellite columns. Nothing new for an atmospheric model assessment, and this comparison does not discriminate one simulation from the other. This part may be removed and put in the supplementary material.

**500**

**Reply:**

**As suggested by the reviewer, we have moved this part to the Supplement.**

Line 325. By construction?

505

Reply:

Yes. Both simulations reproduce the global growth rates by design. But the model with optimized emissions is still able to reproduce the growth rates at different latitude bands.

510 Line 332-333. This sentence should be removed and put in Section2.

**Reply:**

**As suggested by the reviewer, we have moved this sentence to Section2.**

515 Line335\_336. This sentence could be removed with causes of interannual variability put in sentence of line 329-330.

**Reply:**

As suggested by the reviewer, we have combined these sentences as below:

- 520 "As shown in Figure 5, the optimized emissions in general increase during 1980-2017, with an annual mean of 580±34 Tg yr-1 (mean±standard deviation) and show much larger interannual variability during 1991-1993 and 1997-2000, which is likely due to the strong El Niño events during 1991-1992 and 1997-1998 as well as the Mt Pinatubo eruption in 1991 (Dlugokencky et al., 1996; Bousquet et al., 2006; Bândă et al., 2016)."
- 525 Also 1991-1992 high IAV is also related to Mt Pinatubo eruption (decrease in CH4) (ref. e.g., Banda et al., 2016, https://www.atmos-chemphys.net/16/195/2016/; Dlugokencky, E. J. et al. Changes in CH4 and CO growth rates after the eruption of Mt Pinatubo and their link with changes in tropical tropospheric UV flux. Geophys. Res. Lett. 23, 2761–2764 (1996); 21. Bousquet, P. et al. Contribution of anthropogenic and natural sources to atmospheric methane variability. Nature 443, 439–443 (2006)).

**530**

**Reply:**

**We thank the reviewer for providing the references. We have included these references into the revised manuscript.**

Line 343. Is it possible to overplot Naus et al. OH level on Figure 8?

**535**

**We have included OH anomaly from Naus et al. (2019) in the Figure 6 in the revised manuscript.**

Table 3. Single values are provided for the emission estimates while several simulations were performed. The authors compare their initial emission to the literature – which could have been done in Section2. However they stated that there initial emission were underestimate and then all the paper is on the optimized emissions. So Table 3 should compare their optimized emissions with the literature!

**Reply:**

**Reply:**

- 545 Table 3 includes numbers for optimized emissions as reflected in rows of ΔE and sum of sources. Also, for ΔE and sum of sources, we include estimated ranges based on different OH levels as well. To address the reviewer's comment, we have included estimated ranges for individual sectors based on the Aopt and Wopt under different OH levels, which is now shown in Table 4 in the revised manuscript.
- 550 Line 351-353. Total natural emissions from bottom-up estimates are much lower than top-down because there are not constrained and just an "addition" from independent individual source estimates, knowing the large uncertainty of each natural source. . . The initial emissions should be much comparable to top-down estimates, as the large source from freshwater is not included in the initial emissions (about 100Tg in the bottom up methane budget).

555 Reply:

We thank the reviewer for the explanation. We have included this discussion in the revised manuscript as below:

"Since there is no observational constraint on bottom-up estimates, total natural emissions are simply summed over independent individual sources, which could be overestimated in the bottom-up approach considering the relatively large

- 560 uncertainties in each individual source. In addition, in the bottom-up estimate from Kirschke et al. (2013) and Saunois et al. (2016), some other natural sources, such as freshwater, are not included in the initial emission inventories in this work; however, they are likely double counted in the bottom-up estimates (e.g., high-latitude inland waters are likely also considered as wetland areas) as pointed out in Saunois et al. (2019)."
- 565 Line 353-354. Not really, see comment above. The difference between the initial emission of this study and the bottom-up estimate from Kirschke et al and Saunois et al., is mainly driven by source not included in the initial emission set up (freshwater), and probably double counted in the bottom-up budget. Estimates for other natural sources from Kirschke et al., 2013 and Saunois et al., 2016 should be added. This is also due to the use of a climatological value for wetland emissions from the 2000s applied to the whole period. As IAV and trends are missing in the initial emissions, some signal is lost compared to estimates reported by Kirschke et al. and Saunois et al.

**Reply:**

**We thank the reviewer for the explanation. We have included this discussion in the revised manuscript (see reply above) .**

**575**

Line 356. Remove Saunois et al., 2016 as the study starts in 2000.

**Reply:**

**Removed.**

**580**

Line 358 and following. Well, that would be better to show the range of total anthropogenic and wetland optimized emissions and to compare these ones with the literature. Table 3. It also presents values for the more recent period. These values may be compared with the updated global methane budget recently released in ESSDD Saunois et al., 2019 (https://www.earth-syst-sci-data-discuss.net/essd-2019-128/)

**585**

**Reply:**

**We have updated Table 3 (now Table 4) with values from Saunois et al. (2019).**

Section 3.3.1. These results are quite expected knowing the distribution of anthropogenic emissions and wetland emissions. There are interesting but could be more when compared against observations. If the surface methane DMF and growth rate observed values at each sites are over plotted on these spatial distributions, does it help in discriminating which optimized emissions fit the best?

**Reply:**

595 The overlay plots have been generated as shown in Figure S7 and Figure S9 in the Supplement. But S0Aopt and S0Wopt gives very similar methane DMF and growth rates. We included site-level comparisons in Section 3.1. As also suggested by the other reviewer, we have removed this part into the Supplement.

Line 400-401. By construction?

**600**

**Reply:**

For global trends, the match of model simulations with observations are by design.

Line 406. .. are KEPT constant. . .

605

Reply: We have corrected this in the revised manuscript as below:

"Since wetland emissions and other natural emissions are kept constant every year in S0Aopt, with increases in OH levels during 1983-1998, all tagged natural tracers show a weak decreasing trend."

Line 432 and line 437. How can we explain higher sink?

**Reply:**

615 Although concentrations of tracer CH4WET decrease, with the increases in OH levels during 1999-2006, CH4WET sinks also increase, which could be higher than wetland emissions. Also, during this time period, CH4WET decreasing trend is very week (e.g., -0.6 ppb/yr). During 2007-2017, wetland emissions are lower compared to 1999-2006 (see Figure 1) while OH levels are higher. Therefore, the decreasing trend (e.g., -4.6 ppb/yr) is much larger compared to 1999-2006.

**620**

Line 453. Indeed. . . Is testing a different initial inventory an option? It would help, confirming or infirming some results. . .

**Reply:**

**See reply to General comments above.**

625

Line 454. Other sectors (than wetland?)

**Reply:**

Yes. For S0Wopt, except wetland, which is optimized, all other sources are based on the initial emission inventories.

Section 3.3.2: could this section be shortened? It's quite hard to follow. . . Instead of describing each simulation in detail (with numbers etc..), would it be shorter to conclude for each period, which sector(s) drive the changes (increase. . .) and if the two simulations agree or not? And keep the details and numbers for the supplementary. . ..

**635 Reply:**

We thank the reviewer for the suggestion. We have shortened and revised this section as Section 3.3 in the revised manuscript.

Line 457 and following. The two sensitivity tests may be presented in Section 2 as well. And included in the above discussion.

**Reply:**

As suggested by the reviewer, we have included two sensitivity tests in Section 2 as well as in Section3.3.

**645 Section 3.4 Sensitivity to OH.**

How does the model compare with the other CCMI models? (see Zhao et al., 2019 Fig 4 and 7. https://www.atmos-chem-physdiscuss.net/acp-2019-281/acp-2019-281.pdf)

**Reply:**

650 In general, our OH trend is within the range of OH trends in Zhao et al. (2019). From 1980 to 2000, OH in AM4.1 increases by 4.7%, comparable to 4.6±2.4% in Zhao et al. (2019). During 2000-2010, OH anomaly varies from -0.29 mole cm-3 to 0.34 molec cm-3.

TECHNICAL COMMENTS Line 87. 1980-2017 instead of 1980-2014

655

Reply: Thanks, corrected.

The methane sinks considered in AM4.1 include oxidation by OH radicals, Cl, and O(1D), and dry deposition. Since tThe

- model does not represent tropospheric halogen chemistry, it does not consider removal of methane by Cl in the troposphere, a sink that remains poorly constrained which has been shown to be extremely minor (Hossaini et al., 2016[VN2]; Gromov et al., 2018; Wang et al., 2019[VN3]). The dry deposition flux of methane is calculated estimated based on a monthly climatology of deposition velocities (Horowitz et al, 2003) calculated by a resistance-in-series scheme (Wesely, 1989; Hess et al., 2000) and used to mimic methane loss by soil uptake, which accounts for about 5% of the total methane sink (Kirschke et al., 2013; Saunois et al., 2016).
- In this work, we included 12 additional methane tracers tagged by source sector to attribute methane from agriculture (CH4AGR), energy (CH4ENE), industry (CH4IND), transportation (CH4TRA), residents (CH4RCO), waste (CH4WST), shipping (CH4SHP), biomass burning (CH4BMB), ocean (CH4OCN), wetland (CH4WET), termites (CH4TMI), and mud volcanoes (CH4VOL). The tracers are emitted from corresponding sources, and undergo the same chemical pathways and
- 825 dynamics as the full CH4 tracer. For analysis, we combine CH4IND, CH4TRA, CH4RCO, and CH4SHP as other anthropogenic tracers (i.e., CH4OAT), and combine CH4OCN, CH4TMI, and CH4VOL as other natural tracers (i.e., CH4ONA).

Initially the model was spun up in a 50-year run with repetitive 1979 emissions until stable atmospheric burdens of methane and tagged tracers were obtained. After the spin-up, several sets of simulations were conducted for 1980-2017 to quantify the

830 methane budget and investigate the impacts of changes in methane sources and sinks on methane abundance (see Section 2.23). All model simulations are forced with interannually-varying sea surface temperatures and sea ice from Taylor et al. (2000), prepared in support of the CMIP6 Atmospheric Model Intercomparison Project (AMIP) simulations. Horizontal winds are nudged to the National Centers for Environmental Prediction (NCEP) reanalysis (Kalnay et al., 1996) using a pressure-dependent nudging technique (Lin et al., 2012).

**835 2.2 Observations**

We evaluate the simulated methane dry-air mole fraction (DMF) against a suite of ground-based and aircraft observations to thoroughly evaluate the model simulated spatial and temporal distribution of methane. To evaluate surface CH4, we use measurements from a globally distributed network of air sampling sites maintained by the Global Monitoring Division (GMD) of the Earth System Research Laboratory at the National Oceanic and Atmospheric Administration (NOAA) (Dlugokencky et

840 al., 2018). The global estimates derived from surface measurements are based on a number of sites at remote marine sea level

locations with well-mixed marine boundary layer (MBL) to represent background methane. The locations of MBL sites are shown in Figure S1 and the information for each MBL site is listed in Table S1 in the Supplement. A function fit consisting of yearly harmonics and a polynomial trend, with fast fourier transform and low pass filtering of the residuals are applied to the monthly mean methane DMF to approximate the long-term trend and average seasonal cycle at each MBL site (Thoning

- 845 et al., 1989; Thoning, 2019). We divide the sine (latitude) at 0.05 interval VN4]s and calculate methane DMF at each latitude band based on the selected sites within that latitude band. 
[revised manuscript text omitted]
 and Table 5 summarizes the estimated linear trend for each time period along with the
- 205 dominating tracers. For S0Aopt, total methane shows a strong increasing trend of 10.5 ppb yr-1. The tagged anthropogenic tracers all show increasing trends during 1983-1998 despite the increases in OH levels, with dominant increasing trends by CH4AGR and CH4WST consistent with emission trends. Since wetland emissions and other natural emissions are kept constant every year in S0Aopt, with the increases in OH levels during 1983-1998, all tagged natural tracers show a weak decreasing trend. During 1999-2006, total methane shows a small increasing trend of 1.0 ppb yr-1, due to the increasing trends
- 210 of CH4ENE and CH4WST compensated by the decreasing trends of other source tagged tracers. The increasing trends of CH4ENE and CH4WST are mainly driven by the increases in the emissions in S0Aopt whereas the decreasing trends of other source tagged tracers are mainly determined by the increases in OH levels. During 2007-2017, total methane shows a renewed increasing trend of 5.3 ppb yr-1, dominated by a strong increasing trend of CH4ENE and smaller increasing trends of CH4AGR and CH4WST. The results from S0Aopt suggest that globally, anthropogenic tracers dominate total methane trends during the
- 215 entire simulation period. During the 1980s and 1990s, emissions from agriculture, energy, and waste sectors are the major contributors to the methane increase. During 1999-2006, wherewhen methane stabilizes, increases in methane sinks and methane sources alternatively dominate the trend for different tracers. During 2007-2017, emissions from agriculture, energy and waste sectors are the major contributors to the methane-renewed growth in methane, with energy sector as the largest contributor.
- 220 The source tagged tracers behave slightly differently in S0Wopt. For S0Wopt, total methane shows a similar increasing trend as S0Aopt. During 1983-1998, the tagged anthropogenic tracers all show increasing trends except CH4ENE, with overall smaller increasing trends than those in S0Aopt. CH4WET shows a strong increasing trend (7.0 ppb yr-1), dominating the total methane trend. This is mainly because wetland emission growth is larger than anthropogenic emission growth due to the emission optimization in S0Wopt during this period. During 1999-2006, similar to S0Aopt, total methane trend is resulted
- from the increasing trends of CH4ENE and CH4WST compensated by the decreasing trends of other source tagged tracers. During this time period, CH4WET shows a slightly decreasing trend (-0.8 ppb yr-1), mainly due to the slightly higher CH4WET sinks (226 Tg yr-1) than sources (223 Tg yr-1). During 2007-2017, the total methane trend is dominated by the increasing trends of CH4AGR, CH4ENE, and CH4WST, with CH4ENE as the largest contributor, similar to S0Aopt. On the other hand, CH4WET shows a significant decreasing trend during this period, mainly due to higher CH4WET sinks (217 Tg yr-1) than sources (206 Tg yr-1). Compared to the S0Aopt results, S0Wopt suggest CH4WET as the largest contributor for the methane trends during 1980s and 1990s, mainly due to the emission optimization of the different sectors. However, both scenarios suggest
- CH4AGR, CH4WST, and CH4ENE are the major contributors to the renewed growth, with CH4ENE as the largest contributor.

As shown in Figures 7-5 and 86, OH levels show a slightslightly decrease and methane sinks are relatively stable during 2007-2013, but large interannual variability exists during 2013-2017. Decreasing OH levels could lead to increases in methane

- 235 lifetime and therefore methane build\_up. Combined with increases in the emissions, methane starts to increase again during this period. However, it is difficult to separate the contributions from methane emissions and sinks as optimized methane emissions are based on methane mass balance (e.g., changes in the methane loss would act as a feedback on estimates of optimized total emissions). HoweverNevertheless, it is clear that the decrease in OH levels alone (e.g., if emissions are kept constant) would not be enough to reproduce the renewed growth. The remaining question then is which emission sector(s) is
- (are) the major contributor(s) to the renewed growth over 2007 to 2017. Both S0Wopt and S0Aopt suggest that agriculture, waste and energy sectors is the major sector contributingare the major contributors to renewed methane growth. However, both cases depend largely on the initial emission inventory. For example, S0Wopt relies on the emission growth of other sectors from the initial emission inventory, which means if the emission growth of a certain sector is overestimated or underestimated in the initial emission inventory, it would give a different resultlead to different results. Therefore, we conducted two additional sensitivity simulations (i.e., S0A06 and S0Comb as described in Section 2.3) with different emission
- growths for anthropogenic and wetland sectors as in S0Aopt and S0Wopt for 2006-2014. Based on evidence from isotopic composition (δ13CH4), recent studies suggest increasing wetland emissions may be responsible for the renewed growth of methane (Dlugokencky et al., 2009; Nisbet et al., 2016). To test this hypothesis in our modeling framework, we conducted another sensitivity simulation for 2006-2014, by repeating 2006 anthropogenic emissions
- 250 for all the years but adjusting wetland emissions to ensure that the total methane emissions are the same as in S0Wopt (or S0Aopt), which would imply that the increases in methane emissions are only due to the increases in wetland emissions. This sensitivity simulation is referred to as "S0A06" and the The trends for source tagged tracers and total methane by S0A06 and S0Comb are shown in Figure S5 in the supplement4 and Table 5. Interestingly, in S0A06, where anthropogenic and biomass burning emissions are kept constant every year for 2006-2014,- anthropogenic tracers still show an-increasing trends during

[revised manuscript text omitted]
 decreasing trende from 1980 to 2007  $(-0.04 \text{ vear vr}^{-1} \text{ in S0}, -0.05 \text{ vear vr}^{-1} \text{ in S1}, \text{ and } -0.03 \text{ vear vr}^{-1} \text{ in S2})$ , a clear increasing trend during 2011-2015 (0.08 vear vr}^{-1} \text{ in S2}). 1 in all three simulations), and a decreasing trend during 2015-2017(-0.2 year  $vr^{-1}$  in all three simulations). The mean 315 tropospheric methane lifetime due to OH loss for 1980-2017 is 9.9±0.4 years in S0Wopt, which is about 0.5 year lower than S1Wopt (10.4 $\pm$ 0.5 years), and about 0.7 year higher than S2Wopt (9.2  $\pm$ 0.3 years), due to different OH levels and therefore

methane sinks, but with similar methane burdens. This indicates that a 1% change in OH levels could lead to about 0.08 year difference in the tropospheric methane lifetime. The mean tropospheric methane lifetimes simulated by the three simulations are is within the uncertainty range of observation-derived estimates for the 2000s (Prather et al., 2012) and model estimates

- 320 (Voulgarakis et al., 2013; Naik et al., 2013). All simulations show an increase in methane lifetime during 2011-2015, which could be a signal of the methane feedback on its lifetime (Holmes, 2018) in the model. Continued increases in methane emissions (Figure 75) during this time period, along with decreases in tropospheric OH concentrations (Figure 138), lengthen the lifetime of methane and therefore amplify the methane's response to emission changes. If methane emissions continue to increase, we can expect stronger increases in atmospheric methane due to the amplifying effect of the methane-OH feedback as occurred during in the significant increases in methane growth rates during 2014 and 2015.
- 325

330

**4** Conclusions**

[revised manuscript text omitted]

|                           | 539 [411-671]°                                 | 571 [521-621]°                           | 604 [483-738] °                                          |                              |                      |                      |
|---------------------------|-------------------------------------------------------|-------------------------------------------------|-----------------------------------------------------------------|------------------------------|----------------------|----------------------|
|                           |                                                       |                                                 | $\frac{505 [459-516]^d}{505 [480 740]^e}$                       |                              |                      |                      |
|                           | 442 [419-476]                                         | 486 [462-519]                                   | 526 [502-559]                                                   |                              |                      |                      |
| OH loss            | 468 [382-567]°                                 | 479 [457-501]°                           | 528 [454-617] c
553 [476-677] e        | 534 [510-567]         | 519 [495-552] | 542 [519-576] |
|                           | 38                                             | 47                                       | 43                                                       |                              |                      |                      |
| O1D loss           | 46 [16-67]°                                    | 67 [51-83]°                              | 51 [16-84] °
31 [12-37] °                      | 43                    | 44            | 42            |
|                           | 5                                              | 7                                        | 7                                                        | 7                     | 8             | 7             |
| CI loss            | 25 [13-37] °                                   | 25 [13-37] °                             | 25 [13-37] e
11 [1-35] e |                              |                      |                      |
|                           | 13                                             | 14                                       | 14                                                       | 14                    | 14            | 14            |
| Soila                     | $\frac{21 [10-27]^{\text{b}}}{28 [0, 47]^{\text{c}}}$ | $\frac{27 [27-27]^{\circ}}{28 [0, 47]^{\circ}}$ | $\frac{32 [26-42]^{\circ}}{28 [0, 47]^{\circ}}$                 | 38 [27-45] ª          |                      |                      |
| 50115              | 28 [9-47]*                                            | 28 [9-47]                                | $\frac{28[9-47]^2}{34[27-41]^d}$                                |                              |                      |                      |
|                           |                                                       |                                                 | 30 [11-49] e                                  |                              |                      |                      |
| Totalsg |                                                       |                                                 |                                                                 |                              |                      |                      |
|                           | 539 [515-571]                                  | 574 [549-608]                            | 595 [572-628]                                            | 605 [582-640]         | 589 [565-625] | 620 [597-653] |
| Sum of                    | 551 [500-592] 6                     | 554 [529-596] o               | 548 [526-569] 6                               | 572 [538-593]ª        |                      |                      |
| sources                   | 663 [536-789]c                      | 649 [511-812] °                          | $\frac{678}{542}$                                               | /3/[593-880] e        |                      |                      |
|                           |                                                       |                                                 | $\frac{343[322-339]^2}{703[570-842]^2}$                         |                              |                      |                      |
|                           | 499 [475-532]                                         | 554 [530-586]                                   | 591 [567-624]                                                   | 598 [574-632]                | 584 [560-617]        | 606 [582-639]        |
|                           | 511 [460-559] b                            | 542 [518-579] b                      | 540 [514-560] b                                      | 556 [501-574] d   |                      |                      |
| Sum of sinks              | 539 [420-718]°                                        | 596 [530-668]°                           | 632 [592-785]°                                           |                              |                      |                      |
|                           |                                                       |                                                 | 540 [486-556]d                                |                              |                      |                      |
|                           |                                                       |                                                 | 625 [600-798] e                               |                              |                      |                      |
| Turketen                  | 40[39-40]                                             | 20 [19-22]                               | $\frac{4[4-5]}{2}$                                              | $\frac{7[8-8]}{16[6,47]}$    | 5 5-8                | 14 [15-14]    |
| Impaiance                 | 30 [16-40]6                         | 12[/-1/]8                     | 8 [-4-19]
4 [-11-36] d                     | 10 [U-4/]a |                      |                      |
|                           | 36                                                    | 19                                              | 4.8                                                             | 7.4                          | 3.5                  | 16.6-17.2            |
| Atmospheric
growth     | 34 b                                | $1\overline{7^{b,h}}$                           | 6 b,h                                                |                              | $1.9 \pm 1.6^{h}$    | $18.9 \pm 1.7^{h}$   |
|                    | 32 n                                       |                                                 |                                                                 |                              |                      |                      |

a The decadal mean values are based on initial emission inventories. The lower and upper limits of the ranges are based on the minimum and maximum among all the optimized emission scenarios (i.e., S0Aopt, S0Wopt, S1Aopt, S1Wopt, S2Aopt, S2Aopt, S0Wopt, S1Aopt, S1Wopt, S2Aopt, S0Wopt, S1Aopt, S1Wopt, S2Aopt, S1Wopt, S2Aopt, S1Wopt, S2Aopt, S1Wopt, S1Wopt, S2Aopt, S1Wopt, S2Aopt, S1Wopt, S1Wopt, S1Wopt, S2Aopt, S1Wopt, S1Wopt,

and S2Wopt) conducted in this work.
 bValues are based on Kirschke et al. (2013) top-down approach.
 cValues are based on Kirschke et al. (2013) bottom-up ap